# Matrix Quantum Mechanics and Entanglement Entropy: A Review

**DOI:** 10.3390/e28010058

**Published:** 2025-12-31

**Authors:** Jackson R. Fliss, Alexander Frenkel

**Affiliations:** 1Department of Applied Mathematics and Theoretical Physics, University of Cambridge, Cambridge CB3 0WA, UK; 2Physique Théoretique et Mathématique, Université Libre de Bruxelles & International Solvay Institutes, CP 231, 1050 Brussels, Belgium; 3Simons Center for Geometry and Physics, Stony Brook University, Stony Brook, NY 11794, USA; afrenkel@scgp.stonybrook.edu

**Keywords:** matrix quantum mechanics, entanglement entropy, large *N*

## Abstract

We review aspects of entanglement entropy in the quantum mechanics of N×N matrices, i.e., matrix quantum mechanics (MQM), at large *N*. In doing so, we review standard models of MQM and their relation to string theory, D-brane physics, and emergent non-commutative geometries. We overview, in generality, definitions of subsystems and entanglement entropies in theories with gauge redundancy and discuss the additional structure required for definining subsystems in MQMs possessing a U(N) gauge redundancy. In connecting these subsystems to non-commutative geometry, we review several works on ‘target space entanglement,’ and entanglement in non-commutative field theories, highlighting the conditions in which target space entanglement entropy displays an ‘area law’ at large *N*. We summarize several example calculations of entanglement entropy in non-commutative geometries and MQMs. We review recent work in connecting the area law entanglement of MQM to the Ryu–Takayanagi formula, highlighting the conditions in which U(N) invariance implies a minimal area formula for the entanglement entropy at large *N*. Finally, we make comments on open questions and research directions.

## 1. Notational Conventions

Due to possible conflation of matrix notations and quantum operator notations, for the reader’s convenience, a summary of our notational conventions follows.

### 1.1. Indices

Classical N×N matrices act on an auxiliary ‘color’ space, which we take to be $C^{N}$. Indices on color space will by indexed by *a*, *b*, or *c* (or dressed variants), e.g., a matrix, *X*, could be written in components as $X_{ab}$.The second type of index runs over the total number, *D*, of Hermitian matrices in our system. Such indices will be taken from *i*, *j*, *k*, … (in the event where the matrix theory is the low energy effective theory of D-branes, *i* runs over the dimension of the target space of the string theory) (for example, in Section 3.2, the BFSS model is a theory with nine (bosonic) matrices $X^{i}$, with *i* running from 1 to 9).The third type of index runs over states of a quantum Hilbert space, H, of the theory. Such indices are labeled *m*, *n*, *p*, ….

We will use an Einstein summation convention with indices when it does not cause confusion; it will be tacitly assumed that repeated indices are summed over with a Kronecker delta. When this is not the case, such as the contraction with a metric tensor, we will denote so explicitly. Lastly, to avoid confusion with indices, we will denote with a sans-serif font the imaginary unit:(1)$$\sqrt{-1}:=i\;.$$

### 1.2. Bras and Kets

It will be useful to represent matrices as acting on states of the color space so we will distinguish these from quantum states spanning the Hilbert space.

Color space bras and kets will use a round bracket notation, e.g., |·) for kets and (·| for bras. A matrix, *X*, then may be written as follows:(2)$$X=\sum_{a,b=1}^{N}X_{ab}|a)(b|\;.$$Hilbert space kets and bras will utilize the standard quantum mechanics notation, |·〉 and 〈·|, respectively.

### 1.3. Traces

Traces over the color space, $C^{N}$, are denoted with an upper case ‘T’:(3)$$\mathrm{Tr}X:=\sum_{a=1}^{N}X_{aa}\;.$$Traces over a quantum Hilbert space are denoted with a lower case ‘t’ as ‘tr’. When necessary to distinguish the Hilbert space, we will do so with a subscript, e.g., the following:(4)$${\mathrm{tr}}_{H}O:=\sum_{|n\rangle \in H}\langle n|O|n\rangle \;.$$

### 1.4. Commutators

Matrix algebras and non-commutative algebras in general, will have commutators expressed in terms of ordinary brackets, $\left[a_{1},a_{2}\right]=a_{1}a_{2}-a_{2}a_{1}$. For non-commutative functions, we might explicit denote the non-commutative product, e.g., ${\left[\cdot ,\cdot \right]}_{★}$, although we will often drop this notation when it is understood. For matrices, [·,·] is always the commutator with respect to matrix multiplication.Commutators for operators on a quantum Hilbert space will use a ‘double bracket’ notation, 〚·,·〛.

### 1.5. Subsystems

Subsystems, in general, will be denoted by Σ which could indicate a spacelike subregion or a more abstract division of degrees of freedom. Its complement is denoted as $\bar{\Sigma }$. We may also refer to a subspace of color space and we reserve the symbol *M* to refer to the dimension of such a subspace. It is never used to refer to a matrix. In the event that there is only a single matrix degree of freedom, we denote it as *X*.

## 2. Introduction

Insights into the non-perturbative nature of quantum gravity are vexingly rare. It is, therefore, somewhat miraculous that two deep insights arose a half-century ago within the span of a few years. The first, arising out of a series of papers between 1973 and 1975 [1,2,3,4,5], was that black holes are thermodynamic objects with a temperature, an entropy, and a spectrum of radiation. The second, in July of 1974, was ‘t Hooft’s formulation of the planar limit of large *N* QCD [6]. Although within their original context these two results seem disparate, in retrospect, we may appreciate that the two works have given us deeper, non-perturbative, understanding of a common puzzle—the microscopic connection between entropy and geometry.

At the center of the flurry of activity in understanding black hole thermodynamics [1,2,3,4,5,7] was the identification of the event horizon area with the microcanonical entropy,(5)$$S=\frac{A}{4G_{N}}\;.$$
Soon afterwards, Bekenstein proposed the generalized second law of thermodynamics [8], in which the sum of horizon area and the entropy of all matter exterior to the horizon never decreases. The generalized second law is compatible with Hawking’s realization that black holes radiate away energy due to quantum effects [3] because of the entropy increase of the pair production [4]. Bekenstein further conjectured that $\frac{A}{4G_{N}}$ is in fact an upper bound on the entropy able to be contained in a region of space before a black hole is formed [8]. This provided some initial hints of the deep connection between information and energy density that is a common theme in our modern understanding of quantum gravity. It was subsequently realized that the black hole horizon area can be thought of as an entanglement entropy between the fundamental degrees of freedom consituting the black hole and the rest of the world [9]. That this entanglement entropy is reproducible (up to ambiguity in the overall UV cutoff, which is conjectured to renormalize Newton’s constant) by the entanglement of low-energy effective field theory modes inside and outside the horizon lends credence to the idea that the low-energy dynamics in the black hole interior may be recovered from a microscopic theory consisting of some fundamental boundary degrees of freedom.

On the other hand, it took nearly two decades—until the second superstring revolution and the crystallization of holography—for the relevance of the large *N* limit of matrix systems to black hole physics to be fully appreciated. In [10], the first example of an explicit duality between a string theory (defined by an explicit worldsheet action) and a large *N* matrix theory was found—the c=1 matrix model. Shortly after, a family of dualities between interacting D*p*-brane systems, with low-energy effective field theories described by a d=p+1 dimensional super Yang–Mills (SYM), and quantum gravity in asymptotically Anti-de Sitter (AdS) spacetimes with a negative cosmological constant was proposed [11,12,13,14,15,16]. The most famous example of this family is 4dN=4 super Yang–Mills dualilty with quantum gravity in ${\mathrm{AdS}}_{5}\times S^{5}$—this is an instantiation of the far more general AdS/CFT correspondence. (More generally, the d≠4 cases, being non-conformal, describe a broader class of non-AdS dualities known as “D-brane holography.”)

Since then, many pieces of intuition have emerged for how exactly a fluctuating spacetime emerges from the degrees of freedom of its holographic dual. We touch on only a couple of them here. One of the best-established correspondences is the observation of Ryu and Takayanagi that entanglement entropy of geometric subregions on the boundary is given by minimal-area surfaces in the bulk [17]. The minimal-area surface is a direct probe of bulk geometry, and subregion-subregion duality (the correspondence between a specific boundary subregion and a specific bulk subregion) provides a picture of how the degrees of freedom of the boundary rearrange themselves into the bulk.

This standard picture of holography, where the deep interior of emergent geometry emerges from the IR of the holographic dual and the boundary is emergent from the UV, cannot be the whole story. This intuition only describes the radial direction of the emergent geometry, but d=p+1 SYM is dual to a gravitational system with 9−p emergent dimensions, as the bulk dual is always the 10-dimensional background of superstring theory. We, therefore, need an additional mechanism to describe the compact dimensions (like the $S^{5}$ of ${\mathrm{AdS}}_{5}\times S^{5}$) that fill out the additional dimensions of the emergent string theory.

This additional intuition comes from matrix eigenvalues—the large *N* theories with known holographic duals have matrix degrees of freedom (as we expect from ‘t Hooft’s argument). The matrices describe branes and the configuration of open strings between them, and the eigenvalues of the matrices correspond to the distribution of branes along a particular dimension. Close to a stack of branes, within an order 1 number of string lengths, the branes will fluctuate away from the center of mass of the stack, giving some width to the eigenvalue distribution of the matrices. In the c=1 matrix model, the fluctuations of the eigenvalue distribution build up an emergent geometry, as we explain further in Section 3.1.

Taken together, these two pictures of emergent geometry in holography complement each other. Geometry above the curvature scale scale seems to emerge from the energy spectrum and entanglement structure of the boundary theory, in a manner almost agnostic as to the precise field content of the dual theory, whereas geometry below the energy scale seems to emerge from the N×N adjoint degrees of freedom directly. This disjoint picture is not fully satisfying. It would be great to find a unifying picture of the emergence of geometry, where the entanglement structure of the matrix degrees of freedom gives rise to geometry below the AdS lengthscale, just like the CFT entanglement structure gives rise to geometry above the AdS lengthscale. This article reviews recent progress towards this goal—developments in understanding subalgebras and area-law entanglement structure emerging from the quantum mechanics of matrix systems, i.e., *matrix quantum mechanics* (MQM), that have *no* spatial base space geometry to speak of.

### 2.1. MQM and String Theory

The relationship between large *N* theories and string theory begins with ‘t Hooft’s seminal work on the planar limit of QCD [6,18,19,20]. The crux of the idea is as follows:evident if we consider a matrix integral of a single Hermitian N×N matrix, *X*, that transforms in the Adjoint representation of U(N),(6)$$X\to U\;X\;U^{†}\;,\;U\in U\left(N\right)\;.$$
See [21] for a detailed description of these types of integrals and their relationship to graph theory. The partition function defining this matrix integral is as follows: (7)$$\begin{array}{cc} \; & Z\left(N,g\right):=\int d^{N^{2}}X\exp\left[-S\left(X;N,g\right)\right]\;,\;S\left(X;N,g\right):=\frac{N}{g}\mathrm{Tr}\left[-\frac{1}{2}X^{2}-\;X^{4}\right]\;. \end{array}$$
*g* is coupling constant and large *g* is a strong coupling regime of this model. The propagator of the free, quadratic theory is easy to compute:(8)$$\langle X_{ab}X_{cd}^{*}\rangle =\frac{g}{N}\delta_{ac}\delta_{bd}.$$

To perform perturbation theory, we build diagrams using the standard double-line notation introduced by ‘t Hooft (Figure 1):

Any given Feynman diagram is a graph with some number of vertices $N_{V}$, edges (the propagators) $N_{E}$, and faces (closed loops representing the sum over an index) $N_{F}$. We now note that every face corresponds to a closed loop, and so adds an overall factor of *N*, essentially from a trace of $\delta_{ab}$. Every edge corresponds to a factor of the propagator, so adds an overall factor of $N^{-1}g$. Every vertex corresponds to a factor of the coupling, and so adds an overall factor of $g^{-1}N$. The total weight of a Feynman diagram is, therefore, proportional to $g^{N_{E}-N_{V}}N^{N_{V}-N_{E}+N_{F}}=g^{N_{E}-N_{V}}N^{\chi }$, where χ is the Euler characteristic of the graph formed by the Feynman diagram. See Figure 2 for an example Feynman diagram.

The ‘t Hooft diagrams begin to resemble smooth worldsheets when the typical number of faces per diagram grows large. We, therefore, want to understand the regime of the parameters g,N, for which this becomes true. This regime is model-dependent. Consider the model in (Equation 7), which only has a quartic interaction, and, therefore, only generates diagrams with four-valent vertices. This determines the number of vertices in terms of the number of edges: there are two vertices touching each edge, four edges touching each vertex, so we have $2N_{E}=4N_{V}$. For the topology of Euler characteristic χ, this further implies $N_{F}=\chi +N_{V}$, which, for a fixed genus, may be truncated to just $N_{F}=N_{V}+O\left(1/N_{V}\right)$ as the number of vertices gets large. Call $\Gamma_{4}\left(N_{F}\right)$ the set of four-valent graphs with exactly $N_{F}$ faces. We can, therefore, compute that the typical number of faces per graph scales at both large *N* and $N_{F}$ as (using ∼ to signify equality up to 1/N corrections and overall O(1) constants):(9)$$\langle N_{F}\rangle ∼\frac{\sum_{\gamma \in \Gamma_{4}\left(N_{F}\right)}N_{F}g^{N_{E}-N_{V}}N^{\chi }}{\sum_{\gamma \in \Gamma_{4}\left(N_{F}\right)}g^{N_{E}-N_{V}}N^{\chi }}∼\frac{\sum_{N_{F}}N_{F}\left|\Gamma_{4}\left(N_{F}\right)\right|g^{N_{F}}N^{\chi }}{\sum_{N_{F}}\left|\Gamma_{4}\left(N_{F}\right)\right|g^{N_{F}}N^{\chi }}.$$
To make progress, we need to know the asymptotic number of four-valent graphs:(10)$$|\Gamma_{4}\left(N_{F}\right)|∼CN_{F}^{-7/2}\kappa_{0}^{N_{F}},$$
where $\kappa_{0}\approx 4.1$ is a known constant [19,20,22] and *C* is some overall order one coefficient. We, therefore, have (at large *N*, where the sphere dominates) the following:(11)$$\langle N_{F}\rangle ∼C\frac{\sum_{N_{F}}N_{F}^{-5/2}{\left(\kappa_{0}g\right)}^{N_{F}}}{\sum_{N_{F}}N_{F}^{-7/2}{\left(\kappa_{0}g\right)}^{N_{F}}}∼C{\left(\log g+\log\kappa_{0}\right)}^{-1}.$$
We may recognize $\kappa_{0}^{-1}$ as a critical coupling where the number of faces per diagram diverges. To recover ‘smooth’ worldsheets, we then take the limit N→∞, $g\to \kappa_{0}^{-1}$. This is known as the double scaling limit. It is interesting that the double scaling limit is essentially determined by graph combinatorics, as in (Equation 11). We will revisit this exact same double scaling limit in Section 3.1 when we more fully describe the c=1 matrix model.

We reiterate that, in this double scaling limit, the number of faces of the dominant Feynman diagrams contributing at fixed topology diverges and so smooth geometries emerge essentially from the loop expansion of large *N* matrices. This picture and ‘t Hooft’s argument predates the complementary picture of the matrix quantum mechanics present in super Yang–Mills theories as the low-energy effective worldvolume theories of D-branes, a result which emerged during the second superstring revolution [23,24]. Just like in the c=1 matrix model, positions of D-branes in target space are associated to the matrix eigenvalues, with each matrix (typically denoted $X^{i}$) corresponding to a spacetime dimension orthogonal to the brane worldvolume. A familiar family of actions for such stacks of branes are SYM, and in [12,13] these were argued to be dual to a family of asymptotically AdS spacetimes. Specifically, the low-energy theory of D*p* branes gives rise to quantum gravity in asymptotically ${\mathrm{AdS}}_{p+2}S^{10-p-2}$. For p=3, where SYM is conformal, the emergent spacetime is exactly ${\mathrm{AdS}}_{5}\times S^{5}$ [12].

### 2.2. Susskind–Uglum and Open String Edge Modes

In [25], Susskind and Uglum argue that the main contribution to entanglement entropy across some horizon in string theory is due to the endpoints of open strings anchored to the horizon (see Figure 3). Accordingly, they conjecture is that the global Hilbert space of closed string theory may be expressed as follows:(12)$$H_{\mathrm{closed}}\subset H_{\mathrm{open},\Sigma }⊗H_{\mathrm{open},\bar{\Sigma }},$$
where Σ and $\bar{\Sigma }$ are complementary subregions. This contribution to the entanglement is extremely reminiscent of the contribution of entanglement edge modes in gauge theories with Wilson line observables that may intersect the entanglement cut [26,27,28,29]. The argument for this comes from computing the string partition function in a spacetime with a conical deficit, and applying Tseytlin’s prescriptions [30,31,32,33] to extract the contribution to the effective action. They find that the action is dominated by strings punctured by the conical deficit, which, when sliced along Euclidean time, become open strings anchored to a horizon (see Figure 3).

It is reasonable to guess that any non-perturbative definition of string theory, such as a large *N* matrix theory, must somehow reproduce this picture. In the context of thermal states of matrix quantum mechanics, which are thought to be dual to black hole geometries in the emergent geometry [34], we may consider the SU(N) symmetry of the Lagrangian to be global, as opposed to gauged. Such ‘ungauged’ MQM models have been studied extensively, especially in the context of c=1 [35,36,37,38,39,40], and there is a large body of evidence that not gauging SU(N) does not spoil the holographic interpretation of MQM models [41]. In these contexts a tantalizing picture emerges directly from the ‘t Hooft expansion in the thermal partition function: vortices open up on the worldsheet, wrapping the thermal circle (see Figure 4 and Figure 5).

To understand better why this happens, we compare the thermal path integral of the gauged and ungauged theories:(13)$$\begin{array}{cc} \; & Z_{\mathrm{ungauged}}=\int {\left[DX\left(\tau \right)\right]}_{X(\beta )=X(0)}\exp\left(-\int_{0}^{\beta }d\tau \;L\left(X,\dot{X}\right)\right),\\ \; & Z_{\mathrm{gauged}}=\int_{U(N)}dU\int {\left[DX\left(\tau \right)\right]}_{X\left(\beta \right)=UX\left(0\right)U^{†}}\exp\left(-\int_{0}^{\beta }d\tau \;L\left(X,\dot{X}\right)\right)\;. \end{array}$$
The ungauged model has the simple periodic boundary condition X(β)=X(0), whereas the gauged model includes an additional integral over a unitary matrix U, as any identification $X\left(\beta \right)=UX\left(0\right)U^{†}$ is a valid periodic configuration. We may now study the gauged model in a basis where U is diagonal, with *N* eigenvalues $e^{i\theta_{a}}$, a∈{1,…,N}, so that the measure over the dU integral is transformed to the following: (14)$$dU\to \prod_{a\neq b}\sin\left(\frac{\theta_{a}-\theta_{b}}{2}\right)d^{N}\theta \;.$$
In this basis, the periodic boundary conditions are simply as follows:(15)$$x_{ab}\left(\beta \right)=e^{i(\theta_{a}-\theta_{b})}x_{ab}\left(0\right)\;,$$
which means that the propagator corresponding to an ‘t Hooft line carrying index *a* that wraps around the thermal circle (as in Figure 4) comes with an overall factor of $e^{i\theta_{a}}$. Because we are integrating over $e^{i\theta_{a}}$, the value of any diagram that has such an index loop (that is not canceled by some corresponding factor of $e^{-i\theta_{a}}$ due to a line cycling in the opposite direction) is nullified.

However, in the ungauged model, there is no such factor of $e^{i\theta_{a}}$, meaning that the ‘t Hooft diagram is free to be ‘punctured’ by the thermal circle, as in Figure 4, creating vortices on the worldsheet [38,40]. If we take the temperature large, and, therefore, β small, a BKT phase transition occurs [40,42] and vortices proliferate. This is qualitatively similar to the Hagedorn transition described in [43,44], and Kazakov, Kostov, Kutasov, and Tseytlin argued exactly corresponds to universal physics regarding black hole formation [39,40].

We are left with a striking qualitative and quantitative similarity between Figure 5 and Figure 3. Specifically, it appears that the non-singlet sector of matrix quantum mechanics and the Susskind–Uglum open-string edge modes responsible for black hole area-law entanglement (or indeed, entanglement across any Ryu–Takayanagi horizon [17]). It, therefore, appears plausible that the non-singlet sector of matrix quantum mechanics, in the large *N* limit, reproduces the physics of the open strings anchored to an entangling surface.

The Hilbert space structure posited by Susskind and Uglum, (Equation 12), as well as the relation between gauged and un-gauged U(N) symmetries in the microscopic MQM, are indicative of a broader structure of entanglement in gauge theories. As we will explain further in this review, this is the structure of embedding a gauge-invariant set of states into an extended Hilbert space carrying representations of gauge redundancies promoted to a global symmetry. In local gauge theories, these are known commonly as “edge modes” and the appearance of area laws in their entanglement is tied to their being localized to an entangling surface. The amazing aspect of MQM is that the structure of the area law is not a priori and instead emerges from the large *N* behavior of this entanglement. In this review, we will collect and reports various results in this direction in MQMs relevant for string theory (such as the c=1 matrix model) as well as D-brane physics and even condensed matter physics. Along the way, we will establish necessary backgrounds so that the contexts and imports of these models are understood and the machinery necessary for a self-contained understanding of the key results.

## 3. A Brisk Review of MQM Models in String and M-Theory

In a theory of matrix quantum mechanics (MQM), the classical degrees of freedom of the systems are N×N Hermitian matrices $X^{i}$, with labels *i* running from 1 to *D*. The number of matrices is often denoted *D*—as we will see in the examples below, it is related to the number of spatial dimensions in the emergent geometry. Roughly speaking, the eigenvalue distribution of $X^{i}$ contains the data of the spatial geometry along the *i*-th direction.

The matrices $X^{i}$ admit an adjoint action of the unitary group U(N) as follows:(16)$$X^{i}\to U\;X^{i}\;U^{†}.$$
We consider Lagrangians symmetric under this action, typically of the single trace form so that an ‘t Hooft expansion of the type described in the introduction applies: (17)$$L=\mathrm{Tr}[\sum_{i}{\dot{X}}_{i}^{2}-V\left(X^{i}\right)]\;,$$
where $V(X^{i})$ is some analytic function of the $X^{i}$. Once we quantize the theory, the generators of the symmetry (Equation 16) are as follows:(18)$$G_{ab}=2i\sum_{i}{\left(:\left[X^{i},\Pi^{i}\right]:\right)}_{ab},\;$$
with $\Pi^{i}$ the conjugate momentum to $X^{i}$. The normal ordering symbol places the position $X_{ab}^{i}$ operators to the left of the momentum $\Pi_{ab}^{i}$ operators.

The Hilbert space of the theory is the space of square-integrable functions of the matrices, $\psi (X^{i})$. The symmetry (Equation 16) may be taken to be global or gauged. The difference is whether or not we project onto the subspace of Hilbert space that is annihilated by all of the generators (Equation 18). As we highlighted in Section 2.2, both choices produce interesting stringy physics [41], with non-singlets often associated to sources of open strings.

In the remainder of this section, we will review some standard examples of MQM systems relevant to stringy physics, to build up some intuition for how geometry emerges from matrices. In Section 3.1, we review perhaps the simplest model of string theory emergent from MQM, the c=1 matrix model (so called because the target space is 1 + 1 dimensional). In Section 3.2 and Section 3.2.3, we review the “BFSS” matrix quantum mechanics and its “BMN” deformation—proposals for a non-perturbative definition of M theory or type IIA string theory in certain backgrounds, depending on the regime.

### 3.1. The c=1 Matrix Model

As described in the introduction, the c=1 matrix model is one of the simplest instantiations of holography, wherein an exactly solvable quantum system is dual to a precisely known worldsheet theory. There are several clear reviews on the subject [18,19,45,46,47]. On the quantum mechanics side, the model consists of a single matrix *X* with Lagrangian: (19)$$L_{c=1}=N\mathrm{Tr}[{\dot{X}}^{2}-V\left(X\right)].$$
The potential V(X) for the model resembles that of an inverted harmonic oscillator stabilized by a right-side-up quartic term (the actual potential used to stabilize the inverted harmonic oscillator is not important for recovering low energy fluctuations of the emergent string theory (see e.g., [48], which even deviates from a purely single-trace potential). Early proposals studied an unstable version that considered a cubic stabilizing interaction $\frac{g}{\sqrt{N}}\mathrm{Tr}[X^{3}]$ [10], but more recent work [49,50] suggests that non-perturbative effects are best captured by the quartic form): $V\left(X\right)=-\frac{1}{2}X^{2}+gX^{4}$ (see Figure 6).

The dual worldsheet Lagrangian is that of strings propagating in a two dimensional target space (The nomenclature c=1 refers to the free boson representing target space time T(z), but the spacelike Liouville wall direction ϕ(z) is another target space dimension eating up the additional 25 units of central charge), stabilized by a Liouville wall which prevents the theory from going to strong coupling. The action is string units is as follows: (20)$$S_{c=1,w.s.}=\frac{1}{4\pi }\int d^{2}\sigma \left({\left(\partial \phi \right)}^{2}+{\left(\partial T\right)}^{2}+QR\phi +\mu \;e^{2b\phi }\right).$$
*T* and ϕ are both real scalar fields, R is the worldsheet Ricci scalar, and μ a coupling constant. The other constants satisfy the following(21)$$b=\frac{Q}{2}-\sqrt{\frac{Q^{2}}{4}-1}$$
so that the vertex operator is marginal and the following applies:(22)$$c_{\mathrm{total}}=1+3Q^{2}=26\;,$$
so that the theory is on-shell. We will discuss the evidence for this dual description to (Equation 19) shortly.

The Lagrangian (Equation 19) is often most conveniently studied by using the U(N) symmetry to diagonalize *X* and passing to the collective field description [51,52]. First, we fix the gauge to diagonalize *X*:(23)$$X=U\;\Lambda \;U^{†}\;,\;\Lambda =\mathrm{diag}\left(\lambda_{1},\lambda_{2},\ldots ,\lambda_{N}\right)\;.$$
We then integrate out the gauge redundancy U. The measure on the eigenvalues has induced on it a Van der Monde determinant:(24)$$∆\left(\lambda \right):=\prod_{a<b}\left(\lambda_{a}-\lambda_{b}\right),$$
and we write the partition function as follows: (25)$$Z_{c=1}=\int \prod_{a}D^{N}\lambda_{a}\;∆{\left(\lambda \right)}^{2}\;\exp\left(i\int dt\left(\sum_{a}{\dot{\lambda }}_{a}^{2}+\lambda_{a}^{2}-\frac{g}{N}\lambda_{a}^{4}\right)\right).$$
The most straightforward way to analyze this system is through the Schrödinger equation implied by this path integral [18]:(26)$$\frac{1}{∆(\lambda )}\sum_{a}\frac{\partial^{2}}{\partial \lambda_{a}^{2}}\left(∆\left(\lambda \right)\psi \left(\lambda_{a}\right)\right)+∆\left(\lambda \right)\left(-\lambda_{a}^{2}+\frac{g}{N}\lambda_{a}^{4}\right)\psi \left(\lambda_{i}\right).$$
If we introduce the new wavefunction:(27)$$\Psi \left(\lambda_{a}\right):=∆\left(\lambda \right)\psi \left(\lambda_{a}\right),$$
then $\Psi (\lambda_{a})$ exactly satisfies the Schrödinger equation for *N* non-interacting particles on a line that obey a Pauli exclusion principle, i.e., spin-less fermions. The potential seen by the fermions is exactly the same potential *V* seen by the matrices, e.g., the following:(28)$$V_{c=1}\left(\lambda \right)=-\lambda^{2}+\frac{g}{N}\lambda^{4}\;,$$
in the case of the c=1 model. Thus, the fermions see an inverted harmonic oscillator potential corrected by a quartic term.

The emergent holographic direction, identified with the eigenvalue parameter λ, is most manifest in the collective field description (see [47] for a crisp pedagogical review). The collective field ρ(λ) rewrites the system in terms of the fluctuations of an eigenvalue density and momentum density:(29)$$\rho \left(\lambda \right)=\sum_{a}\delta \left(\lambda -\lambda_{a}\right),\;\Pi \left(\lambda \right):=\sum_{a}p_{a}\delta \left(\lambda -\lambda_{a}\right).$$
When considered as operators, we have the following commutation relation: (30)$$\begin{array}{cc} \left[\rho \left(\lambda \right),\Pi \left(\lambda^{\prime }\right)\right] & =i\sum_{a}\delta \left(\lambda^{\prime }-\lambda_{a}\right)\frac{d}{d\lambda }\delta \left(\lambda -\lambda_{a}\right)=i\int d\tilde{\lambda }\;\rho \left(\tilde{\lambda }\right)\delta \left(\lambda^{\prime }-\tilde{\lambda }\right)\frac{d}{d\lambda }\delta \left(\lambda -\tilde{\lambda }\right)\\ \; & =i\rho \left(\lambda^{\prime }\right)\frac{d}{d\lambda }\delta \left(\lambda -\lambda^{\prime }\right). \end{array}$$

Since the eigenvalues behave as fermions states, they fill up the potential well up to some chemical potential, μ. The notational coincidence to μ appearing in (Equation 20) is intentional, and the physics about this Fermi surface can be matched to string scattering off the exponential wall of (Equation 20). More specifically, the region of the eigenvalue space matched to scattering amplitudes in string theory is the ‘double-scaling’ limit, where we zoom in very close to the crest of the inverted harmonic oscillator (see Figure 7). This is the limit in which large ‘t Hooft diagrams dominate, so begin to resemble a smooth worldsheet, as described in Section 2.1. Specifically, the double scaling limit is taken by sending *N* to infinity while keeping μ, the chemical potential for the eigenvalues, fixed.

The ground state eigenvalue distribution in the double scaling limit, near the crest of the inverted harmonic oscillator, may be extracted as an analytic continuation of the familiar Wigner semicircle for the standard matrix harmonic oscillator [38]:(31)$$\rho_{0}\left(\lambda \right)\approx \frac{1}{4\pi }\sqrt{\lambda -2\mu },\;\lambda \ll \sqrt{N}.$$
Because the number of eigenvalues N=∫dλρ(λ) is conserved, we may parameterize low-energy fluctuations as follows:(32)$$\rho \left(\lambda \right)=\rho_{0}\left(\lambda \right)+\partial_{\lambda }\eta \left(\lambda \right),$$
where the zero mode of η(λ) is pure gauge. Expressed in terms of the eigenvalues, η(λ) is as follows:(33)$$\eta \left(\lambda \right)=\sum_{a}\theta \left(\lambda -\lambda_{a}\right)-\int_{-\infty }^{\lambda }\lambda^{\prime }\rho_{0}\left(\lambda^{\prime }\right).$$
We end up with the Hamiltonian (see [47] for a detailed derivation): (34)$$H=\int d\lambda \;\left(\rho_{0}\left(\lambda \right)\left(\Pi_{\lambda }^{2}-{\left(\partial_{\lambda }\eta \right)}^{2}\right)+\frac{1}{2}\left(\Pi_{\lambda }^{2}\partial_{\lambda }\eta +\frac{1}{3}{\left(\partial_{\lambda }\eta \right)}^{3}\right)\right).$$
Despite starting with a theory of free fermions, we have found that the collective field description contains a cubic interaction term with an effective coupling that scales inversely with the ground state eigenvalue distribution $\rho_{0}\left(\lambda \right)$. As demonstrated in [49,50,53,54], this target space physics is reproduced by the string field theory of (Equation 20).

Above we have treated U(N) as a gauge redundancy to diagonalize *X*. Non-singlet excitations, allowed when the overall U(N) symmetry of the model is global instead of gauged, are known to be described by long open strings propagating in the c=1 string theory target space [35,55]. Highly excited states in the non-singlet sector cause a subset of the eigenvalues to clump together and form a metastable bound state which resembles a black hole [39,40,42,56].

### 3.2. BFSS

In 1996, Banks, Fischler, Shenker, and Susskind proposed a matrix model dual to M-theory compactified on a nearly (the original paper took a light-like limit of the compact direction, whereas in [57] it was pointed out that the backreaction of the D-branes rendered the compact direction spacelike at any finite distance from the origin) light-like direction [11,57,58,59,60]. The theory consists of nine bosonic matrices, $X^{i}$ and sixteen fermionic matrices, $\Psi^{\alpha }$, governed by the Lagrangian: (35)$$L_{\mathrm{BFSS}}=\frac{N}{g_{\mathrm{YM}}^{2}}\mathrm{Tr}[\sum_{i}{\dot{X}}_{i}^{2}+2\sum_{\alpha }\Psi^{\alpha †}{\dot{\Psi }}^{\alpha }-\frac{1}{2}\sum_{i\neq j}{\left[X^{i},X^{j}\right]}^{2}-2\sum_{i,\alpha }\Psi^{\alpha †}\gamma_{i}\left[X^{i},\Psi^{\alpha }\right]]\;.$$
$\gamma_{i}$ are the standard 32×32 gamma matrices, as befitting 16-component spinors. There are two distinct limits of (Equation 35) that we can consider, depending on how the overall coupling $g_{\mathrm{YM}}$ scales with *N*. The original version of the proposal built on studies of scattering amplitudes of D0-branes in string theory, and took $g_{\mathrm{YM}}^{2}$ to be of order *N*.

Instead of following the historical development of this model, we will first review it from the more modern perspective of holography and the decoupling limit in Section 3.2.1, before returning to give some of the original intuition in Section 3.2.2, in which $g_{\mathrm{YM}}$ is taken not to scale with *N*, remaining O(1) as we take *N* to infinity. We also catalog (but do not review in detail (but see [61] for a recent review.) some recent interesting developments.

The Lagrangian (Equation 35) has 32 supercharges, and is a dimensional reduction of N=1 10d SYM (also, consequently, a dimensional reduction of N=4 4d SYM). In particular, the commutator squared term $\mathrm{Tr}[{\left[X_{i},X_{j}\right]}^{2}]$ may be recognized as the zero mode piece of the covariant derivative $D_{i}A_{j}-D_{j}A_{i}$ of the ten dimensional theory. The supersymmetry transformations are as follows [11]:(36)$$\begin{array}{cc} \; & \delta X_{i}=-2\epsilon^{T}\gamma^{i}\theta ,\\ \; & \delta \theta =\frac{1}{2}\left[D_{t}X^{i}\gamma_{i}+\gamma_{-}+\frac{1}{2}\left[X^{i},X^{j}\right]\gamma_{ij}\right]\epsilon ,\\ \; & \delta A=-2\epsilon^{T}\theta . \end{array}$$

#### 3.2.1. The Decoupling Limit

In his seminal work [12], Maldacena considered the decoupling limit of a stack of D3-branes in 10-dimensional flat space. Here, this limit refers to the decoupling between the open string fields propagating on the D-brane worldvolume and the closed string fields propagating in the ambient space the branes are embedded in. Roughly, we take a limit where string modes propagating deep into the bulk, away from the branes, are ‘gapped out’, while open strings stretching between branes retain a finite mass (see Figure 8 for a cartoon).

Miraculously, in order to take the decoupling limit we scale the couplings of the worldsheet theory exactly as we would in the ‘t Hooft diagrammatic expansion, with *g* scaling as $N^{-1}$. The decoupling limit is derived by starting with the Dirac–Born–Infeld action (on flat branes with worldvolume metric $\eta_{\mu \nu }$) [62](37)$$S=-\int d^{p}x\mathrm{Tr}(\sqrt{\det(\eta_{\mu \nu }+F_{\mu \nu })}),$$
and expanding to quadratic order in F to recover SYM the $\alpha^{\prime }\to 0$ limit ($\alpha^{\prime }$ controls the derivative expansion, equivalent to an expansion in powers of F). For clarity, in (Equation 37) the trace Tr is over the color indices, and the determinant det is over the *p* worldvolume indices. This limit must be taken carefully, and for D0-branes takes the following form:(38)$$g_{\mathrm{YM}}^{2}=\frac{1}{4\pi^{2}}\frac{g_{s}}{{\alpha^{\prime }}^{3/2}}=\mathrm{fixed}\;\mathrm{as}\;\alpha^{\prime }\to 0\;.$$
We consider the fluctuation of D-branes whose distance from the origin, *r*, is set by the following limit:(39)$$U=\frac{r}{\alpha^{\prime }}=\mathrm{fixed}\;.$$
At this length-scale target space geometry is determined by the backreaction of the branes on the metric [63]: (40)$$\begin{array}{cc} \; & ds^{2}=\alpha^{\prime }\left(-\frac{U^{7/2}}{4\pi^{2}g_{\mathrm{YM}}^{2}\sqrt{15\pi N}}dt^{2}+\frac{4\pi^{2}g_{\mathrm{YM}}\sqrt{15\pi N}}{U^{7/2}}dU^{2}+\frac{4\pi^{2}g_{\mathrm{YM}}\sqrt{15\pi N}}{U^{3/2}}d\Omega^{2}\right),\\ \; & e^{\phi }=4\pi^{2}g_{\mathrm{YM}}^{2}{\left(\frac{240\pi^{5}g_{\mathrm{YM}}^{2}N}{U^{7}}\right)}^{3/4}\;, \end{array}$$
where $d\Omega^{2}$ is the line element on the remaining sphere. Note that the dilaton, $e^{\phi }$, is small at large radius and large at small radius. The geometry is thus strongly curved but classical in the IR, and weakly curved but quantum in the UV. A useful cartoon is given in Figure 9, in which we qualitatively sketch the radius $R_{11}\left(r\right)$ of the compact 11th dimension as a function of the radial coordinate of the other ten dimensions (the original 10 dimensions the D0 branes propagate in). Recently, it was pointed out that this bulk dual (and, as a result, BFSS) has a scale similarity [64].

#### 3.2.2. The Flat Space Limit

The proposal in the original BFSS paper considered a different limit of the theory, in which $g_{\mathrm{YM}}^{2}$ scales linearly with *N* as we take the infinite *N* limit. This version of the theory was obtained by considering a non-commutative regularization of a membrane propagating in 11 dimensions [11].

To gain some intuition for the commutator squared interaction, first consider two 2×2 matrices *X* and *Y* evolving under the Hamiltonian:(41)$$H=\Pi_{X}^{2}+\Pi_{Y}^{2}+{\left[X,Y\right]}^{2}.$$
If we diagonalize matrix *X* into eigenvalues $x_{1}$ and $x_{2}$, the interaction term becomes the following:(42)$$\mathrm{Tr}\left({\left[X,Y\right]}^{2}\right)=\sum_{ab}{\left|y_{ab}\right|}^{2}{\left(x_{a}-x_{b}\right)}^{2}.$$
The off-diagonal *Y* entries $y_{12}$ see a harmonic potential with frequency $\omega_{12}=\left|x_{1}-x_{2}\right|$. When $x_{1}$ and $x_{2}$ are well separated, we can integrate $y_{12}$ out in a Born-Oppenheimer approximation. The resulting low-energy interaction between $x_{1}$ and $x_{2}$ is determined by the ground state energy of this harmonic oscillator, $\frac{1}{2}\omega =\frac{1}{2}\left|x_{1}-x_{2}\right|$. The eigenvalues of *X* see an attractive force linear in the separation, exactly as we would expect from a string stretching between two well-separated D-branes.

If we add an adjoint fermion, Ψ, then a term such as $\mathrm{Tr}[\Psi^{†}\left[X,\Psi \right]]$ creates a potential for the fermions that becomes the following: (43)$$\mathrm{Tr}[\Psi^{†}\left[X,\Psi \right]]=\sum_{ab}\left(x_{a}-x_{b}\right)\left({\bar{\Psi }}_{ab}\right)\Psi_{ab}.$$
Again, this has exactly the appropriate form of a fermionic oscillator. Excited states of the fermions, at large separations, will have energy linear in the eigenvalue separation. Consequently, (Equation 43) has exactly the correct structure to cancel the ground state energy of (Equation 42). Consequently, in the full BFSS model (Equation 35), if we take separated clumps of eigenvalues (in a configuration where the matrices nearly commute), their attractive force tends to zero as their separation tends to infinity. In this sense, the flat directions of the commutator squared interaction are no longer lifted, as in the purely bosonic model.

It, therefore, may be surprising that (Equation 35) is conjectured to have a bound ground state. This conjecture has been tested to at least N=3 [65]. It is an unusual bound state, with the eigenvalue distribution falling off as a power law instead of exponentially [57]. In fact, the ground state is conjectured to be the *only* bound state, below a continuum of scattering states that form the remainder of the spectrum. This scattering behavior in the spectrum is part of the intuition for why BFSS reproduces flat space physics in the large *N* limit (Asymptotically AdS spaces, for example, have infinitely many bound states).

Further intuition can be obtained by considering scattering amplitudes of well-separated eigenvalues [66]. Early work, part of the inspiration for the BFSS conjecture, recovered graviton scattering amplitudes by scattering well-separated clumps of D0-branes under the BFSS interaction. More recently, the soft graviton limit studied in [67,68,69] has provided evidence that 11-dimensional Lorentz invariance is recovered in the large *N* limit. Refs. [70,71] have extended the match between bulk and boundary to three and higher point functions using T-duality to the matrix string [14,15].

Furthermore, it is intersting that black hole states can be studied in this limit, and are associated to thermal sub-blocks of the matrices [72,73,74].

#### 3.2.3. BMN Mass Deformation

In 1998, Berenstein, Maldacena, and Nastase considered a version of BFSS deformed by a adding a mass term for the nine matrices [75]. This matrix model is found by considering an M2 brane propagating in a pp-wave M-theory background, as opposed to flat space M-theory in the BFSS case. In order to preserve supersymmetry, this term breaks the SO(9) symmetry into SO(6)×SO(3) by giving different masses to six of the matrices. To emphasize this explicit breaking of the symmetry, we reserve the labels $X^{i}$ for the set of three, and take $\phi^{i}$ to represent the set of six. The BMN Lagrangian is as follows: (44)$$L_{\mathrm{BMN}}=L_{\mathrm{BFSS}}-\mathrm{Tr}[{\left(\frac{\mu }{3}\right)}^{2}X^{i2}+{\left(\frac{\mu }{6}\right)}^{2}\phi^{i2}+\frac{\mu }{4}\Psi^{†}\gamma_{123}\Psi +i\frac{\mu }{3}\sum_{i,j,k}\mathrm{Tr}\left[X^{i}X^{j}X^{k}\right]].$$

Because the Yang–Mills coupling in 0+1 dimensions has a negative beta function, the commutator squared interaction dominates at low energies and the theory is approximately free at high energies. The dual geometry is, therefore, approximately (Equation 40) deep in the interior, at small *r*, and a strongly-coupled deviation away at large *r*.

The added mass terms in (Equation 44) manifestly lift the flat directions of the (classical) BFSS model. This significantly affects the structure of the excited states of the theory—just like in the harmonic oscillator, all excited states are now bound states. The BPS physics of the BMN matrix integral has been studied using supersymmetric localization [76], and was found to be dominated by the three matrices $X^{i}$ localizing to “fuzzy sphere configurations” (we will describe the fuzzy sphere background further in Section 4).

Just like in the c=1 model, non-singlet excitations in the BMN matrix model have been found to correspond to long strings anchored to the asymptotic boundary [41].

#### 3.2.4. Mini-BMN

A truncated (but still supersymmetric) version of BMN, called mini-BMN, was studied in [77,78,79], and its entanglement spectrum considered in [80,81]. It essentially just keeps the three $X^{i}$ matrices of (Equation 44) and a single matrix of spinors λ: (45)$$\begin{array}{cc} L_{\mathrm{Mini}\text{-}\mathrm{BMN}}= & \mathrm{Tr}\left[{\left(D_{t}X^{i}\right)}^{2}-\frac{1}{4}{\left(i\;\nu \epsilon^{ijk}X^{k}-\left[X^{i},X^{j}\right]\right)}^{2}\right.\\ \; & \left.\;\;+\lambda^{†}D_{t}\lambda -\lambda^{†}\sigma^{k}\left[X^{k},\lambda \right]+\frac{3}{2}\nu \lambda^{†}\lambda \right]. \end{array}$$
This model lacks the clean relationship to string theory present in [13] in terms of a derivation from a stack of branes and a decoupling limit, but it shares many of the same technical and qualitative features in a more tractable form. It is easier to simulate [79], and its ground state wavefunction has been studied numerically [80].

The Lagrangian (Equation 45) is designed to have as its classical minima configurations of the $X^{i}$ satisfying the su(2) algebra:(46)$$\left[X^{i},X^{j}\right]=i\;\nu \epsilon^{ijk}X^{k}.$$
This is precisely the defining relation of a non-commutative sphere [82] that we will describe in more detail in Section 4. The simplicity of (Equation 45) and its fluctuations around its classical saddles allows for great technical control, and makes it an interesting toy model to further study matrix theories with holographic properties.

### 3.3. The Big Picture: Eigenvalues and Target Space Geometry

There is a common theme to the emergence of target space fluctuations from matrices—the way the matrix eigenvalues are distributed in target space are related in some way to the target space geometry [83,84,85,86]. In particular, if there is a saddle point eigenvalue distribution emergent at large *N*, it encodes the structure of a classical background in target space.

The relationship between the value of a matrix eigenvalue and a geometric location in target space appears to hold whether we consider the eigenvalue fluctuations of c=1, matrices localizing to a non-commutative target space background such as in Mini-BMN [80] (see Section 4), or even richer holographic theories such as BFSS or BMN. In BFSS or BMN, the Einstein gravity region only appears in a window computed as the radius below which there is a substantial eigenvalue density. This was estimated in [57] to scale as follows:(47)$$g_{s}^{1/3}{\alpha^{\prime }}^{1/2}<r<N^{1/3}g_{s}^{1/3}{\alpha^{\prime }}^{1/2}\;.$$
In fact, this scale exactly matches the estimate for the size of the eigenvalue distribution, estimate in [57] as follows:(48)$$\sqrt{\frac{1}{N}\sum_{i}\mathrm{Tr}[{\left(X^{i}\right)}^{2}]}∼N^{1/3}.$$

The relationship between eigenvalues and target space geometry then sets the stage for why we are interested in entanglement from both the microscopic perspective, as correlations between individual collections of matrix elements, as well the target space perspective, i.e., how those correlations are realized in the dual geometric description. There are by now several different directions and results exploring how these two perspectives are related and the aim of this review article is to summarize and contextualize these results.

## 4. Non-Commutative Geometries and Matrices

A key feature of theories of MQM and one that will play a central role in the target space interpretations of the entanglement entropy in MQM is the interplay between matrices, D-brane physics, and non-commutative geometries.

The connection of MQM and D0-branes has been referenced several times in the introductory sections, and here we more fully expand on that connection. Namely, in configurations in which all $X^{i}$ commute, we may simultaneously diagonalize them. Each matrix then is a collection of *N* real numbers, $\left\{X_{ab}^{i}\right\}\to \left\{x_{a}^{i}\delta_{ab}\right\}$, which we can view as the embedding coordinates of *N* D0-branes in a *D*-dimensional target space. Non-commutativity of the matrices arises from open strings attached to and stretching between differing D0-branes. In certain cases, these strings can coalesce and broaden, or ‘blow up,’ a collection of point-like D0-branes into a higher dimensional membrane [87]. These membranes then display non-commutative features in their low-energy effective field theories owing to their origin of non-commutating matrix backgrounds. More broadly, non-commutative field theories have long played a role in the low energy descriptions of D-brane physics where the non-commutativity is mediated by string interactions [88]. Non-commutative physics also features in descriptions of the quantum Hall effect where non-commutativity arises from a strong background magnetic field and mediates anyonic statistics of particle exchange; we will describe below the connections of this system to MQM as well.

In what follows we will introduce what is meant by a non-commutative or ‘fuzzy’ space and their connections to MQM, including those introduced in Section 3. A good review of non-commutative field theory is given by [89] (and for the first (to our knowledge) historical mention of non-commutative spactime physics see [90]). This section will set the stage for the target space interpretation of MQM entanglement in the remaining sections of this review.

Non-commutative geometries are manifolds equipped with a non-Abelian ★-product on functions. Acting on the coordinate functions themselves yields a non-commutative structure with an associated non-commutativity parameter (more specfically, the single parameter θ can be defined as $\theta^{ij}:=\theta \;\theta_{0}^{ij}$ where $\theta^{ij}$ is a fiducial Poisson structure [91]), θ, with units of [length]^2^:(49)$${\left[x^{i},x^{j}\right]}_{★}:=x^{i}★x^{j}-x^{j}★x^{i}=i\;\theta^{ij}\left(x\right)\;.$$
The prototypical example is the Moyal-star product acting on functions of two variables [92]:(50)$$\left(f_{1}★f_{2}\right)\left(x,y\right):=e^{\frac{i}{2}\left(\partial_{\alpha_{1}}\partial_{\beta_{2}}-\partial_{\beta_{1}}\partial_{\alpha_{2}}\right)}f_{1}\left(x+\alpha_{1},y+\beta_{1}\right)f_{2}\left(x+\alpha_{2},y+\beta_{2}\right){|}_{\alpha_{1,2}=\beta_{1,2}=0}\;,$$
which defines a non-commutative plane described in more detail below. This example also highlights the connection between non-commutative spaces and geometric quantization of phase spaces [91]. We will denote non-commutative manifolds with a subscript θ, e.g., $M_{\theta }$. For the sake of brevity of notation, for the rest of this section, we will suppress the explicit ‘★’ notation, with it understood tacitly that products and commutators are defined with respect to it.

Commutators possess a Leibniz rule:(51)$$\left[f_{1},f_{2}f_{3}\right]=\left[f_{1},f_{2}\right]f_{3}+f_{2}\left[f_{1},f_{3}\right]\;,$$
and so defines a natural differential structure on a non-commutative manifold. Namely, we can define a non-commutative derivative, $\partial_{a}$, acting on functions implicitly through the following:(52)$$i\;\theta^{ij}\partial_{j}f:=\left[x^{i},f\right]\;.$$
This naturally defines a scalar Laplacian on $M_{\theta }$ through the following:(53)$$\nabla^{2}f:=-\delta_{ij}\left[x^{i},\left[x^{j},f\right]\right]\;.$$
In the commutative limit, i.e., leading order in θ as θ→0, we can express this as $\nabla^{2}\left(\cdot \right)=\frac{1}{\sqrt{G}}\partial_{i}\left(\sqrt{G}\;G^{ij}\partial_{j}\left(\cdot \right)\right)$ with $G_{ij}$, an effective metric on M which may differ from the standard metric on M, e.g., one induced by its embedding in flat space [91].

A practical way of working with non-commutative geometries is to find a representation for their algebra of functions. These will often be matrix representations, however possibly infinite dimensional if the manifold is non-compact (as in the canonical example of the non-commutative plane, as we review below) (that is to say the algebra of functions is a von-Neumann algebra of Type-$I_{N}$ when finite or Type-$I_{\infty }$ when infinite). When we have a matrix representation of the algebra of functions on a non-commutative space, which we denote as f→F, the ★-multiplication can be regarded as the usual matrix multiplication. Additionally the trace over that representation provides a notion of integration over $M_{\theta }$: namely, for a matrix *F* representing *f*, the following: (54)$$\frac{1}{N}\mathrm{Tr}[F]=:\int_{M_{\theta }}d^{D}x\;\mu \left(\vec{x}\right)\;f\left(\vec{x}\right)\;,$$
where $\mu (\vec{x})$ is a unit-volume measure that in the commutative limit agrees with $\sqrt{G}$ up to a constant (see [93] for a precise definition). This notion of integration plays nicely with our definition of derivation, (Equation 52), as the cyclicity of the trace implies the following: (55)$$\frac{1}{N}\mathrm{Tr}\left(\left[x^{i},f_{1}\right]f_{2}\right)=-\frac{1}{N}\mathrm{Tr}\left(f_{1}\left[x^{i},f_{2}\right]\right)$$
which we regard as a notion of ‘integration by parts.’

### 4.1. Volume Preserving Diffeomorphisms, Symplectomorphisms, and UV/IR Mixing

A surprising feature of non-commutative geometries is the emergence of symplectomorphism invariance (or volume preserving diffeomorphism (to avoid conflation with the ‘area’ of area law entanglement, we will use the term ‘volume preserving diffeomorphisms’ in the sense of two-dimensional volume) invariance in the case of two dimensions) in the U(N) gauge group at large *N*. This correspondence has been known for some time [94,95]; however, it can be simply understood from the relation between traces and integration underneath the Moyal map, (Equation 54). At large *N*, the non-commutativity of manifold, $M_{\theta }$, is small and we may regard $X\to \tilde{X}=U\;X\;U^{†}$ for U∈U(N) as a map, $x\to \tilde{x}\left(x\right)$ of commuting coordinates. The preservation of the trace of any polynomial function of matrices, F[X] under conjugation of U(N) implies a preservation of any polynomial function, f(x), under the corresponding map: (56)$$\frac{1}{N}\mathrm{Tr}\left[F(X)\right]=\frac{1}{N}\mathrm{Tr}\left[F(U^{†}XU)\right]\;\Leftrightarrow \;\int d^{D}x\;\mu \left(\vec{x}\right)f\left(\vec{x}\right)=\int d^{D}x\;\mu \left(\vec{x}\right)f\left(\vec{\tilde{x}}\left(\vec{x}\right)\right)\;,$$
which can only be true if $x\to \tilde{x}\left(\vec{x}\right)$ preserves the volume form, $\mu (\vec{x})$ itself.

We can see this map in more explicit detail from the structure of infinitesimal gauge transformations $U=e^{i\epsilon H}$:(57)$$U\;X^{i}\;U^{†}\approx X^{i}+i\;\epsilon \left[H,X^{i}\right]-\frac{\epsilon^{2}}{2}\left[H,\left[H,X^{i}\right]\right]+\ldots$$
which under the Moyal map becomes the following:(58)$${\tilde{x}}^{i}=x^{i}-\epsilon \;\theta^{ij}\partial_{j}h\left(x\right)+\frac{\epsilon^{2}}{2}\theta^{ij}\theta^{kl}\partial_{k}h\partial_{j}\partial_{l}h\ldots$$
This has the structure of an infinitesimal volume preserving diffeomorphism. For instance, the generic infinitesimal form of a volume preserving diffeomorphism of the real plane, $R^{2}$, sends $\left(x,y\right)\to \left(\tilde{x},\tilde{y}\right)$ to the following:(59)$$\begin{array}{cc} \tilde{x}= & x-\epsilon \;\partial_{y}h+\frac{\epsilon^{2}}{2}\left(\partial_{x}\partial_{y}h\;\partial_{y}h-\partial_{x}h\partial_{y}^{2}h\right)+\ldots \;,\;\\ \tilde{y}= & y+\epsilon \;\partial_{x}h+\frac{\epsilon^{2}}{2}\left(\partial_{x}\partial_{y}h\;\partial_{x}h-\partial_{x}^{2}h\partial_{y}h\right)+\ldots \;, \end{array}$$
for some function *h*, which can be seen as a specific instance of (Equation 58) with $\theta^{ij}=\epsilon^{ij}$ (we will return to this example shortly). More generally, the geometric analog of the infinitesimal changes of basis (Equation 57) are infinitesimal changes of coordinates that are symplectomorphisms due to the antisymmetry of $\theta^{ij}$. It is in the specific cases of two dimensional fuzzy geometries that we wind up with volume preserving diffeomorphisms.

While the perturbative example makes the matching explicit, it is clear from the argument given above Equation (Equation 56) that the relationship between basis changes and coordinate changes is not just perturbative. A basis may be specified by choosing some polynomial $f(X^{i})$ of the matrices $X^{i}$ to diagonalize. In this basis, we think of the fuzzy geometry as being decomposed into regions of unit area localized to slices of the geometry with constant $f(X^{i})$ coordinate. Different choices of weakly curved coordinate systems (corresponding to slowly-varying coordinate functions $f(X^{i})$) are not necessarily perturbatively close to one another, so the subset of U(N) rotations that take slowly-varying coordinates to other slowly-varying coordinates have a good interpretation as volume preserving diffeomorphisms in the emergent geometry; see Figure 10 for a cartoon.

The connection to symplecticomorphisms also highlights another surprising feature of non-commutative geometries, which is a form of *UV/IR mixing.* Symplectomorphisms are more commonly seen as maps on phase space variables preserving the symplectic form. Quantum mechanically, these preserve canonical commutators and their associated Heisenberg uncertainty relations. The non-commutativity of coordinate functions indicates a similar uncertainty, however in the coordinate locations themselves. This is a form of UV/IR mixing: short wavelength modes along one direction are tied to long wavelength modes in a mutually non-commutating direction. This is preserved under symplectomorphisms. In the matrix language, we may use U(N) to diagonalize a polynomial $f(X^{i})$ to make its eigenvalues the definite level sets of a function, $f(x^{i})$, however directions along those level sets will be ‘smeared out.’

Lastly, it is important to emphasize that the large *N* limit of the U(N) coordinate-like transformations is *not* just symplectomorphisms—U(∞) is much bigger, allowing for transformations that preserve the area but change the topology of subregions of the fuzzy manifold in question (see [97] and the recent discussion in [98]).

### 4.2. Examples

Below, let us describe some basic non-commutative geometries, as well as their matrix representations. In what follows, we will use lower-case letters to describe abstract non-commutative relations while specific matrix realisations will be denoted with capital letters.

#### 4.2.1. The Non-Commutative Plane

As basic example of a non-commutative geometry is the non-commutative plane, $R_{\theta }^{2}≃C_{\theta }$, spanned by coordinate function {x,y}, whose algebra of functions is generated by the following relation:(60)[x,y]=iθ.
This is also known as the Heisenberg algebra and is the familiar relation between position and momentum of a quantum particle, with θ playing the role of *ℏ*. It is no coincidence that the geometric quantization of the complex plane through the Moyal star product leads precisely to the canonical quantization of a quantum particle. It is obvious that (Equation 60) cannot be represented finitely: if so, taking a trace of the left-hand side would yield zero, while the right-hand side would yield the dimension of the representation, a contradiction. However, we can still represent (Equation 60) formally as an infinite dimensional yet discrete representation: unsurprisingly, this is isomorphic to the representation of harmonic oscillator creation and annihilation operators:(61)$$Z=\sum_{j=0}^{\infty }\sqrt{j}\;|j)(j+1|,\;X=\sqrt{\frac{\theta }{2}}\left(Z+Z^{†}\right),\;Y=i\sqrt{\frac{\theta }{2}}\left(Z^{†}-Z\right).$$

#### 4.2.2. The Non-Commutative Disc

One simple way for modification to the non-commutative plane algebra to be able support a finite dimensional representation is to append to (Equation 60) the outer product of a vector, ϕ:(62)$${\left[x,y\right]}_{ab}=i\;\theta \;\delta_{ab}-i\;\phi_{a}\phi_{b}^{†}\;.$$
This allows us to evade the trace argument and (Equation 62) admits a representation in terms of rank *N* matrices when the norm of ϕ obeys the following:(63)$${\left|\phi \right|}^{2}=\sum_{a}\phi_{a}^{†}\phi_{a}=\theta \;N\;.$$
In this case, a suitable *N*-dimensional matrix representation of (Equation 62) mirrors the harmonic oscillator representation of the plane, e.g., with *x* and *y* defined as (Equation 61), however with oscillator occupation numbers (this is an *N* dimensional representation because the 0-occuptation number is also a state) truncated to N−1:(64)$$Z=\sum_{j=0}^{N-2}\sqrt{j}\;|j)(j+1|\;,\;\phi =\sqrt{\theta \;N}\left|N-1\right)\;.$$
We note that, in this representation, the radius squared operator is diagonal in occupation number:(65)$$\begin{array}{cc} R^{2}=X^{2}+Y^{2} & =\theta \left(2Z^{†}Z+1\right)-\phi \;\phi^{†}\\ \; & =\theta \left[\sum_{j=0}^{N-2}\left(2j+1\right)|j)(j|+\left(N-1\right)|N-1)(N-1|\right]\;, \end{array}$$
and so we can think of |j)(j| as a projector onto a radial shell of width $\sqrt{\theta }$ that is maximally uncertain in the angular direction. The maximum occupation number indicates that there is a maximum radius of $r∼\sqrt{N\theta }$, which we can think of as the boundary of a non-commutative, or fuzzy, disc. Note then that $\phi \phi^{†}=\theta N|N-1)(N-1|$ is a projector that lives at the edge of this disc. This relation between U(N) fundamentals and boundaries of non-commutative spaces is a motif that will reoccur in Section 7.2.

#### 4.2.3. The Non-Commutative Torus

To understand how finite dimensional matrix representations can arise from compact non-commutative manifolds, we can consider the double compactification of the non-commutative plane by the following identifications:(66)$$x∼x+R_{1}\;,\;y∼y+R_{2}\;,$$
while maintaining the relation (Equation 60). Because of the compactifications (Equation 66), polynomial of *x* and *y* are no longer single-valued functions on the geometry and so we should instead consider functions of the following:(67)$$u=e^{i\frac{2\pi x}{R_{1}}}\;,\;v=e^{i\frac{2\pi y}{R_{2}}}\;.$$
The exponentiation of (Equation 60) then yields the definining relation of the *fuzzy torus*, $T_{\theta }^{2}$ [99,100,101]:(68)$$uv=e^{-i\frac{4\pi^{2}}{R_{1}R_{2}}\theta }vu.$$
When $\frac{4\pi^{2}}{R_{1}R_{2}}\theta$ is rational (and without loss of generality we can take it to be 1/N), then (Equation 68) admits a finite dimensional representation, (u,v)→(U,V) which are the *N*-dimensional clock and shift operators:(69)$$U=\sum_{k=1}^{N}|k+1)(k|\;,\;V=\sum_{k=1}^{N}e^{i\frac{2\pi }{N}\left(k-1\right)}|k)(k|\;,\;\left(\mathrm{with}\;|N+1):=|1)\right)\;.$$
The fuzzy torus and its quantization as the clock and shift algebra play a central role in the stabilizer formalism of quantum computation [102].

#### 4.2.4. The Non-Commutative Sphere

The final example (and one that will be canonical for the remainder of this review) of a compact non-commutative manifold with finite dimensional representation of its algebra is the non-commutative or *fuzzy sphere* [82], $S_{\nu }^{2}$, whose algebra is generated by three functions $x^{1}$, $x^{2}$, and $x^{3}$, that satisfy the following:(70)$$\left[x^{i},x^{j}\right]=i\nu \;\epsilon^{ijk}\;x^{k}\;,\;\sum_{i}{\left(x^{i}\right)}^{2}=R^{2}1\;,$$
for some real parameters, ν and *R*, and 1 is the unit element of the algebra of functions. The first relation of (Equation 70) may be recognized as the defining relations of the algebra su(2), while the second relation is a relation of the quadratic Casimir of this su(2). Indeed, when(71)$$R^{2}=\nu^{2}\frac{N^{2}-1}{4}\;,$$
for an integer *N*, then this algebra can be represented as an *N*-dimensional representation of su(2),(72)$$x^{i}\to X^{i}=\nu \;J^{i}\;.$$
which may be either reducible or irreducible. Notice that the expression for the fuzzy sphere radius (Equation 71) is representation-dependent. For this reason, because $R^{2}$ must be a constant in (Equation 70), if $X^{i}$ is a reducible representation each irreducible factor must be of the same dimension. This means that $x_{i}$ may be represented as follows:(73)$$x_{i}\to \nu J_{N;i}⊗1_{p},$$
where $J_{N:i}$ is the generator of the *N*-dimensional representation of su(2) and *p* is some positive integer. In all that follows, we suppress *N*. In the case that p=1, any N×N matrix may be expanded as a finite-degree polynomial of the generators $J_{i}$.

The fuzzy sphere geometries feature as the classical minima of the bosonic potentials of the BMN and mini-BMN models introduced in Section 3.2.3.

### 4.3. Matrix Quantum Hall

We can point to a real quantum system that manifests the features of non-commutative geometry, including its UV/IR mixing and invariance under volume preserving diffeomorphism in an intuitive way: the quantum Hall fluid. In particular, Susskind argued that the Chern–Simons theory on a non-commutative geometry describes a droplet of charged, incompressible fluid reproducing the physics of the quantum Hall effect [103]. This construction was later refined and extended in [104,105,106,107,108,109]. The original action considered by Susskind is the non-commutative Chern–Simons (NCCS) theory:(74)$$L_{\mathrm{NCCS}}=\frac{1}{4\pi \nu }\epsilon^{\mu \nu \rho }\left(A_{\mu }★\partial_{\nu }A_{\rho }-\frac{2}{3}A_{\mu }★A_{\nu }★A_{\rho }\right).$$
★ is the standard Moyal product reviewed in Section 4. We will realize (Equation 74) as an MQM with the Moyal product replaced by matrix multiplication. We will focus on reviewing the most modern incarnation of the theory due to Tong and Turner in [108].

The system we consider is that of a single complex matrix, *Z*, and a complex vector ϕ. *Z* may be decomposed into two Hermitian matrices *X* and *Y* as Z=X+iY. The Lagrangian is as follows: (75)$$\begin{array}{cc} L & =\mathrm{Tr}[i\;Z^{†}D_{t}Z+i\;\left(D_{t}\phi \right)\phi^{†}-k\;A_{0}-Z^{†}Z]\\ & =\mathrm{Tr}[YD_{t}X-XD_{t}Y+i\;\left(D_{t}\phi \right)\phi^{†}-k\;A_{0}-\left(X^{2}+Y^{2}\right)]. \end{array}$$
where(76)$$D_{t}Z=\partial_{t}Z-i\;\left[A_{0},Z\right]\;,\;D_{t}\phi =\partial_{t}\phi -i\;A_{0}\phi \;.$$
The second line of (Equation 75) makes clear that *X* and *Y* are conjugate momenta, exactly as we would expect from particles in an infinitely strong magnetic field. The field $A_{0}$ appears only linearly in the Lagrangian and so its equations of motion function as a gauge constraint:(77)$$\left[Z,Z^{†}\right]+\phi \phi^{†}=k1.$$
This is precisely the algebra of the fuzzy disc described in the previous section above. The trace of this constraint implies(78)$$\phi^{†}\phi =kN.$$
The classical ground state solution is given by the *N*-dimensional representation we constructed in the previous section:(79)$$Z_{\mathrm{cl}}=\sqrt{k}\sum_{j=0}^{N-2}\sqrt{j}|j)(j+1|,\;\phi_{\mathrm{cl}}=\sqrt{N}\left|N\right).$$
The eigenvalues of $X_{\mathrm{cl}}=\frac{1}{2}\left(Z_{\mathrm{cl}}+Z_{\mathrm{cl}}^{†}\right)$, plotted as a histogram, form a semicircle (see Figure 11).

The quantum commutation relations of the matrix entries are as follows:(80)$$〚Z_{ab},Z_{cd}〛=〚Z_{ab}^{†},Z_{cd}^{†}〛=0\;,\;〚Z_{ab},Z_{cd}^{†}〛=\delta_{ac}\delta_{bd}\;,\;〚\phi_{a},\phi_{b}^{†}〛=\delta_{ab}\;.$$
The Hamiltonian is simply as follows: (81)$$H=\mathrm{Tr}[Z^{†}Z]=\sum_{ab}Z_{ab}^{†}Z_{ab}\;,$$
which is a harmonic trap biasing the interior of the droplet. Given the commutation relations (Equation 80), we may solve (Equation 81) by recognizing it as a sum of $N^{2}$ harmonic oscillators. After quantizing, we can treat the algebra (Equation 77) as an operator constraint which we must normal order as quantum operators:(82)$$G_{ab}=:{\left[Z,Z^{†}\right]}_{ab}:+\phi_{b}^{†}\phi_{a}-\left(k-1\right)\delta_{ab}$$
which must annihilate physical states. This normal ordering, with all creation operators implicitly moved to the left, is responsible for the shift k→k−1 in the Gauss constraint (see the discussion in [109]).

The Fock ground state, |0〉, which is annihilated by $Z_{ab}$ and $\phi_{a}$, can be expressed as a function of *X* as follows:(83)$$\psi_{0}\left(X\right)=e^{-\frac{1}{2}\mathrm{Tr}[X^{2}]},$$
and the resulting probability density is, again, a Gaussian one. This ground state eigenvalue distribution localizes to the Wigner semicircle, exactly in line (to leading order in *N*) with the expectation from the classical ground state. Quantum mechanically, in order to satisfy the Gauss constraint, this state is dressed. This wavefunction is known explicitly and given by the following [104,106,108]:(84)$$|\Psi_{k}\rangle =N_{k}{\left[\epsilon^{i_{1}\ldots i_{N}}\phi_{i_{1}}^{†}{\left(\phi^{†}Z^{†}\right)}_{i_{2}}\ldots {\left(\phi^{†}Z^{†(N-1)}\right)}_{i_{N}}\right]}^{k-1}|0\rangle ,$$
where $N_{k}$ is a normalization. (It is interesting to note that the ground state wavefunction (Equation 84) is intimately related to matrix integrals. If we diagonalize *X* into eigenvalues $\lambda_{n}$ the wavefunction becomes of the Calogero form [106](85)$$\Psi \left(\left\{\lambda_{n}\right\}\right)=N\prod_{m<n}{\left(\lambda_{m}-\lambda_{n}\right)}^{k}e^{-\sum_{n}\frac{1}{2}\lambda_{n}^{2}}.$$
The corresponding probability distribution may be recognized as a β-ensemble [110,111], corresponding to a U(N) invariant matrix integral when k=1 and a symplectic matrix integral for Sp(N) matrix integral for k=2.)

There are two ways to study low-energy fluctuations around the ground state. The first is to use the collective field formalism of the c=1 model in Section 3.1, which will result in a 1 + 1 dimensional theory of eigenvalue density fluctuations. The second way is by expanding the following:(86)$$X=X_{\mathrm{cl}}+A_{x},\;Y=Y_{\mathrm{cl}}+A_{y},$$
Resulting in the Lagrangian as follows: (87)$$L=\mathrm{Tr}\left[A_{y}D_{t}A_{x}-A_{x}D_{t}A_{y}-A_{x}^{2}-A_{y}^{2}\right].$$
$A_{x}$ and $A_{y}$ are N×N matrices, and so may be interpreted as functions on the fuzzy disc when expanded in powers of $Z_{\mathrm{cl}}$ and $Z_{\mathrm{cl}}^{†}$:(88)$$A_{i}=\sum_{mn=0}^{N}c_{i,mn}{\left(Z_{\mathrm{cl}}\right)}^{m}{\left(Z_{\mathrm{cl}}^{†}\right)}^{n}\;↔\;A_{i}\left(z,\bar{z}\right)=\sum_{mn}c_{i,mn}z^{m}{\bar{z}}^{n}.$$
The action (Equation 87), therefore, naturally carries an interpretation as a Chern–Simons-like action on the fuzzy disc, (Equation 74). We will study the entanglement entropy of this system in Section 6.4.

## 5. Subsystems and Entanglement

Entanglement is a measure of how quantum information is distributed across separate *subsystems* in a quantum theory. What we define as a ‘subsystem’ is a matter of choice and depends on the type of question one wishes to ask of a physical theory. A subsystem could consist of a particular collection of particles, a collection of sites in a lattice system, or a more abstract division of internal degrees of freedom. However, in a local quantum field theory we are typically interested in how information is spread across differing spatial regions. As we will soon see, the declaration of a ‘subsystem’ is more interesting in matrix quantum mechanics for the reason that space itself emerges from internal degrees of freedom. Precisely defining a partition of matrix degrees of freedom that results in a spatial subsystem is one goal of this section.

Before doing so, we will more concretely define what is meant when we use the word ‘subsystem.’ A subsystem, Σ, in any quantum theory is defined through a factorization of its Hilbert space, H. At its simplest level such a factorization takes the following form:(89)$$H=H_{\Sigma }⊗H_{\bar{\Sigma }}\;,$$
where the factor $H_{\Sigma }$ can be identified with the degrees of freedom of the subsystem, Σ, and $H_{\bar{\Sigma }}$ with its complement, $\bar{\Sigma }$. While such a factorization cleanly separates the degrees of freedom of Σ from its complement, $\bar{\Sigma }$, this does not imply every states is a product of a state in Σ and a state in $\bar{\Sigma }$. A generic state, |ψ〉∈H, is entangled and shares information across Σ and $\bar{\Sigma }$ in the form of a superposition of such products. One measure for this entanglement is the *entanglement entropy* defined in the following way.

From |ψ〉 we form its density matrix, ρ=|ψ〉〈ψ| and reduce it to subsystem Σ by “tracing out” $H_{\bar{\Sigma }}$. There is no subtlety in this precisely because H is a tensor product, (Equation 89). This defines the reduced density matrix $\rho_{\Sigma }$: (90)$$\rho_{\Sigma }={\mathrm{tr}}_{H_{\bar{\Sigma }}}\rho \;.$$
While ρ is a projector onto a pure state, when |ψ〉 is entangled $\rho_{\Sigma }$ will be a probabilistic mixture indicating a loss of information in forgetting $\bar{\Sigma }$. The von Neumann entropy of $\rho_{\Sigma }$ provides a measure of this information loss which is the entanglement entropy: (91)$$S_{\Sigma }=-{\mathrm{tr}}_{H_{\Sigma }}\left(\rho_{\Sigma }\log\rho_{\Sigma }\right)\;.$$

When Σ is a spatial subregion of a local quantum field, the above basic steps are rife with subtleties and obstructions. At a basic level, this is because the vacuum of a local quantum field theory is entangled at all length scales about the boundary of Σ (the *entangling surface*), which leads to a UV divergence in $S_{\Sigma }$. In many cases, the tensor factorization (Equation 89) can be defined in a cutoff sense leading to regulated entanglement entropy. A more modern perspective is to define entanglement entropy with respect to an algebra of observables. Given an algebra of operators A acting on H, we assign a subalgebra $A_{\Sigma }\subset A$ as the operators with support within Σ. The reduced density matrix can be assigned as the state in $A_{\Sigma }$ reproducing the expectation values: (92)$$\mathrm{tr}\left(\rho_{\Sigma }\;O_{\Sigma }\right)=\mathrm{tr}\left(\rho \;O_{\Sigma }\right)\;,\;\forall \;O_{\Sigma }\in A_{\Sigma }\;,$$
and its entanglement entropy can be computed through (Equation 91). According to the classification of von Neumann algebras, the subalgebra associated to a spatial region of a local quantum field theory is type-III and does not possess a trace [112,113,114,115]; this is a restatement of the above divergence. Computing (Equation 91) still requires regulating. A comprehensive study of the evolving study of entanglement entropy in quantum field theory is not within the scope of this review article. (See [116] for such an overview (especially regarding the modern algebraic approach).) Our focus is instead on how it is implemented specifically in matrix quantum mechanics and so we will introduce concepts as needed.

### 5.1. Entanglement and Gauge Invariance

Even barring the short distance obstructions to a Hilbert space factorization, such as (Equation 89), there are further obstructions in quantum gauge theories. The constraints of eliminating gauge redundancy is in tension with a local decomposition of the Hilbert space. (In a moral sense, gauge redundancy arises from writing a constrained system as a local field theory.) A simple example is the Gauss law constraint, which correlates electric charge contained within any region Σ with its complement. Thus, the more generic structure associated to a system with gauge invariance is of the following form:(93)$$H=\underset{q}{⨁}H_{\Sigma ,q}⊗H_{\bar{\Sigma },q}\;.$$
The index *q* should be thought of as running over superselection sectors for the state of the subsystem Σ. In our Gauss law example, *q* denotes the total charge contained in Σ.

#### 5.1.1. The Extended Hilbert Space

The lack of factorization of the Hilbert space of physical states means we need to provide a prescription for defining reduced density matrices and for computing entanglement entropies. There is no canonical prescription, and, in principle, the entanglement entropy that we compute will depend on our prescription. In essence, what we want is to embed the Hilbert space of physical states into a larger *extended Hilbert space* admitting a tensor factorization:(94)$$H\overset{I}{↪}H_{\mathrm{ext}}=H_{\mathrm{ext},\Sigma }⊗H_{\mathrm{ext},\bar{\Sigma }}\;.$$
Precisely because gauge invariance precludes the factorization (Equation 89), the extended Hilbert space contains non-physical, gauge-variant, states, and the physical Hilbert space, H, is embedded only as a subspace. The embedding, I, is what is known as a *factorization map*. Given the structure of (Equation 93) one natural embedding is given by the following:(95)$$H_{\mathrm{ext}}=\left(\underset{q}{⨁}H_{\Sigma ,q}\right)⊗\left(\underset{q}{⨁}H_{\bar{\Sigma },q}\right)\;,$$
although there can exist multiple inequivalent factorization maps. Under the image of I, any physical state is represented on a factorized Hilbert space; from there a reduced density matrix and entanglement entropy can be computed per the procedure outlined in Equations (Equation 90) and (Equation 91). In doing so it is important to appreciate that the resulting entanglement entropy is sensitive to our choice of factorization map. The factorization (Equation 95) is natural from the point of view of the structure (Equation 93): given a basis $|m_{q},{\bar{m}}_{q}\rangle$ of $H_{\Sigma ,q}⊗H_{\bar{\Sigma },q}$, a state(96)$$|\psi \rangle =\sum_{q}\sum_{m_{q},{\bar{m}}_{q}}\psi_{m_{q},{\bar{m}}_{q}}^{\left(q\right)}|m_{q},{\bar{m}}_{q}\rangle \equiv \sum_{q}|\psi^{\left(q\right)}\rangle \;,$$
embeds trivially to a state $|\tilde{\psi }\rangle \in H_{\mathrm{ext}}$(97)$$|\tilde{\psi }\rangle =\sum_{q,q^{\prime }}\sum_{m_{q},{\bar{m}}_{q^{\prime }}}{\tilde{\psi }}_{m_{q},{\bar{m}}_{q^{\prime }}}^{\left(q,q^{\prime }\right)}|m_{q},m_{q^{\prime }}\rangle \;,\;{\tilde{\psi }}_{m_{q},{\bar{m}}_{q^{\prime }}}^{\left(q,q^{\prime }\right)}=\delta^{qq^{\prime }}\;\psi_{m_{q},{\bar{m}}_{q}}^{\left(q\right)}\;.$$
Within this embedding the reduced density matrix takes the following form: (98)$$\begin{array}{c} {\tilde{\rho }}_{\Sigma }={\mathrm{tr}}_{H_{\mathrm{ext},\bar{\Sigma }}}|\tilde{\psi }\rangle \langle \tilde{\psi }|=\sum_{q}\sum_{{\bar{m}}_{q}}{\left(\psi_{n_{q},{\bar{m}}_{q}}^{\left(q\right)}\right)}^{*}\psi_{m_{q},{\bar{m}}_{q}}^{\left(q\right)}|m_{q}\rangle \langle n_{q}|=\sum_{q}{\mathrm{tr}}_{H_{\bar{\Sigma },q}}|\psi^{\left(q\right)}\rangle \langle \psi^{\left(q\right)}|\;. \end{array}$$
That is the density matrix reduced in the extended Hilbert space is equivalent to an orthogonal sum over the physical state reduced on each sector of (Equation 93). Because the individual $|\psi^{\left(q\right)}\rangle$’s are not normalized to unity (indeed it is the sum over their normalizations that must be unity) we can write the following:(99)$$|\psi^{\left(q\right)}\rangle \equiv \sqrt{p_{q}}|{\hat{\psi }}^{\left(q\right)}\rangle \;,\;\langle {\hat{\psi }}^{\left(q\right)}|{\hat{\psi }}^{\left(q\right)}\rangle =1\;,\;\sum_{q}p_{q}=1\;.$$
That is $\left\{p_{q}\right\}$ define a classical probability distribution and the reduced density matrix is a classical mixture of density matrices: (100)$${\tilde{\rho }}_{\Sigma }=\sum_{q}p_{q}\;{\hat{\rho }}_{\Sigma ,q}\;,\;{\hat{\rho }}_{\Sigma ,q}\equiv_{H_{\bar{\Sigma },q}}|{\hat{\psi }}^{\left(q\right)}\rangle \langle \psi^{\left(q\right)}|\;.$$
The resulting entanglement entropy contains a contribution of a classical Shannon entropy of the probability distribution plus a weighted sum over entanglement entropies in each selection sector: (101)$$S_{\Sigma }=-\sum_{q}p_{q}\log p_{q}+\sum_{q}p_{q}\;S_{\Sigma }^{\left(q\right)}\;,\;S_{\Sigma }^{\left(q\right)}\equiv -{\mathrm{tr}}_{H_{\Sigma ,q}}\left({\hat{\rho }}_{\Sigma ,q}\log{\hat{\rho }}_{\Sigma ,q}\right)\;.$$

#### 5.1.2. Edge Modes and Area Laws

Within the extended Hilbert space, what were previously gauge ‘symmetries’ now act as global symmetries. More specifically, let the gauge group be G, and call the generator of a local gauge transformation, $\hat{G}\left(x\right)$. This generator does not necessarily annihilate all states in $H_{\mathrm{ext}}$; it is only required to annihilate the physical states within I(H). The states within $H_{\mathrm{ext}}$ that carry the action of $\hat{G}\left(x\right)$ correspond to new degrees of freedom. Because gauge symmetries act locally, it is easy to argue that the states with non-trivial G action are localized to the boundary between Σ and $\bar{\Sigma }$: gauge transformations acting strictly within the interior of either of Σ or $\bar{\Sigma }$ do not spoil the factorization of H. It is instead the gauge transformations localized directly at the entangling surface, ∂Σ, that act simultaneously on $H_{\mathrm{ext},\Sigma }$ and $H_{\mathrm{ext},\bar{\Sigma }}$. Thus, $H_{\mathrm{ext}}$ will contain degrees of freedom that are localized to ∂Σ and carry a representation of the gauge group restricted to ∂Σ. These are *edge modes*.

Each factor, $H_{\mathrm{ext},\Sigma }$ and $H_{\mathrm{ext},\bar{\Sigma }}$, will contain edge mode degrees of freedom. Physical states in I(H) will precisely correlate the edge modes of Σ with those of $\bar{\Sigma }$ such that the global state is invariant under gauge transformations. Thus, there is a significant amount of entanglement across ∂Σ in physical states carried by edge modes. To be more specific, the generator of a gauge transformation acting on $H_{\mathrm{ext}}$ will take the following form:(102)$${\hat{G}}_{i}={\hat{G}}_{\Sigma ,i}⊗{\hat{1}}_{\bar{\Sigma },i}+{\hat{1}}_{\Sigma ,i}⊗{\hat{G}}_{\bar{\Sigma },i}\;.$$
The index *i* labels the differing gauge generators that ${\tilde{\rho }}_{\Sigma }$ must be a singlet under. In a lattice gauge theory this could be, for instance, all the links that cross the entangling surface, ∂Σ. Both ${\hat{G}}_{\Sigma ,i}$ and ${\hat{G}}_{\bar{\Sigma },i}$ generate G actions at ∂Σ and physical states are invariant under the diagonal of these two actions. In particular, the image of global state in $H_{\mathrm{ext}}$, $\tilde{\rho }=I|\psi \rangle \langle \psi |I^{†}$, satisfies the following:(103)$$〚{\hat{G}}_{i},\tilde{\rho }〛=0\;,$$
which then implies that the reduced states are singlets as well:(104)$$〚{\hat{G}}_{\Sigma ,i},{\tilde{\rho }}_{\Sigma }〛=0\;.$$
This implies that ${\tilde{\rho }}_{\Sigma }$ breaks up into a sum over representation identities of G:(105)$${\tilde{\rho }}_{\Sigma }=\underset{\mu_{i}}{⨁}p_{\mu_{i}}\frac{{\hat{1}}_{\mu_{i}}}{d_{\mu_{i}}}⊗{\hat{\rho }}_{\mu_{i}}\;,$$
where we index representations by $\mu_{i}$ and their representation dimensions by $d_{\mu_{i}}$. Lastly, $p_{\mu_{i}}$ are set of probability measures such that $\sum_{\mu_{i}}p_{\mu_{i}}=1$, and ${\hat{\rho }}_{\mu_{i}}$ is the portion of the state in the $\mu_{i}$ block that is a singlet under the gauge generators contained strictly in Σ. Note that this structure of reduced density matrix is commensurate with the generic structure of the physical Hilbert space given by (Equation 93).

We see that gauge invariance enforces that within the extended Hilbert space, the state is *maximally entangled* amongst the edge modes at ∂Σ in a given representation. The entanglement entropy of the state takes the generic form: (106)$$S_{\Sigma }=-\sum_{\mu_{i}}p_{\mu_{i}}\log p_{\mu_{i}}+\sum_{\mu_{i}}p_{\mu_{i}}\log d_{\mu_{i}}-\sum_{\mu_{i}}p_{\mu_{i}}{\mathrm{tr}}_{H_{\Sigma ,\mu_{i}}}\left({\hat{\rho }}_{\mu_{i}}\log{\hat{\rho }}_{\mu_{i}}\right)\;.$$
The first term is often called the *Shannon term*, as it takes the form of the Shannon entropy of a classical probability distribution. The last term is a weighted sum over ‘bulk’ entanglement arising from gauge singlet degrees of freedom within the interiors of Σ and $\bar{\Sigma }$. This middle term arises from entanglement at the edge ∂Σ and because gauge transformations are local, it is extensive in the area of ∂Σ (i.e., recall in a lattice gauge theory the sum *i* would range over all links crossing the entangling surface). Thus, this term is responsible for an edge mode entanglement entropy that scales with |∂Σ|, the area of ∂Σ:(107)$$S_{\Sigma ,\mathrm{edge}}\equiv -\sum_{\mu_{i}}p_{\mu_{i}}\log p_{\mu_{i}}+\sum_{\mu_{i}}p_{\mu_{i}}\log d_{\mu_{i}}∼\frac{\left|\partial \Sigma \right|}{\epsilon^{d-2}}\;,$$
where ϵ is a dimensionful short distance cutoff (such as a lattice spacing). This behavior is what we mean when say ‘*area law*’ entanglement.

#### 5.1.3. Gauge Invariant Subalgebras

The above description is, in words, indicating that there is not a (uniquely) determined notion of localizing degrees of physical degrees of freedom with a subregion, Σ, or its complement, $\bar{\Sigma }$. This has an operational meaning in terms of the subalgebras, $A_{\Sigma }$, acting on physical states, which we briefly review now.

Given an assignment $\Sigma \to A_{\Sigma }$ of a subalgebra of gauge-invariant operators to a subregion Σ, we then associate to $\bar{\Sigma }$, the commutant:(108)$$A_{\bar{\Sigma }}:={\left(A_{\Sigma }\right)}^{c}=\left\{O\in A\;|\;〚O,O_{\Sigma }〛=0\;,\;\forall \;O_{\Sigma }\in A_{\Sigma }\right\}$$
This definition is in keeping with locality in a quantum field theory: operators at spacelike distances commute. The tension of gauge-invariance and the ability to locally isolate degrees of freedom is then manifested that these two subalgebras have overlap. That is they possess an intersection called *the center* which, by definition, consists of operators that commute with $A_{\Sigma }$ and $A_{\bar{\Sigma }}$:(109)$$Z_{\Sigma }=A_{\Sigma }\cap A_{\bar{\Sigma }}\;,\;〚Z_{\Sigma },A_{\Sigma }〛=〚Z_{\Sigma },A_{\bar{\Sigma }}〛=0\;.$$
Operators in $Z_{\Sigma }$ can be simultaneously diagonalized and take the form of projectors on their mutual eigenbasis:(110)$$Z_{\Sigma }={\mathrm{span}}_{C}\left\{\Pi_{q}\right\}\;,\;\sum_{q}\Pi_{q}=1\;.$$
These are precisely the projectors that isolate a given block of (Equation 93). For a given state, ρ=|ψ〉〈ψ| with an associated reduced density matrix $\rho_{\Sigma }\in A_{\Sigma }$ satisfying (Equation 92), we can work in the eigenbasis of $Z_{\Sigma }$ to write the following:(111)$$\rho_{\Sigma }=\underset{q}{⨁}\;p_{\Sigma ,q}\;{\hat{\rho }}_{\Sigma ,q}\;,$$
We have separated off coefficients such that each sub-block of $\rho_{\Sigma }$ is a normalized and positive density matrix:(112)$$\mathrm{tr}{\hat{\rho }}_{\Sigma ,q}=1\;.$$
Normalization and positivity of the full density matrix then implies that $\left\{p_{\Sigma ,q}\right\}$ define a probability distribution. This is, in fact, the same distribution as defined in (Equation 99). Subsequently, the von Neumann entropy of $\rho_{\Sigma }$ displays the, now familiar, Shannon contribution as well as weighted sum over entanglement entropies: (113)$$S_{\Sigma }=-\sum_{q}p_{q}\log p_{q}-\sum_{q}\mathrm{tr}\left(\rho_{\Sigma ,q}\log\rho_{\Sigma ,q}\right)\;.$$

## 6. Target Space Entanglement

The above discussion has focussed on the subsystems given by local partitions of the base space that a physical system is defined on. However, for models of matrix quantum mechanics, there is no base space to partition; indeed, as we have emphasized above, a intriguing consideration of these models is that spacetime is not the arena in which the dynamics takes place, but instead emerges from the dynamics of the model itself. In the language of string theory and sigma models, the spacetime is part of the *target space* of MQM. Indeed, the base space of MQM is a point and so there is no meaningful notion of ‘spatial subsystem’ with respect to which we can partition a state. Thus, in investigating how the connection between entanglement and locality emerges from MQM, we will need to investigate target space entanglement. Some of what we have established in the previous section also applies here: the target space itself will typically have gauge invariances that will prevent strict factorization of the physical Hilbert space. There will also be new subtleties that we will have to consider.

In a not completely unrelated fashion to the above, the notion of entanglement in target space also plays a role in string theory where the target space of the strings is precisely the real space of the low-energy effective supergravity. There is a growing body of literature developing and investigating target space entanglement in string theory [33,117,118,119,120,121,122,123,124]. In this review, we will focus solely on the aspects most applicable to MQM and unfortunately omit a review of this literature; however, we encourage the interested reader to see the papers cited above. We will build intuition for the problem with single and multiple particles propagating in a $R^{D}$ target space before moving to matrices.

### 6.1. Single Particle

The simplest scenario to begin our discussion of target space entanglement is the quantum mechanics of a single particle. Configurations of the system are given by the position of the particle at a given point in time, $\vec{x}\left(t\right)$, which are the coordinates of a target space, which for simplicity we will take to be $R^{D}$. The quantum Hilbert space is as follows:(114)$$H_{1}=L^{2}\left(R^{D}\right)\;.$$

Indeed, consider the single-particle Hilbert space spanned by $L^{2}\left(R^{D}\right)$ wavefunctions, |ψ〉. We can also use the delta function normalizable basis $|\vec{x}\rangle$. Given a subregion of $\Sigma \subset R^{D}$, this basis makes clear that is the one-particle space is actually a tensor sum:(115)$$H_{1}=H_{\Sigma ,1}⊕H_{\bar{\Sigma },1}\;,\;H_{\Sigma ,1}:={\mathrm{span}}_{C}\left\{|\vec{x}\rangle \;|\;\vec{x}\in \Sigma \right\}$$
and a similar definition for $H_{1,\bar{\Sigma }}$. While perhaps not immediately obvious, (Equation 115) actually fits into the general Hilbert space structure we defined above, (Equation 93). Indeed, defining $H_{\Sigma ,0}=H_{\bar{\Sigma },0}:=C$, then $H_{1}$ can be written as follows:(116)$$H_{1}=\underset{q=0,1}{⨁}\left(H_{\Sigma ,q}⊗H_{\bar{\Sigma },1-q}\right)\;.$$
We can view $H_{\Sigma /\bar{\Sigma },q}$ as the Hilbert space of *q* particles with $\Sigma /\bar{\Sigma }$ as their target. Here, the lack of tensor factorization stems entirely from the global constraint of particle number conservation—if the particle is localized in Σ it is not localized in $\bar{\Sigma }$.

It will be useful notation for later to define a target space subregion theta function, $\theta_{\Sigma }$, associated to Σ as follows:(117)$$\theta_{\Sigma }\left(\vec{x}\right):=\theta \left(f_{\Sigma }\left(\vec{x}\right)-c\right)\;,\;\theta_{\bar{\Sigma }}\left(\vec{x}\right):=1-\theta_{\Sigma }\left(\vec{x}\right)$$
where θ is the standard Heaviside theta function, and $f_{\Sigma }:R^{D}\to R$ is a function whose level set defines ∂Σ, i.e., $\partial \Sigma =\{f_{\Sigma }\left(\vec{x}\right)=c\}$ and $f_{\Sigma }\left(\vec{x}\right)>c$ for $\vec{x}\in \Sigma$ and $f_{\Sigma }\left(\vec{x}\right)<c$ for $\vec{x}\in \bar{\Sigma }$.

In the language of subalgebras, this decomposition of the Hilbert space has a natural action under the subalgebras:(118)$$A_{\Sigma }=U\left\{|\vec{x}\rangle \langle {\vec{x}}^{\prime }|\;|\;\vec{x},{\vec{x}}^{\prime }\in \Sigma \right\}\;,\;A_{\bar{\Sigma }}=U\left\{|\vec{x}\rangle \langle {\vec{x}}^{\prime }|\;|\;\vec{x},{\vec{x}}^{\prime }\in \bar{\Sigma }\right\}\;.$$
where U denotes the universal enveloping algebra and it should be tacitly understood that the identity on $H_{1}$ is included in the generation of this algebra. Associated with these subalgebras are projectors: (119)$$\Pi_{\Sigma }=\int_{\Sigma }d^{D}\vec{x}\;|\vec{x}\rangle \langle \vec{x}|=\int d^{D}\vec{x}\;\theta_{\Sigma }\left(\vec{x}\right)|\vec{x}\rangle \langle \vec{x}|\;,\;\Pi_{\bar{\Sigma }}=\int_{\bar{\Sigma }}d^{D}\vec{x}\;|\vec{x}\rangle \langle \vec{x}|=\int d^{D}\vec{x}\;\theta_{\bar{\Sigma }}\left(\vec{x}\right)|\vec{x}\rangle \langle \vec{x}|\;,$$
which are the natural promotion of (Equation 117) to Hilbert space operators and from which we can project any operator, $\hat{O}$, acting on $H_{1}$ to the corresponding subalgebra through ${\hat{O}}_{A/\bar{\Sigma }}=\Pi_{A/\bar{\Sigma }}\;\hat{O}\;\Pi_{A/\bar{\Sigma }}$. These subalgebras contain a common center:(120)$$Z_{\Sigma }=A_{\Sigma }\cap A_{\bar{\Sigma }}={\mathrm{span}}_{C}\left\{\Pi_{\Sigma },\Pi_{\bar{\Sigma }}\right\}\;,$$
which is a restatement of the lack of factorization of $H_{1}$.

From our previous section, we can write down a natural candidate extended Hilbert space for the one-particle system, (Equation 116), as follows:(121)$$H_{1,\mathrm{ext}}:=\left(\underset{q=0,1}{⨁}H_{\Sigma ,q}\right)⊗\left(\underset{q=0,1}{⨁}H_{\bar{\Sigma },q}\right)\;.$$
In this Hilbert space we have relaxed the constraint of particle number conservation: in $H_{1,\mathrm{ext}}$ we can have either one particle in Σ, one in $\bar{\Sigma }$, one in both, or none at all.

Let us now describe the factorization map of physical states from $H_{1}$ into $H_{1,\mathrm{ext}}$. A generic state of $H_{1}$ can be written as follows: (122)$$\begin{array}{cc} |\psi \rangle =\int d^{D}\vec{x}\;\psi \left(\vec{x}\right)|\vec{x}\rangle & =\int d^{D}\vec{x}\;\theta_{\Sigma }\left(\vec{x}\right)\;\psi \left(\vec{x}\right)|\vec{x}\rangle +\int d^{D}\vec{x}\;\theta_{\bar{\Sigma }}\left(\vec{x}\right)\;\psi \left(\vec{x}\right)|\vec{x}\rangle \\ \; & :=|\psi_{\Sigma }\rangle +|\psi_{\bar{\Sigma }}\rangle \;. \end{array}$$
The probability to find the particle in Σ or $\bar{\Sigma }$ is given by the following: (123)$$p_{\Sigma ,1}=\int d^{D}\vec{x}\;\theta_{\Sigma }\left(\vec{x}\right)\;|\psi (\vec{x}){|}^{2}\;,\;p_{\bar{\Sigma },1}=\int d^{D}\vec{x}\;\theta_{\bar{\Sigma }}\left(\vec{x}\right)\;{\left|\psi (\vec{x})\right|}^{2}\;,$$
which are equivalent, by number conservation, to finding no particles in $\bar{\Sigma }$ or Σ, respectively:(124)$$p_{\bar{\Sigma },0}=p_{\Sigma ,1}=1-p_{\bar{\Sigma },1}\;,\;p_{\Sigma ,0}=p_{\bar{\Sigma },1}=1-p_{\Sigma ,1}\;.$$
We can then designate the image of |ψ〉 in $H_{1,\mathrm{ext}}$ as follows:(125)$$\tilde{|\psi \rangle }=\left(\sqrt{p_{\Sigma ,0}}⊕|\psi_{\Sigma }\rangle \right)⊗\left(\sqrt{p_{\bar{\Sigma },0}}⊕|\psi_{\bar{\Sigma }}\rangle \right)\;.$$
It is easy to verify that this is an isometry:(126)$$\tilde{\langle \psi }|\tilde{\psi \rangle }=\left(p_{\Sigma ,0}+p_{\Sigma ,1}\right)\left(p_{\bar{\Sigma },0}+p_{\bar{\Sigma },1}\right)=1\;.$$
An interesting feature of this factorization map is that, strictly speaking, (Equation 125) is a product state on the state between $H_{\Sigma }$ and $H_{\bar{\Sigma }}$. However the entanglement entropy of |ψ〉 is still non-zero: in the form we illustrated above, (Equation 106), this entanglement arises as the classical, Shannon term induced by the particle number conservation. Let us show this now. It is easy to work out that the reduced density matrix is as follows:(127)$${\tilde{\rho }}_{\Sigma }=p_{\Sigma ,0}⊕|\psi_{\Sigma }\rangle \langle \psi_{\Sigma }|\;.$$
Although $|\psi_{\Sigma }\rangle \langle \psi_{\Sigma }|$ looks to be a rank-one projector, it is not quite since $|\psi_{\Sigma }\rangle$ is not normalized—its norm is $\langle \psi_{\Sigma }|\psi_{\Sigma }\rangle =p_{\Sigma ,1}$. Thus, the von Neumann entropy of ${\tilde{\rho }}_{\Sigma }$ is as follows:(128)$$S_{\Sigma }=-p_{\Sigma ,0}\log p_{\Sigma ,0}-p_{\Sigma ,1}\log p_{\Sigma ,1}\;.$$

We pause to note that we could have arrived at (Equation 127) algebraically from the projectors spanning the center, (Equation 120): notice that $\Pi_{\bar{\Sigma }}$ and $\Pi_{\Sigma }$ are the projectors onto the $H_{\Sigma ,0}⊗H_{\bar{\Sigma },1}$ and $H_{\Sigma ,1}⊗H_{\bar{\Sigma },0}$ blocks of (Equation 116), respectively. The full density matrix, ρ=|ψ〉〈ψ|, then takes the following form:(129)$$\rho =\left(\Pi_{\bar{\Sigma }}\rho \Pi_{\bar{\Sigma }}\right)⊕\left(\Pi_{\Sigma }\rho \Pi_{\Sigma }\right)=\left(|\psi_{\bar{\Sigma }}\rangle \langle \psi_{\bar{\Sigma }}|\right)⊕\left(|\psi_{\Sigma }\rangle \langle \psi_{\Sigma }|\right)$$
whose reduction on each $H_{\bar{\Sigma },q}$ on each block yields (Equation 127).

### 6.2. N Identical Particles

We now increase the complexity by considering the quantum mechanics of multiple identical particles. Much of this section will follow [125,126]. We designate our identical particles by positions ${\left\{{\vec{x}}_{a}\right\}}_{a=1,\ldots ,N}$. The quantum Hilbert space is given by the following:(130)$$H_{N}=\left(\overset{N}{\underset{a=1}{⨂}}H_{1}^{\left(a\right)}\right)/S_{N}\;,$$
where each $H_{1}^{\left(a\right)}$ is the one-particle Hilbert space from before and $S_{N}$ is the symmetric group of *N* elements. An element $\sigma \in S_{N}$ acts on the collection of single-particle wavefunctions symmetrically or anti-symmetrically depending on if the particles are bosons or fermions, respectively:(131)$$\begin{array}{cc} \sigma \;\circ \;\psi_{\mathrm{boson}}\left({\vec{x}}_{1},{\vec{x}}_{2},\ldots ,{\vec{x}}_{N}\right) & =\psi_{\mathrm{boson}}\left({\vec{x}}_{\sigma (1)},{\vec{x}}_{\sigma (2)},\ldots ,{\vec{x}}_{\sigma (N)}\right),\\ \sigma \;\circ \;\psi_{\mathrm{fermion}}\left({\vec{x}}_{1},{\vec{x}}_{2},\ldots ,{\vec{x}}_{N}\right) & =\mathrm{sgn}\left(\sigma \right)\psi_{\mathrm{fermion}}\left({\vec{x}}_{\sigma (1)},{\vec{x}}_{\sigma (2)},\ldots ,{\vec{x}}_{\sigma (N)}\right)\;. \end{array}$$
We will work with bosons in what follows, although fermions only require a simple modification.

We can view (Equation 130) as treating $S_{N}$ as a gauge redundancy: physical states of $H_{N}$ are wavefunctions of *N* variables, with the additional constraint that they are symmetric under permutations. This is an additional target space gauge redundancy in comparison to the single-particle space. We can identify $H_{N}$ as the image of *N*-fold tensor product under the projector, $P_{S_{N}}$(132)$$H_{N}=P_{S_{N}}\left(\overset{N}{\underset{a=1}{⨂}}H_{1}^{\left(a\right)}\right)\;.$$
For bosonic particles $P_{S_{N}}$ can be expressed in terms of basis states as(133)$$P_{S_{N}}=\frac{1}{N!}\sum_{\sigma \in S_{N}}\int d^{D}{\vec{x}}_{1}d^{D}{\vec{x}}_{2}\ldots d^{D}{\vec{x}}_{N}\;|{\vec{x}}_{\sigma (1)}{\vec{x}}_{\sigma (2)}\ldots {\vec{x}}_{\sigma (N)}\rangle \langle {\vec{x}}_{1},{\vec{x}}_{2},\ldots ,{\vec{x}}_{N}|\;.$$

We will be interested once again in partitioning the target space into a region, Σ, and its complement, $\bar{\Sigma }$. Our warmup for the one-particle system acts as a guide here: we first decompose each one-particle factor of (Equation 130) as follows:(134)$$H_{N}=\left(\overset{N}{\underset{a=1}{⨂}}\left(H_{\Sigma ,1}^{\left(a\right)}⊕H_{\bar{\Sigma },1}^{\left(a\right)}\right)\right)/S_{N}\;,$$
which we can further decompose into the following:(135)$$H_{N}=\overset{N}{\underset{a=0}{⨁}}\left(H_{\Sigma ,a}{⊗}_{S}H_{\bar{\Sigma },N-a}\right)\;,\;H_{\Sigma ,a}:={\left(H_{\Sigma ,1}\right)}^{⊗a}/S_{i}\;.$$
where ${⊗}_{S}$ is the symmetrized tensor product. This structure is slightly different than what was discussed in typical gauge theories, (Equation 116): it is an additional structure of gauge redundancy that arises in target space entanglements. We will see this structure to a richer extent when we consider MQM.

We can cast (Equation 135) in the form of (Equation 93) by embedding it into the following:(136)$$H_{N}\subset {\tilde{H}}_{N}=\overset{N}{\underset{a=0}{⨁}}\left(H_{\Sigma ,a}⊗H_{\bar{\Sigma },N-a}\right)\;,$$
with the symmetric tensor factor replaced with the standard one. We can think of ${\tilde{H}}_{N}$ as a collection of ‘partially gauge-fixed’ Hilbert spaces with $S_{N}$ broken to $S_{a}\times S_{N-a}$ in each block by putting the first *a* particles, $\left\{{\vec{x}}_{1},\ldots ,{\vec{x}}_{a}\right\}$, in Σ. $H_{N}$ can once again be isolated from ${\tilde{H}}_{N}$ by the action of the projector, (Equation 133). Going the other direction, states of the $S_{N}$ invariant space can be embedded into ${\tilde{H}}_{N}$ by imposing $S_{N}$ invariance at the level of the wavefunction. We will illustrate this shortly.

Using the projector (Equation 133), we can associate the following $S_{N}$-invariant subalgebra of operators, $A_{\Sigma }$, to Σ [125]:(137)$$A_{\Sigma }=U\left\{P_{S_{N}}\left(|\vec{x}\rangle \langle {\vec{x}}^{\prime }|⊗1⊗\ldots ⊗1\right)P_{S_{N}}\;|\;\vec{x},{\vec{x}}^{\prime }\in \Sigma \right\}$$
This algebra has a commutant(138)$$A_{\bar{\Sigma }}=U\left\{P_{S_{N}}\left(|\vec{x}\rangle \langle {\vec{x}}^{\prime }|⊗1⊗\ldots ⊗1\right)P_{S_{N}}\;|\;\vec{x},{\vec{x}}^{\prime }\in \bar{\Sigma }\right\}$$
and it is easy to verify that these two algebras share a common center, $Z_{\Sigma }$, spanned by projectors(139)$$Z_{\Sigma }={\mathrm{span}}_{C}\overset{N}{\underset{a=0}{⋃}}\left\{\Pi_{a,N-a}\right\}$$
where(140)$$\Pi_{a,N-a}:=\left(\begin{array}{c} N\\ a \end{array}\right)P_{S_{N}}\left(\underset{a}{\underset{︸}{\Pi_{\Sigma }⊗\ldots ⊗\Pi_{\Sigma }}}⊗\underset{N-a}{\underset{︸}{\Pi_{\bar{\Sigma }}⊗\ldots ⊗\Pi_{\bar{\Sigma }}}}\right)P_{S_{N}}\;.$$
with $\Pi_{\Sigma }$ and $\Pi_{\bar{\Sigma }}$ the familiar one-particle projectors, (Equation 119). This center leads to the decomposition of the Hilbert space as in (Equation 135). To the ‘partially gauge-fixed’ Hilbert space instead corresponds to set of projectors with the symmetrization dropped:(141)$${\tilde{\Pi }}_{a,N-a}:=\underset{a}{\underset{︸}{\Pi_{\Sigma }⊗\ldots ⊗\Pi_{\Sigma }}}⊗\underset{N-a}{\underset{︸}{\Pi_{\bar{\Sigma }}⊗\ldots ⊗\Pi_{\bar{\Sigma }}}}\;.$$

From (Equation 136), we can build a candidate extended Hilbert space *mutatis mutandis*, following the single-particle example:(142)$${\tilde{H}}_{N,\mathrm{ext}}:=\left(\overset{N}{\underset{a=0}{⨁}}H_{\Sigma ,a}\right)⊗\left(\overset{N}{\underset{a=0}{⨁}}H_{\bar{\Sigma },a}\right)\;,$$
which has relaxed the global particle number conservation as well as the extends the global $S_{N}$ redundancy to an $S_{N}\times S_{N}$ acting on both subsystems, Σ and $\bar{\Sigma }$. Physical states are then embedded by the satisfaction of particle number conservation as well as being singlets under the diagonal $S_{N}$ living in $S_{N}\times S_{N}$.

Let us illustrate this factorization map for two identical particles. A generic 2-particle bosonic state can be written as follows: (143)$$|\psi \rangle =\int d^{D}{\vec{x}}_{1}d^{D}{\vec{x}}_{2}\;\psi \left({\vec{x}}_{1},{\vec{x}}_{2}\right)\;{|{\vec{x}}_{1},{\vec{x}}_{2}\rangle }_{S}\;,$$
where(144)$$|{\vec{x}}_{1},{\vec{x}}_{2}\rangle_{S}=P_{S_{2}}|{\vec{x}}_{1},{\vec{x}}_{2}\rangle =\frac{1}{2}\left(|{\vec{x}}_{1},{\vec{x}}_{2}\rangle +|{\vec{x}}_{2},{\vec{x}}_{1}\rangle \right)\;.$$
It is easy to rewrite this state as follows: (145)$$\begin{array}{cc} |\psi \rangle = & \int d^{D}{\vec{x}}_{1}\;\theta_{\bar{\Sigma }}\left({\vec{x}}_{1}\right)\theta_{\bar{\Sigma }}\left({\vec{x}}_{2}\right)d^{D}{\vec{x}}_{2}\psi \left({\vec{x}}_{1,2}\right)\;{|{\vec{x}}_{1},{\vec{x}}_{2}\rangle }_{S}\\ \; & \;+2\int d^{D}{\vec{x}}_{1}d^{D}{\vec{x}}_{2}\;\theta_{\Sigma }\left({\vec{x}}_{1}\right)\theta_{\bar{\Sigma }}\left({\vec{x}}_{2}\right)\;\psi \left({\vec{x}}_{1,2}\right)\;{|{\vec{x}}_{1},{\vec{x}}_{2}\rangle }_{S}\\ \; & \;\;+\int d^{D}{\vec{x}}_{1}d^{D}{\vec{x}}_{2}\;\theta_{\Sigma }\left({\vec{x}}_{1}\right)\theta_{\Sigma }\left({\vec{x}}_{2}\right)\;\psi \left({\vec{x}}_{1,2}\right)\;{|{\vec{x}}_{1},{\vec{x}}_{2}\rangle }_{S}\;, \end{array}$$
where the three terms correspond to $\left(H_{\Sigma ,0}{⊗}_{S}H_{\bar{\Sigma },2}\right)⊕\left(H_{\Sigma ,1}{⊗}_{S}H_{\bar{\Sigma },1}\right)⊕\left(H_{\Sigma ,2}{⊗}_{S}H_{\Sigma ,0}\right)$, respectively. We can embed this into the partially gauged fixed Hilbert space, ${\tilde{H}}_{2}$, by noting that we can equivalently write (Equation 143) as follows: (146)$$|\psi \rangle =\int d^{D}{\vec{x}}_{1}d^{D}{\vec{x}}_{2}\;\psi_{S}\left({\vec{x}}_{1},{\vec{x}}_{2}\right)|{\vec{x}}_{1},{\vec{x}}_{2}\rangle$$
with(147)$$\psi_{S}\left({\vec{x}}_{1},{\vec{x}}_{2}\right):=\frac{1}{2}\left(\psi \left(x_{1},x_{2}\right)+\psi \left(x_{2},x_{1}\right)\right)\;,$$
and $|{\vec{x}}_{1},{\vec{x}}_{2}\rangle$ in the standard tensor product, $H_{1}⊗H_{1}$.

We now reduce ρ=|ψ〉〈ψ| within the extended Hilbert space of the following:(148)$${\tilde{H}}_{2,\mathrm{ext}}=\left(\overset{2}{\underset{a=0}{⨁}}H_{\Sigma ,a}\right)⊗\left(\overset{2}{\underset{a=0}{⨁}}H_{\bar{\Sigma },a}\right)\;.$$
From the general discussion of Section 5.1, the reduced density matrix takes the following form:(149)$${\tilde{\rho }}_{\Sigma }=p_{\Sigma ,0}⊕p_{\Sigma ,1}{\hat{\rho }}_{\Sigma ,1}⊕p_{\Sigma ,2}{\hat{\rho }}_{\Sigma ,2}\;,$$
with(150)$$\begin{array}{cc} p_{\Sigma ,0}= & \int_{\bar{\Sigma }}d^{D}{\vec{x}}_{1}\int_{\bar{\Sigma }}d^{D}{\vec{x}}_{2}\;{\left|\psi_{S}\left({\vec{x}}_{1},{\vec{x}}_{2}\right)\right|}^{2}\;,\\ p_{\Sigma ,1}= & 2\int_{\Sigma }d^{D}{\vec{x}}_{1}\int_{\bar{\Sigma }}d^{D}{\vec{x}}_{2}\;{\left|\psi_{S}\left({\vec{x}}_{1},{\vec{x}}_{2}\right)\right|}^{2}\;,\\ p_{\Sigma ,2}= & \int_{\Sigma }d^{D}{\vec{x}}_{1}\int_{\Sigma }d^{D}{\vec{x}}_{2}\;{\left|\psi_{S}\left({\vec{x}}_{1},{\vec{x}}_{2}\right)\right|}^{2}\;, \end{array}$$
and(151)$$\begin{array}{cc} {\hat{\rho }}_{\Sigma ,1}= & p_{\Sigma ,1}^{-1}\int_{\Sigma }d^{D}{\vec{x}}_{1}\int_{\bar{\Sigma }}d^{D}{\vec{x}}_{2}\int_{\Sigma }d^{D}{\vec{y}}_{1}\;\psi_{S}{\left({\vec{y}}_{1},{\vec{x}}_{2}\right)}^{*}\;\psi_{S}\left({\vec{x}}_{1},{\vec{x}}_{2}\right)|{\vec{x}}_{1}\rangle \langle {\vec{y}}_{1}|\;,\\ {\hat{\rho }}_{\Sigma ,2}= & p_{\Sigma ,2}^{-1}\int_{\Sigma }d^{D}{\vec{x}}_{1}\int_{\Sigma }d^{D}{\vec{x}}_{2}\int_{\Sigma }d^{D}{\vec{y}}_{1}\int_{\Sigma }d^{D}{\vec{y}}_{2}\;\psi_{S}{\left({\vec{y}}_{1},{\vec{y}}_{2}\right)}^{*}\psi_{S}\left({\vec{x}}_{1},{\vec{x}}_{2}\right)|{\vec{x}}_{1},{\vec{x}}_{2}\rangle \langle {\vec{y}}_{1},{\vec{y}}_{2}|\;. \end{array}$$
Notice that, while ${\hat{\rho }}_{\Sigma ,2}$ is a product state, ${\hat{\rho }}_{\Sigma ,1}$ could possibly be entangled due to the symmetrization of (Equation 147). This would, for instance, account for the entanglement of a Bell-pair shared between Σ and $\bar{\Sigma }$. This reflected in the entanglement entropy (which we can now easily calculate): (152)$$S_{\Sigma }=-\sum_{a=0}^{2}p_{\Sigma ,a}\log p_{\Sigma ,a}-p_{\Sigma ,1}{\mathrm{tr}}_{H_{\Sigma ,1}}\left({\hat{\rho }}_{\Sigma ,1}\log{\hat{\rho }}_{\Sigma ,1}\right)\;.$$
which, in addition to the standard Shannon terms we saw before, contains a term of genuine quantum entanglement of particles shared between Σ and $\bar{\Sigma }$.

The generalization of the factorization map of *N* identical particles into (Equation 142) is straightforward. Given a physical state(153)$$|\psi \rangle =\int \left(\prod_{a=1}^{N}d^{D}{\vec{x}}_{a}\right)\;\psi \left(\left\{{\vec{x}}_{i}\right\}\right){|\left\{{\vec{x}}_{a}\right\}\rangle }_{S}\;,$$
where $|\left\{{\vec{x}}_{a}\right\}\rangle_{S}$ is a completely symmetrized state over $S_{N}$,(154)$$|\left\{{\vec{x}}_{a}\right\}\rangle_{S}=P_{S_{N}}|\left\{{\vec{x}}_{a}\right\}\rangle =\frac{1}{N!}\sum_{\sigma \in S_{N}}|\left\{{\vec{x}}_{\sigma (a)}\right\}\rangle$$
we define(155)$$\psi_{S}\left(\left\{{\vec{x}}_{a}\right\}\right):=\frac{1}{N!}\sum_{\sigma \in S_{N}}\psi \left(\left\{{\vec{x}}_{\sigma (a)}\right\}\right)$$
such that |ψ〉 can be embedded into (Equation 136) with $\left\{{\vec{x}}_{1},\ldots ,{\vec{x}}_{a}\right\}$ gauge fixed within Σ and the rest in $\bar{\Sigma }$ within each tensor sum. The resulting density matrix reduced inside (Equation 142) is as follows:(156)$${\tilde{\rho }}_{\Sigma }=\overset{N}{\underset{a=0}{⨁}}p_{\Sigma ,a}\;{\hat{\rho }}_{\Sigma ,a}\;,$$
with(157)$$p_{\Sigma ,i\neq 0,N}:=\left(\begin{array}{c} N\\ a \end{array}\right)\int \left(\prod_{l=1}^{N}d^{D}{\vec{x}}_{l}\right)\left(\prod_{b=1}^{a}\theta_{\Sigma }\left({\vec{x}}_{b}\right)\right)\left(\prod_{c=a+1}^{N}\theta_{\bar{\Sigma }}\left({\vec{x}}_{c}\right)\right){\left|\psi_{S}\left(\left\{{\vec{x}}_{b},{\vec{x}}_{c}\right\}\right)\right|}^{2}\;,$$
and(158)$$\begin{array}{cc} {\hat{\rho }}_{\Sigma ,a\neq 0,N}:=p_{\Sigma ,a}^{-1}\int \left(\prod_{b=1}^{a}d^{D}{\vec{x}}_{b}d^{D}y_{b}\right) & \left(\prod_{c=a+1}^{N}d^{D}{\vec{x}}_{c}\right)\left(\prod_{b=1}^{a}\theta_{\Sigma }\left({\vec{x}}_{b}\right)\theta_{\Sigma }\left({\vec{y}}_{b}\right)\right)\left(\prod_{c=a+1}^{N}\theta_{\bar{\Sigma }}\left({\vec{x}}_{c}\right)\right)\\ \; & \times \psi_{S}{\left(\left\{{\vec{y}}_{b},{\vec{x}}_{c}\right\}\right)}^{*}\psi_{S}\left(\left\{{\vec{x}}_{b},{\vec{x}}_{c}\right\}\right)|\left\{{\vec{x}}_{b}\right\}\rangle \langle \left\{{\vec{y}}_{b}\right\}|\;, \end{array}$$
(the special cases of a=0,N being obvious from the two-particle example). The resulting entanglement entropy generically contains a Shannon term as well as genuine quantum entanglement (for the terms where Σ and $\bar{\Sigma }$ share subsets of particles): (159)$$S_{\Sigma }=-\sum_{a=0}^{N}p_{\Sigma ,a}\log p_{\Sigma ,a}-\sum_{a=1}^{N-1}p_{\Sigma ,a}\mathrm{tr}\left({\hat{\rho }}_{\Sigma ,a}\log{\hat{\rho }}_{\Sigma ,a}\right)\;.$$

### 6.3. Single Matrix

We now move our discussion to the target space entanglement of matrices. We will open this by investigating the entanglement of states involving a single N×N matrix. This problem, in fact, shares many features with the previous section on *N* identical particles, as we will soon see.

To set the stage, suppose we have a quantum mechanics of with a single N×N Hermitian matrix, X(t). We will suppose that this model possesses the U(N) gauge redundancy given by the following:(160)$$X\to UXU^{†}\;,\;U\in U\left(N\right)\;.$$
For instance, we can have in mind the c=1 matrix model given by (Equation 7), which we will investigate in more detail shortly. We can utilize the gauge redundancy (Equation 160) to diagonalize *X*:(161)$$X=U^{†}\;\Lambda \;U\;,\;\Lambda =\mathrm{diag}\left(\lambda_{1},\lambda_{2},\ldots ,\lambda_{N}\right)\;,$$
where $\lambda_{a}$ are the eigenvalues of *X*. This gauge fixing makes evident that the physical degrees of freedom are the target space spanned by $\left\{\lambda_{a}\right\}$ which we can think of as the coordinates of *N* D0 branes distributed along a line. There is an additional redundancy in (Equation 161) which is the Weyl symmetry exchanging eigenvalues. Thus, the eigenvalue quantum mechanics of a single matrix are that of identical particles on a line. Upon the reduction (Equation 161), physical wavefunctions pick up a Van der Monde determinant that renders them anti-symmetric under the exchange of eigenvalues. Thus, the $\left\{\lambda_{a}\right\}$ are identical, spin-less, fermions.

The gauge-fixing (Equation 161) corresponds to the breaking of the U(N) redundancy to $S_{N}$ and within this fixing, we can reduce the problem of target space entanglement to that of the previous section. However, we will also want to consider more general classes of gauge-fixing. To this end, it will be convenient to rewrite the definition of the subalgebra in terms of traces of functions of *X*. We do this by first defining the following [127]:(162)$$\Theta_{\Sigma }:=\theta \left(f\left(X\right)-c1\right),$$
where f(X) is some (possibly infinite) polynomial in *X*, c is some constant, and θ(x) is the Heaviside step function. Evaluated in the basis where f(X) is diagonal, $\Theta_{\Sigma }$ is a diagonal matrix with entries that are 0 or 1, and is, therefore, recognized as a projection matrix (in principle, we have to define what the step function θ(x) does at exactly x=0, but because the an eigenvalue of f(X) is only precisely c on a measure zero set of matrix configurations this is not an issue) that satisfies $\Theta_{\Sigma }^{2}=\Theta_{\Sigma }$. In essence, $\Theta_{\Sigma }$ is a basis-independent generalization of the subregion projectors we defined for identical particles, (Equation 117). The subalgebra of observables associated to some subregion of target space is, therefore, generated by strings of traces of $\Theta_{\Sigma }X^{i}\;\Theta_{\Sigma }$ and its canonical conjugate.

#### Example: The c = 1 Matrix Model

Here, we illustrate these ideas to explore entanglement in the c=1 matrix model reviewed in Section 3.1. We will primarly follow [118,128], although see [129,130] for important prior work. To state the setup and our expectations, we recall from Section 3.1 that the string theory description contains a scalar tachyon which far away from its potential wall, x→−∞ and in the weak coupling limit, μ→∞, is effectively massless. We consider the target space entanglement associated to interval $\Sigma =[x_{1},x_{2}]$ to the left of the tachyon wall. In the massless limit, $x_{1,2}\to -\infty$, μ→∞, holding ∆x fixed we might expect to find an entanglement entropy of massless scalar in two-dimensions:(163)$$S_{\Sigma }=\frac{1}{3}\log\frac{∆x}{\epsilon }\;,$$
for an appropriate UV-cutoff, ϵ. Away from this limit, we expect this divergence to be resolved by the finite degrees of freedom of the matrix model. This is what we will we see shortly.

We will proceed in exactly the gauge-fixing mentioned above: we ‘use-up’ the gauge redundancy by diagonalizing the matrix *X* and ordering its eigenvalues, $\left\{\lambda_{a}\right\}$. Physical wavefunctions of $\left\{\lambda_{a}\right\}$ come with a van der Monde determinant (as in Section 3.1):(164)$$\Psi \left(\lambda_{a}\right)=∆\left(\lambda_{a}\right)\psi \left(\lambda_{a}\right)$$
which render them as spin-less fermions. As shown in Section 3.1, the fermions are decoupled, yet see a potential that is an inverted harmonic oscillator plus a quartic correction. The entanglement entropy of identical non-interacting fermions in this target space can be reduced to studying correlations of the subregion density operator [131,132](165)$$N_{\Sigma }:=\int_{\Sigma }d\;\lambda \;\rho \left(\lambda \right)\;,$$
where we recall that ρ(λ) is the collective field density, (Equation 29), which in terms of the second-quantized wavefunction of λ (thought as a fermionic variable) is as follows:(166)$$\rho \left(t,\lambda \right)=\Psi^{†}\left(t,\lambda \right)\Psi \left(t,\lambda \right)\;.$$
Note that ρ here shouldn’t be confused with a reduced density matrix. In terms of the matrix eigenvalues, $\left\{\lambda_{a}\right\}$, $N_{\Sigma }$ is essentially the trace of the projector we introduced earlier, (Equation 162):(167)$$N_{\Sigma }=\sum_{a}\theta \left(\lambda_{a}\in \Sigma \right)\;.$$
The entanglement entropy of free fermions admits an expansion over cumulants of $N_{\Sigma }$ [131,132]: (168)$$S_{\Sigma }=\frac{\pi^{2}}{3}V_{\Sigma }^{\left(2\right)}+\frac{\pi^{4}}{45}V_{\Sigma }^{\left(4\right)}+\frac{2\pi^{6}}{945}V_{\Sigma }^{\left(6\right)}+\ldots \;,\;V_{\Sigma }^{\left(\ell \right)}={\left(-i\frac{d}{d\alpha }\right)}^{\ell }\log\langle e^{i\alpha N_{\Sigma }}\rangle {|}_{\alpha =0}\;.$$
In particular, the weak coupling limit is also a limit of large fermion occupation number. In this limit, it is the second cumulant, $V_{\Sigma }^{\left(2\right)}$, which provides the dominant contribution to $S_{\Sigma }$ (This statement is justified *post facto* in [118] by computing the first correction $V_{\Sigma }^{\left(4\right)}$ to see that it does not contribute to the leading log divergence) [133](169)$$S_{\Sigma }\approx \frac{\pi^{2}}{3}\int_{\Sigma }d\lambda \;d\lambda^{\prime }\left(\langle \rho \left(\lambda \right)\rho \left(\lambda^{\prime }\right)\rangle -\langle \rho \left(\lambda \right)\rangle \langle \rho \left(\lambda^{\prime }\right)\rangle \right)$$
and can be expressed entirely in terms of the fermion two-point functions. In particular, in the weak coupling limit we can express Ψ in terms of single particle wavefunctions, $\psi_{\nu }\left(\lambda \right)$, canonically quantized with respect to the Dirac sea filled up to the chemical potential, μ. We will take the convention as in [118] that both μ and ν are negative energies downwards from zero. The following occurs as a result: (170)$$S_{\Sigma }=\frac{\pi^{2}}{3}\int_{\mu }^{\infty }d\nu_{1}\int_{-\infty }^{\mu }d\nu_{2}{\left(\int_{\Sigma }d\lambda \;\psi_{\nu_{1}}\left(\lambda \right)\psi_{\nu_{2}}\left(\lambda \right)\right)}^{2}\;.$$
We must now evaluate at this for the single-particle wavefunctions solving the Schrödinger equation at energy −ν and potential (Equation 28):(171)$$-\frac{1}{2}\frac{d^{2}}{{d\lambda }^{2}}\psi_{\nu }\left(\lambda \right)+V_{c=1}\left(\lambda \right)\psi_{\nu }\left(\lambda \right)=-\nu \;\psi_{\nu }\left(\lambda \right)\;.$$
However, note that, since occupied states have ν>μ, in the large μ limit, these wavefunctions can be solved by a WKB approximation with $V_{c=1}\left(\lambda \right)\approx -\frac{1}{2}\lambda^{2}$ (in the large *N* limit of the matrix model). The WKB wavefunctions for the oscillatory (classically allowed) region, $\lambda >\sqrt{2\nu }$, is given by the following: (172)$$\psi_{\nu }\left(\lambda \right)=\sqrt{\frac{2}{\pi p(\lambda )}}\sin\left(\int_{\sqrt{2\nu }}^{\lambda }\;d\lambda^{\prime }\;p\left(\lambda^{\prime }\right)-\frac{\pi }{4}\right)\;,\;p\left(\lambda \right)=\sqrt{\lambda^{2}-2\nu }\;.$$
The integrals contributing to the entanglement entropy, Equation (Equation 170), can be performed. The authors of [118] do so by splitting the integral into oscillatory contribution with $\nu_{1}∼\nu_{2}∼\mu$ and a non-oscillatory contribution. Importantly, there is a potential logarithmic singularity that exactly cancels between these two contributions leaving a finite answer as promised. This is most naturally stated in terms of a “time of flight” variable, τ(λ): (173)$$\tau \left(\lambda \right)=-\frac{1}{\sqrt{2}}\int_{\sqrt{2\mu }}^{\lambda }\frac{\lambda^{\prime }}{\sqrt{-V_{c=1}\left(\lambda^{\prime }\right)-\mu }}=-\log\frac{\lambda +\sqrt{\lambda^{2}-2\mu }}{\sqrt{2\mu }}\;.$$
The result is the expression of the entanglement entropy for an interval $\Sigma =[\lambda_{1},\lambda_{2}]$ as follows:(174)$$S_{\Sigma }=\frac{1}{3}\log\frac{\tau \left(\lambda_{2}\right)-\tau \left(\lambda_{1}\right)}{\tau \left(\lambda_{2}\right)+\tau \left(\lambda_{1}\right)}+\frac{1}{6}\log\left(\tau \left(\lambda_{1}\right)\tau \left(\lambda_{2}\right)\right)+\frac{1}{3}\log\left(8e^{\gamma_{E}}\mu \sinh\tau \left(\lambda_{1}\right)\sinh\tau \left(\lambda_{2}\right)\right)\;,$$
where $\gamma_{E}$ is the Euler–Mascheroni constant. We can put this in a more illuminating form by noting that the time of flight variable, (Equation 173), relates the fermionic coordinate to the target space position, *x* [134]. We also introduce a rescaled string coupling, as follows:(175)$$\tilde{g}{\left(\lambda \right)}^{-1}=2\mu \sinh^{2}\left(\tau \left(\lambda \right)\right)\;,$$
which in the weak coupling limit, τ(λ)=x→−∞, is related to the usual string coupling, $g=e^{2x}$, as follows:(176)$$\lim_{x\to -\infty }\tilde{g}\left(x\right)=\frac{g(x)}{2\mu }\;.$$
This allows the expression of the entanglement entropy as follows:(177)$$S_{\Sigma }=\frac{1}{3}\log\frac{x_{2}-x_{1}}{\sqrt{\tilde{g}\left(x_{1}\right)\tilde{g}\left(x_{2}\right)}}+\frac{1}{6}\log\frac{16\;e^{2\gamma_{E}}\;x_{1}x_{2}}{{\left(x_{1}+x_{2}\right)}^{2}}+\ldots \;,$$
with the … coming from the contribution of higher cumulants in (Equation 168). The leading contribution to the entanglement entropy in the weak coupling limit, $x_{1,2}\to -\infty$ indeed matches the expected behavior of a conformal field theory describing its bosonized excitations, (Equation 163). However, the UV divergence has been replaced by the string coupling. This is reminiscent of the phenomena in the Bekenstein–Hawking entropy or the Ryu–Takayanagi holographic entanglement entropy in which an inverse coupling (Newton’s constant in that context) replaces the length scale that would have otherwise been filled by a UV cutoff (as is often the case in quantum field theory). In this case the finiteness of the entanglement entropy stems from the finite depth of the Fermi sea at position λ at finite (but large) μ: this finite depth, which is related to $\tilde{g}$ at λ, cuts off the amount of entanglement [129,130]. As μ→∞ this depth approaches infinity and $S_{\Sigma }$ diverges in the manner expected in (Equation 163).

### 6.4. Multiple Matrices

The additional complication in the case of multiple matrices, ${\left\{X^{i}\right\}}_{i=1,\ldots ,D}$, arises because, depending on the state in question, the $X^{i}$ are not necessarily simultaneously diagonalizable. We must then pick a prescription for how to assign the off-diagonal degrees of freedom to the various regions of target space. This requires only a slight modification to the procedure outlined in Section 6.3. To begin with, the projection matrix (Equation 162) is still perfectly well defined—we choose some Hermitian polynomial $f(X^{i})$ of the *D* matrices $X^{i}$, and plug it into the step function [127]:(178)$$\Theta_{\Sigma }:=\theta \left(f\left(X^{i}\right)-c1\right)\;,\;\Theta_{\bar{\Sigma }}:=1-\Theta_{\Sigma }\;.$$
Each matrix $X^{i}$ is now naturally decomposed into four blocks (see Figure 12):(179)$$X_{AB}^{i}:=\Theta_{A}X^{i}\Theta_{B},\;X^{i}=\sum_{AB}X_{AB}^{i}\;,\;A,B\in \left\{\Sigma ,\bar{\Sigma }\right\}\;.$$

The projector defines a breaking of (Equation 224) breaks the U(N) redundancy of the model to a U(M)×U(N−M), for some *M*. To see this we can work in basis where $f(X^{i})$ in (Equation 162) is diagonal and ordered from largest to smallest such that $\Theta_{\Sigma }$ takes the following form:(180)$$\Theta_{\Sigma }=\left(\begin{array}{cccc} \theta (f_{1}-c) & 0 & \ldots & 0\\ 0 & \theta (f_{2}-c) & \ldots & 0\\ \vdots & \vdots & \ddots & \vdots \\ 0 & 0 & \ldots & \theta (f_{N}-c) \end{array}\right)=\left(\begin{array}{cc} 1_{M\times M} & 0_{M\times (N-M)}\\ 0_{(N-M)\times M} & 0_{\left(N-M)\times (N-M\right)} \end{array}\right)$$
where $\left\{f_{i}\right\}$ are the eigenvalues of *f* and $f_{M}$ is the smallest eigenvalue greater than *c*. (Equation 180) makes clear that there is a U(M)×U(N−M) preserving this basis, while elements in $F:=U\left(N\right)/\left(U(M)\times U(N-M)\right)$ “mix” the matrix blocks of $\Theta_{\Sigma }$. For MQMs admitting states that lead to fuzzy geometries discussed in Section 4, U(M)×U(N−M) are analogous volume preserving diffeomorphisms that act internally and preserving a subregion, Σ, and the location of its entangling surface, ∂Σ, while maps in F deform Σ and moves its boundary.

It is important to stress at this point that, although $\Theta_{\Sigma }$ defines a canonical breaking of U(N)→U(M)×U(N−M), preserving its diagonal basis, the partition of the matrix degrees of freedom defined by (Equation 224) is a *gauge invariant partitioning* [81,93,127] because $\Theta_{\Sigma }$ is a function of *X* and so U(N) acts on it as well. It is only once we fix a basis (say one diagonalizing *f*) that we break the redundancy. The clearest way to see the gauge invariance is to note that operators of the form $\mathrm{Tr}[{\left(\Theta_{\Sigma }X^{i}\Theta_{\Sigma }\right)}^{q}]$, which are generators of the sub-algebra, are invariant under the full U(N). This is morally similar to the statement that a subregion as a subset of points a manifold is independent of a coordinatization of the manifold; it is only after coordinates are fixed does the subregion break diffeomorphisms into those preserving its coordinate locations and those not. We will return to these points in Section 8. However, for now, we will explain the general structure of factorization that $\Theta_{\Sigma }$ induces followed by how non-commutative edge modes arise from the residual U(M)×U(N−M) preserving a fixed basis. We will return to these edge modes in Section 7 showing that for MQMs with states strongly peaked on non-commutative geometries they lead to area law entanglement.

In order to proceed we will introduce an extended Hilbert space and tensor product partition of it. A natural extended Hilbert space is simply the span of all Hermitian matrix elements of $\left\{X^{i}\right\}$:(181)$$\begin{array}{cc} H_{\mathrm{ext}}= & {\mathrm{span}}_{C}\left\{|X^{i}\rangle \;|\;{\left(X^{i}\right)}^{†}=X^{i}\;,i=1,\ldots ,D\right\}\;,\\ = & \overset{D}{\underset{i=1}{⨂}}\left[\underset{1\leq a<b\leq N}{⨂}{\mathrm{span}}_{C}\left\{|X_{ab}^{i}\rangle \;|\;X_{ab}^{i}\in C\right\}⊗\overset{N}{\underset{a=1}{⨂}}{\mathrm{span}}_{C}\left\{|X_{aa}^{i}\rangle \;|\;X_{aa}^{i}\in R\right\}\right]\;. \end{array}$$
The second line is written explicitly as the inner-product on $H_{\mathrm{ext}}$ is a Dirac delta function on each of the factors. In a fixed basis, the matrix blocks distinguished by (Equation 224) have a natural corresponding tensor product decomposition of the extended Hilbert space:(182)$$H_{\mathrm{ext}}=H_{\Sigma \Sigma }⊗H_{\Sigma \bar{\Sigma }}⊗H_{\bar{\Sigma }\bar{\Sigma }}\;.$$
States in $H_{\Sigma \Sigma }$ are in the Adjoint representation of U(M) and likewise for $H_{\bar{\Sigma }\bar{\Sigma }}$ and U(N−M), while states in $H_{\Sigma \bar{\Sigma }}$ are in the bifundamental representation of U(M)×U(N−M). i.e., for a basis configuration and a $\left(U,\bar{U}\right)\in U\left(M\right)\times U\left(N-M\right)$(183)$$|\;X_{\Sigma \Sigma }^{i}\;,\;X_{\Sigma \bar{\Sigma }}^{i}\;,\;X_{\bar{\Sigma }\bar{\Sigma }}^{i}\;\rangle \to |\;UX_{\Sigma \Sigma }^{i}U^{†}\;,\;UX_{\Sigma \bar{\Sigma }}^{i}{\bar{U}}^{†}\;,\;\bar{U}X_{\bar{\Sigma }\bar{\Sigma }}^{i}{\bar{U}}^{†}\;\rangle \;.$$
Because of the off-diagonal factor, $H_{\Sigma \bar{\Sigma }}$, there is not a canonical separation of $H_{\mathrm{ext}}=H_{\Sigma }⊗H_{\bar{\Sigma }}$ and we have to make a choice how to place the off-diagonal matrix elements. In terms of D0-brane physics, these off-diagonal elements represent strings stretching across ∂Σ between branes inside and outside Σ and we must choose such open strings are assigned to the subregion or not. There are now two natural choices of subalgebra, $A_{\Sigma }$, associated to two choices of factorization, as sketched in Figure 12. The first subalgebra choice is generated traces of products of just $X_{\Sigma \Sigma }^{i}$, keeping only one of the diagonal matrix blocks, and is generated by strings of traces of $X_{\Sigma \Sigma }^{i}$ and their canonical conjugates. This is the choice of subalgebra considered in [81,98,127,135].

The second choice keeps *two* blocks, the diagonal block and one of the off-diagonal components, as depicted on the right subfigure of Figure 12. This decomposition is more subtle, due to the Hermiticity constraint, which enforces $X_{\Sigma \bar{\Sigma }}^{i†}=X_{\bar{\Sigma }\Sigma }^{i}$. We can deal with this by noting that for some color indices *a* and *b*, $X_{ab}^{i}$ and $X_{ba}^{i}$ together contain two real degrees of freedom. We need some prescription for partitioning these two into $X_{ab,\Sigma }$ and $X_{ba,\bar{\Sigma }}$. To ensure that the degrees of freedom associated to the two subregions still transform as bi-fundamentals under U(M) and U(N−M), we take $X_{ab;\Sigma }$ and $X_{ba;\bar{\Sigma }}$ to be linear combinations of $X_{ab}$ and $X_{ba}$. The most general such prescription is to take the following:(184)XΣΣ¯;Σi:=ReeiαXΣΣ¯i,XΣ¯Σ;Σ¯i:=ImeiαXΣΣ¯i,
for some choice of phase $e^{i\alpha }$. This choice of subalgebra is considered in [93,136]. It is useful to note that both choices of subalgebra reduce to Section 6.3 when we only have a single matrix.

In what follows we will make the former assignment of $A_{\Sigma }$ corresponding to a factorization:(185)$$H_{\Sigma }:=H_{\Sigma \Sigma }\;,\;H_{\bar{\Sigma }}:=H_{\Sigma \bar{\Sigma }}⊗H_{\bar{\Sigma }\bar{\Sigma }}\;,$$
i.e., only strings beginning and ending on branes in Σ are assigned to Σ.

Although (Equation 224) is a U(N) invariant decomposition, the factorization (Equation 182) is definitely not and U(N) not only acts within individual tensor factors, but also exchanging matrix elements from different tensor factors. There are multiple isomorphic ways to embed the Hilbert space of physical states within $H_{\mathrm{ext}}$ corresponding to different factorization maps. One such Hilbert space is the *invariant Hilbert space* which is annihilated by matrix elements of each generator of U(N):(186)$$H_{\mathrm{inv}}:=\left\{|\Psi \rangle \in H_{\mathrm{ext}}\;|\;{\hat{G}}_{ab}|\Psi \rangle =0\right\}$$
where the generator is given by the following:(187)$${\hat{G}}_{ab}=2i\sum_{i=1}^{D}\;\left(X_{ac}^{i}\Pi_{cb}^{i}-X_{cb}^{i}\Pi_{ac}^{i}\right):=2i\sum_{i=1}^{D}\;:{\left[X^{i},\Pi^{i}\right]}_{ab}:\;,$$
where the normal-ordering :: places all Π’s to the right.

An alternative embedding of physical states is given by choosing a gauge-fixing to eliminate all of the redundant degrees of freedom and express physical states as wavefunctions of a particular ‘gauge-slice’ of physical degrees of freedom. In the MQM of a single matrix, for example, such a slice is given by diagonalizing *X* and ordering its eigenvalues as in Section 6.3. Such a Hilbert space will be denoted as $H_{\mathrm{gf}}$. Of course there are infinitely many such gauge-fixed Hilbert spaces all related by moving along a gauge orbit; these are isomorphic to each to other and to $H_{\mathrm{inv}}$.

We can also imagine a middle ground between a fully gauged fixed Hilbert space, $H_{\mathrm{gf}}$ and a gauge-averaged Hilbert space, $H_{\mathrm{inv}}$, by partially gauge-fixing some degrees of freedom and averaging the rest over a suborbit of U(N). In what follows, it is this partial gauge-fixed embedding of physical states that we will consider. Specifically, below we will describe the situation where we partially gauge-fix U(N)/G for some subgroup *G* and construct a physical Hilbert as the orbit of *G*.

Within the factorization (Equation 182) and (Equation 185), U(M) acts on $H_{\Sigma }=H_{\Sigma \Sigma }$ as well on $H_{\bar{\Sigma }}$ through the $H_{\Sigma \bar{\Sigma }}$ factor it contains. We can further decompose the degrees of freedom in $H_{\Sigma }$ in terms of fully gauge-fixed states and their orbit under U(M):(188)$$H_{\Sigma }=H_{U(M)}⊗H_{\Sigma ,\mathrm{gf}}\;,$$
and states in $H_{\Sigma }$ can be expressed as follows:(189)$$|\tilde{\psi }\rangle =|U\rangle |{\tilde{\psi }}_{\mathrm{gf}}\rangle \;,\;U\in U\left(M\right)\;,$$
where $|{\tilde{\psi }}_{\mathrm{gf}}\rangle$ is spanned a completely gauge fixed configuration of $|X_{\Sigma \Sigma }^{i}\rangle$.

Under the Peter–Weyl theorem $H_{U(M)}$ can be expressed as a direct sum over irreducible representations of U(M) which (mirroring notation from Section 5.1) will be labelled by μ and so the general structure of the Hilbert space factor is as follows:(190)$$H_{\Sigma }=\left(\underset{\mu }{⨁}H_{\mu }\right)⊗H_{\Sigma ,\mathrm{gf}}\;.$$

Lastly, we note an alternative proposal for associating geometry to eigenvalues (and, therefore, choosing a notion of geometric subregion) due to [84,137]: for a state $\Psi (X^{i})$, the target space geometry is associated to the eigenvalues of the expectation value $Y^{i}:=\langle X^{i}\rangle$. For instance, ref. [84] considers wave-packets sharply localized to the configuration $Y^{i}$. Because $X_{ab}^{i}$ is not a gauge invariant operator, this manifestly requires a working with a gauge-fixed state in the extended Hilbert space, as $Y^{i}=0$ for any gauge invariant wavefunction. Subregions are associated to separated wave-packets distributed in this target space [137]. Since spec$\left(\langle X^{i}\rangle \right)$ is a nonlinear operation on a state, this is a distinct gauge-fixed factorization than that described underneath (Equation 187). We will see an example of this factorization in Section 7.1.

#### 6.4.1. Edge Modes in the Extended Hilbert Space

We now describe how for partially gauge-fixed states the decomposition (Equation 190) leads to maximally entangled matrix ‘edge modes.’ We will assume that the subgroup, *G*, over which we average physical states in $H_{\mathrm{ext}}$ includes U(M)×U(N−M). This implies that the any physical state $|\tilde{\psi }\rangle$ is annihilated by the U(M) subgroup of the U(N) quantum generators, which are the ΣΣ block of (Equation 187):(191)$${\hat{G}}_{\Sigma \Sigma }=2i\sum_{i=1}^{D}:\left[X_{\Sigma \Sigma }^{i},\Pi_{\Sigma \Sigma }^{i}\right]:+2i\sum_{i=1}^{D}\left(X_{\Sigma \bar{\Sigma }}^{i}\Pi_{\bar{\Sigma }\Sigma }^{i}-X_{\bar{\Sigma }\Sigma }^{i}\Pi_{\Sigma \bar{\Sigma }}^{i}\right):={\hat{G}}_{\Sigma }+{\hat{G}}_{\bar{\Sigma }}\;.$$
Notice that ${\hat{G}}_{\Sigma }$ and ${\hat{G}}_{\bar{\Sigma }}$ are the $H_{\Sigma }$ and $H_{\bar{\Sigma }}$ tensor factors of (Equation 185), respectively. Notice that this is explicitly in the form we encountered earlier (Equation 102) in our discussion of entanglement and gauge invariance in Section 5.1. There we saw that this structure arose from the action of gauge transformations on edge modes localized to ∂Σ. Here, the interpretation is wholly similar and we will make this connection more concrete in Section 7 when discussing entanglement on non-commutative geometries.

Much like the story in Section 5.1, this structure and U(M) invariance of physical states leads to maximal entanglement of these edge modes. In particular, both ${\hat{G}}_{\Sigma }$ and ${\hat{G}}_{\bar{\Sigma }}$ individually generate U(M). A U(M) invariant physical state $\tilde{\rho }=|\tilde{\psi }\rangle \langle \tilde{\psi }|$ satisfies the following:(192)$$〚{\hat{G}}_{\Sigma \Sigma },\tilde{\rho }〛=0\;.$$
The split (Equation 191) implies for the state ${\tilde{\rho }}_{\Sigma }$, reduced on $H_{\Sigma }$:(193)$$〚{\hat{G}}_{\Sigma },{\tilde{\rho }}_{\Sigma }〛=0\;,$$
as ${\hat{G}}_{\bar{\Sigma }}$ acts on a separate tensor factor. Thus, as we saw before, under Schur’s Lemma, the reduced state is maximally mixed on the representations of U(M): (194)$${\tilde{\rho }}_{\Sigma }=\underset{\mu }{⨁}\;p_{\mu }\frac{1_{\mu }}{d_{\mu }}⊗{\hat{\rho }}_{\mu }\;,$$
where we remind the reader that $p_{\mu }$ is the probability of the state being in the μ representation, $d_{\mu }$ is the dimension of that representation, and ${\hat{\rho }}_{\mu }$ is the reduced state of the degrees of freedom in $H_{\Sigma ,\mathrm{gf}}$ in the μ block of (Equation 190). Thus, the entanglement induced by U(M) invariance (what [81] calls ‘Gauss law entanglement’) takes the following form: (195)$$S_{\Sigma }=\sum_{\mu }p_{\mu }\log d_{\mu }-\sum_{\mu }p_{\mu }\log p_{\mu }-\sum_{\mu }\mathrm{tr}\left({\hat{\rho }}_{\mu }\log{\hat{\rho }}_{\mu }\right)\;.$$
The first term is the expectation value of the representation dimension in the state, $\langle \log d_{\mu }\rangle$, the second is a Shannon entropy of the representation distribution in the state, $S_{\mathrm{Shannon}}\left[\left\{p_{\mu }\right\}\right]$, and the final term is an ‘interior’ entanglement of singlet fluctations within a given representation block, $S_{\Sigma }\left[{\hat{\rho }}_{\mu }\right]$. Of particular interest is the first term: we will show by example below, and more generally in Section 7, that for sufficiently ‘geometric’ states in certain MQMs, there is a dominating representation with $p_{\mu^{*}}\approx 1$ and whose dimension can be interpreted as an area law.

Before moving onward, we also pause to mention that in our general set-up (Equation 195) is not yet our final answer for the entanglement entropy in the cases that orbit average subgroup G≠U(M)×U(N−M). That is, we have only considered the consequences of U(M) invariance on reduced states; however, we have yet to incorporate invariance of the state on the G/(U(M)×U(N−M)). We will revisit this point in Section 8, however for now we simply note that for factorization maps based on partial gauge fixing U(N)→G=U(M)×U(N−M), (Equation 195) represents the final expression.

#### 6.4.2. Example: Matrix Quantum Hall

As a simple example, we first apply this technology to the Matrix quantum Hall model reviewed in Section 4.3, following [136]. For simplicity we work in the classical limit, taking the coupling *k* large. Our starting point is the Gauss law, (Equation 82), which we recall here:(196)$$G_{ab}=:{\left[Z,Z^{†}\right]}_{ab}:+\phi_{b}^{†}\phi_{a}-\left(k-1\right)\delta_{ab}=0\;.$$
This constraint annihilates all physical states of the system. To avoid clutter we suppress the explicit normal ordering in future such expressions—this will in any case only result in 1/k corrections to resulting expressions.

We now decompose the matrices into sub-blocks conjugating with the projector $\Theta_{\Sigma }$ as in Section 6.4. The ΣΣ sub-block of the Gauss constraint is given by the following:(197)$$\Theta_{\Sigma }G\Theta_{\Sigma }=\left[Z_{\Sigma \Sigma },Z_{\Sigma \Sigma }^{†}\right]+Z_{\Sigma \bar{\Sigma }}Z_{\bar{\Sigma }\Sigma }^{†}-Z_{\Sigma \bar{\Sigma }}^{†}Z_{\bar{\Sigma }\Sigma }+\phi_{\Sigma }\phi_{\Sigma }^{†}-\left(k-1\right)\Theta_{\Sigma }=0.$$
We note that $Z_{\Sigma \bar{\Sigma }}^{†}$ is the $\Sigma \bar{\Sigma }$ block of $Z^{†}$, and is *not* equal to ${\left(Z_{\Sigma \bar{\Sigma }}\right)}^{†}$. This constraint entangles the diagonal block $Z_{\Sigma \Sigma },$$\phi_{\Sigma }$ which act on $H_{\Sigma }$ and off-diagonal block $Z_{\Sigma \bar{\Sigma }}$ degrees of freedom which act on $H_{\bar{\Sigma }}$. We may interpret the off-diagonal blocks $Z_{\Sigma \bar{\Sigma }}$ and $Z_{\bar{\Sigma }\Sigma }$ as acting as sources for the non-U(M)-singlet modes in the ΣΣ degrees of freedom. In particular, note that the $Z_{\Sigma \bar{\Sigma }}$ modes transform as fundamentals under U(M) transformations preserving the subspace defined by the projector $\Theta_{\Sigma }$. As in the non-commutative disc described in Section 4, fundamentals define boundaries of non-commutative manifolds, so it is natural to identify $Z_{\Sigma \bar{\Sigma }}$ as edge modes in the language of [26,27].

To make progress, we must choose a wavefunction and a subsystem of which to compute the entanglement. The wavefunction we pick is the ground state of the Hamiltonian $H=\mathrm{Tr}[Z^{†}Z]$ given the constraint (Equation 196), (Equation 84), which we recall below: (198)$$|\Psi_{k}\rangle =N_{k}{\left[\epsilon^{i_{1}\ldots i_{N}}\phi_{i_{1}}^{†}{\left(\phi^{†}Z^{†}\right)}_{i_{2}}\ldots {\left(\phi^{†}Z^{†(N-1)}\right)}_{i_{N}}\right]}^{k-1}|0\rangle ,$$
where we remind the reader that 0 is the Fock space vacuum annihilated by all ϕ and *Z* annihilation operators, and $N_{k}$ is a normalization. In the large *k* limit, this wavefunction localizes to the classical minimum (Equation 79), in the following sense: (199)$$\langle \Psi_{k}|\mathrm{Tr}\left[{\left(Z^{†}Z\right)}^{q}\right]|\Psi_{k}\rangle ={\left(Z_{\mathrm{cl}}^{†}Z_{\mathrm{cl}}\right)}^{q}+O\left(qk^{-1}\right)\;.$$
This classical limit holds equally well when we gauge-fix the system by diagonalizing some Hermitian function of *Z* and $Z^{†}$, say the radial matrix $R^{2}:=Z^{†}Z$. For the wavefunction in any given choice of gauge ${|\Psi_{k}\rangle }_{g.f.}$, the localization of the wavefunction will ensure the following:(200)$${\langle \Psi_{k}|}_{g.f.}{\left(Z_{ab}\right)}^{†}Z_{ab}{|\Psi_{k}\rangle }_{g.f.}={\left|Z_{\mathrm{cl},ab}\right|}^{2}+O\left(k^{-1}\right),$$
with $Z_{\mathrm{cl},ab}$ taken to be the unitary rotation of (Equation 79) to the appropriate gauge. This now allows us to read off the structure of the sources for U(M) non-singlets in (Equation 197) simply by gauge fixing and reading off the structure of the classical matrix elements $Z_{\mathrm{cl},\Sigma \bar{\Sigma }}$, which in turn become geometric data of the fuzzy disc, as in Section 4.

As in [26,27], the edge modes are maximally entangled by the Gauss constraint, and their entanglement entropy is the log of the edge mode Hilbert space dimension. In this case, the operators $Z_{\Sigma \bar{\Sigma }}$, $Z_{\bar{\Sigma }\Sigma }$ are Fock operators furnishing this edge mode Hilbert space. A careful analysis of the appropriate gauge fixing and edge mode algebra was done in [136], but the result may be summarized by noting that it reduces to the counting problem of distributing $Z_{\Sigma \bar{\Sigma }}$ harmonic oscillator modes. The set of singular values of $Z_{\Sigma \bar{\Sigma }}$, let us call it $q_{i}$, is a U(M) invariant. The counting problem copmuting the edge mode Hilbert space dimension then comes from counting the number of ways to distribute $q_{i}$ quanta among U(M)-fundamental vectors, with each vector representing a singular vector of $Z_{\Sigma \bar{\Sigma }}$. At large *k*, these vectors are extracted using (Equation 200).

The result depends on the factorization map into the extended Hilbert space. As we mentioned in Section 5.1, there may exist multiple choices of factorization, and, in this case, there are two options that lead to distinct, interesting answers. The first is to take the full U(M) symmetry as a global symmetry of the edge modes and compute the logarithm of the dominant U(M) irrep dimension. Combining (Equation 200) and (Equation 197), we see that the edge mode structure is determined (in the large *k* limit) by the classical saddle ${\left(Z_{\mathrm{cl}}\right)}_{\Sigma ,\bar{\Sigma }}$. In particular, the relevant data are the singular values of this rectangular matrix (which are manifestly gauge-invariant). As computed in [136], the behavior of these singular values depends on the type of entanglement cut—whether or not it crosses the boundary of the disc (see Figure 13, and Appendix D of [136] for additional details). We call this **Prescription 1**.

The second option keeps an $S_{M}$ subgroup of U(M) gauged, so that the edge mode spectrum is organized into irreps of ‘$U\left(M\right)/S_{M}$’. $S_{M}$ is not a normal subgroup of U(M), so this object may seem a priori ill defined, but it has a natural interpretation as a non-invertible symmetry (A.F. would like to thank Arkya Chatterjee and Jacob McNamara for emphasizing this point) [138,139,140]. We call this **Prescription 2**.

In either case, denoting the singular values of ${\left(Z_{\mathrm{cl}}\right)}_{\Sigma \bar{\Sigma }}$ as $\left\{l_{a}\right\}$, we find entanglement entropies for the two prescriptions (up to 1/min(M,N−M) corrections) as follows: (201)$$\begin{array}{cc} \; & \mathrm{Prescription}1:\;S=\sum_{a}N^{2}k^{2}{\left|l_{a}\right|}^{2}\log\left(l_{a}/N\right), \end{array}$$(202)$$\begin{array}{cc} \; & \mathrm{Prescription}2:\;S=\sum_{a}\frac{Nk}{\sqrt{6}}\left|l_{a}\right|. \end{array}$$
For entanglement cuts of the form $\gamma_{1}$ in Figure 13, ${\left(Z_{\mathrm{cl}}\right)}_{\Sigma \bar{\Sigma }}$ has just one singular value $l_{1}$ of magnitude equal to the length $|\gamma_{1}|$. For this case we may, therefore, interpret (Equation 202) as a perimeter law and (Equation 201) as a ‘perimeter squared’ law. This perimeter squared law may seem pathological from a 2d geometry point of view, but, as emphasized in [141], this appears to be the correct behavior for entanglement cuts in the phase space picture of the collective field in c=1 string theory.

For entanglement cuts of the form $\gamma_{2}$, the $l_{a}$ are all proporional to the length of the cut $\gamma_{2}$ traces through the bulk of the disk, but the two distinct behaviors (Equation 201) and (Equation 202) pick up an additional multiplicative factor of logN. The interpretation of these deviations from geometric behavior is not currently clear.

## 7. Entanglement in Emergent Non-Commutative Geometries

In the previous section, we set up the necessary background to discuss the target space entanglement of MQMs by introducing a notion of factorization of matrix entries. We illustrated two examples, the c=1 matrix model and the matrix quantum Hall, where this entanglement entropy led to interpretation of a entanglement entropy in an emergent target space. The matrix quantum Hall example is particularly interesting because as we reviewed in Section 4.3, the target space of this model can be thought as the coordinates of a non-commutative disc. Correspondingly the fluctuations about the classical background is a non-commutative Chern–Simons field theory. As we reviewed in Section 4, the connection between MQM and non-commutative spaces and non-commutative field theories is generic and is tied to the low energy descriptions of D-brane physics where the non-commutativity is mediated by string interactions [88]. Thus, we discuss in more depth the entanglement of non-commutative geometries and non-commutative field theories.

In all of the examples to come, we may think of a non-commutative geometry as *classical* geometries of a MQM. That is, there may be a quantum wavefunction, Ψ(X), that is strongly peaked around a particular configuration, $X_{\mathrm{cl}}^{i}$, satisfying one of the fuzzy space algebras, (Equation 60), (Equation 68), or (Equation 70). Small fluctuations about this peak are described by an effective field theory living on a background fuzzy space. In considering entanglement on these spaces we may consider the entanglement of degrees of some subset of those degrees of freedom (such as the scalar sector of that effective field theory) or the fluctuations in their entirety. In some cases, particularly when a non-commutative space arises a D-brane worldvolume theory, the large *N* limit of that space and its effective field theory admit a holographic dual [63,142]. Within certain regimes where the supergravity approximation can be trusted, the dual geometries are standard (albeit, curved, non-AdS) geometries. For simple choices of subregion, Σ, one may then compute the entanglement entropy using the Ryu–Takayanagi formula [17] in the dual geometry:(203)$$S_{\Sigma }=\frac{\mathrm{Area}[\Gamma_{\Sigma }]}{4G_{N}}\;,$$
where $\Gamma_{\Sigma }$ is a bulk surface of minimal area anchored to ∂Σ at the boundary, and $G_{N}$ is the bulk gravitational constant [143,144,145,146]. A key feature in these calculations is the violation of the area law described in Section 5. The intuition for this violation of the area law stems from the UV/IR mixing described above, namely excitations that are short distance with respect to directions along ∂Σ are delocalized transverse to ∂Σ and vice-versa. Present in these holographic calculations is an explicit UV cutoff given roughly by the distance of the boundary to asymptotic conformal boundary, as is standard in the holographic dictionary. Interestingly, in considering the entanglement entropy of infinite strips, refs. [144,145] also find an intrinsic non-locality scale, $\ell_{\mathrm{NC}}$, in which the effects of the non-commutativity of boundary field become important. Strip regions with widths below $\ell_{\mathrm{NC}}$ follow volume law entanglement entropies, however for widths above a critical length of the order of $\ell_{\mathrm{NC}}^{2}/\ell_{\mathrm{UV}}$, (what one might regard as a ‘non-locality scale’ [147]) the authors note a sharp transition to area law entanglement. This critical length diverges as the UV cutoff is taken to infinity (or $\ell_{\mathrm{UV}}$ to zero).

In this review, we will instead focus on the effective field theory side of the story. Below we will begin with an illustrative example of a scalar field on a fuzzy sphere; this example will provide some lessons for the more general treatment of entanglement in non-commutative matrix geometries to follow.

### 7.1. Entanglement of a Free Field on the Non-Commutative Sphere

In this section, we will ‘dip our toes’ into some of the features of entanglement on non-commutative spaces by focussing on the ground state entanglement of a complex scalar field on the fuzzy sphere described by (Equation 70). Interpreting this scalar theory as an effective theory of fluctuations about a classical state in a multiple matrix MQM (such as the ‘mini-BMN’ model), we can interpret this entanglement entropy as the ‘interior’ contribution in the general structure of entanglement of multiple matrices, (Equation 195). We will follow closely the computations described in [148,149,150]. (See [151,152] for prior and related calculations.) As mentioned above, when the radius of the background sphere, *R*, is suitably quantized the algebra of functions can be represented by finite dimensional matrices, and here we will be interested in the case when that representation is irreducible, i.e., the following:(204)$$R^{2}=\nu^{2}\frac{N^{2}-1}{4}\;,$$
for an integer, *N*. We will ultimately be interested in the large *N* limit.

To introduce the non-commutative theory, we remind ourselves of the action of a free complex scalar field on a commutative spacetime: (205)$$S=\frac{1}{2}\int dtd^{D}\vec{x}\;\sqrt{g}\left({\left(\partial_{t}\phi \right)}^{2}-{\left(\nabla \phi \right)}^{2}-m^{2}\phi^{2}\right)\;,$$
which has an associated Hamiltonian(206)$$H=\frac{1}{2}\int d^{D}\vec{x}\;\sqrt{g}\left(\pi^{2}+{\left(\nabla \phi \right)}^{2}+m^{2}\phi^{2}\right)\;,$$
where $\pi =\dot{\phi }$ is the conjugate momentum. The corresponding non-commutative theory is constructed by promoting ϕ to an N×N Hermitian matrix, Φ, and the replacing the Laplacian by(207)$$\nabla^{2}\left(\cdot \right)\to -\frac{1}{R^{2}}\left[J^{i},\left[J^{i},\cdot \right]\right]\;,$$
where, again, $\left\{J^{i}\right\}$ generate su(2). Equation (Equation 207) follows from noting that $\left\{J^{i}\right\}$ generate the isometries of the sphere and so their Casimir yields the Laplacian. This is of course an example of the general statement in Section 4 that commutators of non-commutative coordinates define differential operators, Equation (Equation 52). Recalling the relation between the matrix trace and integration, Equation (Equation 54), we can then write down the Hamiltonian: (208)$$H=\frac{1}{2}\frac{4\pi R^{2}}{N}\mathrm{Tr}\left(\pi^{2}-\frac{1}{R^{2}}\left[J^{i},\Phi \right]\left[J^{i},\Phi \right]+m^{2}\Phi^{2}\right)\;,$$
where the conjugate momentum, $\pi =\dot{\Phi }$, is also an N×N Hermitian matrix.

In what follows we will consider the entanglement of this field theory reduced to Σ, a U(1) symmetric cap on the sphere, say, centered at the North pole. We can parametrize this cap by the azimuthal angle its border, ∂Σ, makes with North pole, as depicted in Figure 14. We will denote this angle as $\theta_{\mathrm{cap}}$.

We will need a prescription for ‘finding’ this subregion as a subsystem of the matrices Φ and π. We will follow the prescription described in [148], generalizing [151,152]. We will revisit the interpretation of this prescription when we treat the question of entanglement on non-commutative spaces more broadly afterwards.

The ‘color space’ of Φ is simply the *N*-dimensional representation of su(2), which we can span by states, {|m)}, that diagonalize $J^{3}$:(209)$$J^{3}\left|m\right)=m|m)\;,\;m=-j,-j+1,\ldots ,j-1,j\;,\;j=\frac{N-1}{2}\;.$$
We can associate a wavefunction on $S_{\nu }^{2}$ to |m) in the following way. Consider the unit vector, $\hat{n}$, pointing at (θ,φ) on $S_{\nu }^{2}$. It has an associated coherent state $|\hat{n})$ which is highest-weight with respect to $J^{3}$ in a tangent space centered at (θ,φ):(210)$${\hat{n}}_{i}J^{i}|\hat{n})=j|\hat{n})\;.$$
Importantly this determines its overlap with |m) and gives a notion for a wavepacket of |m) centered at (θ,φ):(211)$$\left(\hat{n}|m\right)=\sqrt{\frac{(2j)!}{(j+m)!(j-m)!}}{\left(\frac{1}{2}\sin\frac{\theta }{2}\right)}^{j-m}{\left(\frac{1}{2}\cos\frac{\theta }{2}\right)}^{j+m}e^{im\varphi }\;.$$
Similarly, the overlap of two coherent states(212)$$|\left({\hat{n}}_{1}|{\hat{n}}_{2}\right)|={\left(\frac{1+{\hat{n}}_{1}\cdot {\hat{n}}_{2}}{2}\right)}^{j}\;,$$
becomes sharply peaked in the 2j∼N→∞ limit, falling quickly to zero beyond an angular separation of $\chi =\arccos\left({\hat{n}}_{1}\cdot {\hat{n}}_{2}\right)∼\frac{2}{\sqrt{j}}$. In this limit we can think of a single coherent state as covering an approximate area of $R^{2}/N$; this is what is expected for a D2-brane of area $4\pi R^{2}$ divided into *N* units of flux. This then sets the non-commutativity cutoff scale, $\ell_{\mathrm{NC}}$ (the length scale where non-commutative effects become important) at the following:(213)$$\ell_{\mathrm{NC}}∼\frac{R}{\sqrt{N}}\;.$$
This is *parametrically larger* than the UV cutoff scale arrived at by dividing the total area, $4\pi R^{2}$, by the total number of degrees of freedom, $N^{2}$:(214)$$\ell_{\mathrm{UV}}∼\frac{R}{N}\;.$$
The mismatch between these two scales is a diagonstic of the UV/IR mixing in the model and has interesting implications for the entanglement problem at hand. For one, unlike the entanglement calculations in standard quantum field theory, where the entangling cut, ∂Σ, can be specified up to the UV cutoff, $\ell_{\mathrm{UV}}$, in this model the entangling cut will possess an inherent ‘fuzziness’ and we can only specify it up to angular separations of order $∆\theta ∼N^{-1/2}$.

The localization of coherent states (up to $\ell_{\mathrm{NC}}$) in the large *N* limit leads the authors of [148] to a particular subdivision of matrix elements to associate to the polar cap. In particular, the expectation value of a generic matrix element in a coherent state(215)$$|\left(\hat{n}|m_{1}\right)\left(m_{2}|\hat{n}\right)|=\frac{(2j)!}{\sqrt{\left(j+m_{1}\right)!\left(j-m_{1}\right)!\left(j+m_{2}\right)!\left(j-m_{2}\right)!}}\cos{\left(\frac{\theta }{2}\right)}^{2j+m_{1}+m_{2}}\sin{\left(\frac{\theta }{2}\right)}^{2j-m_{1}-m_{2}}\;,$$
is sharply peaked in the large j limit at(216)$$\theta ∼\theta_{0}=\arccos\left(\frac{m_{1}+m_{2}}{2j}\right)\;,$$
with, again, a variation of the order $∆\theta ∼j^{-1/2}∼N^{-1/2}$. Thus, in considering a region with $\theta \leq \theta_{\mathrm{cap}}$ the authors make the association of Σ to the matrix elements $\pi_{m_{1},m_{2}}$ and $\Phi_{m_{1},m_{2}}$ with(217)$$m_{1}+m_{2}>2j\;\cos\theta_{\mathrm{cap}}\;.$$
This is depicted on the right in Figure 14. When $\theta_{\mathrm{cap}}=\frac{\pi }{2}$ (i.e., Σ is the entire hemisphere), this matches the prescription of [151,152]. Note that this factorization based on coherent states is in line with the prescription of [137] and is decidely different than the prescription described in the body of Section 6.4.

We will compute the entanglement of the ground state with respect to this subdvision of matrix elements. Because the Hamiltonian, (Equation 208), is quadratic, the ground state wavefunction is Gaussian(218)$$\Psi \left[\Phi \right]=N\;\exp\left(-\frac{1}{2}\left(\Phi \;K^{1/2}\;\Phi \right)\right)\;,$$
where $K=\frac{4\pi }{N}\left(\left[J^{i},\left[J^{i},\cdot \right]\right]+m^{2}R^{2}\right)$, $K^{1/2}$ is its formal square root, and N is a normalization factor. Subsequently, tracing out specific matrix elements from the reduced density matrix of |Ψ〉〈Ψ| amounts to straightforward Gaussian integration and one can follow the general procedure described by Srednicki in the seminal work [9] for computing the entanglement entropy in discrete Gaussian states. For the system at hand the authors of [148] perform this calculation numerically. A sample plot of $S_{\Sigma }$ against the cap angle from that work is reproduced here in Figure 15.

We can immediately recognize from the plot in Figure 15 the violation of the area law: at small $\theta_{\mathrm{cap}}$, the entanglement entropy has a function form of $S_{\Sigma }∼\theta_{\mathrm{cap}}^{2}$ indicating that it scales *extensively* with increasing the cap size, i.e., it is *volume law* for small caps. The entropy continues up smoothly towards its maximum at $\theta_{\mathrm{cap}}=\frac{\pi }{2}$; because Ψ[Φ] is a pure state and is symmetric under the sphere isometries, $S_{\Sigma }{|}_{\theta_{\mathrm{cap}}}=S_{\Sigma }{|}_{\pi -\theta_{\mathrm{cap}}}$ and so must obtain a maximum there.

Unlike the holographic examples of [144,145], there is no transition to area law entanglement above a critical angle. This can be seen as a consequence of the compactness of the $S_{\nu }^{2}$: the putative scale of non-locality in which one would find a transition between volume law and area law entanglement is one the order of the sphere radius itself:(219)$$\ell_{\mathrm{crit}}=R\theta_{\mathrm{crit}}=\ell_{\mathrm{NC}}^{2}/\ell_{\mathrm{UV}}∼R\;.$$
Simply put, on the fuzzy sphere, there is not enough room to escape the effects of non-locality due to the parametric separation between $\ell_{\mathrm{NC}}$ and $\ell_{\mathrm{UV}}$. This intuition is bolstered by the indication that UV-finite measures of quantum correlations (such as the mutual information) behave similarly to their commutative counterparts [149]. This intuition was further bolstered in [150] by explicitly lowering the UV cutoff (i.e., increasing $\ell_{\mathrm{UV}}$) from $\ell_{\mathrm{UV}}=R/N$ to ${\tilde{\ell }}_{\mathrm{UV}}=R/n$ for n<N, which reduces the critical angle from order one to the following:(220)$${\tilde{\theta }}_{\mathrm{crit}}∼n/N\;.$$
As illustrated in Figure 16, the resulting ground state entanglement entropy then displays a smooth transition to an area law, $S_{\Sigma }\propto \sin\theta_{\mathrm{cap}}$, at a transition point that scales linearly with *n*. The necessity of taming the effects of non-locality by imposing a lower UV cutoff will be a theme that we will explore more broadly below.

### 7.2. Entanglement of Emergent Non-Commutative Spaces: Generalities

We will now present a general picture of entanglement entropy in non-commutative spaces described by matrices. In this section, we will not restrict ourselves to effective field theories on a fixed non-commutative background but allow ourselves to discuss the entanglement of fluctuations of the matrices which define the backgrounds themselves. Our treatment will follow closely [93]. Along the way, we will utilize much of the technology of gauge invariance and edge modes from Section 5.1, the structure of subregion algebras in target space from Section 6, as well as lessons about non-commutativity and cutoffs from our non-commutative scalar field in Section 7.1.

We will start by considering a non-commutative manifold $M_{\theta }$ parameterized by coordinates $\vec{x}$ with a non-commutative product parameterized by $\theta^{ij}$ as in (Equation 49). In general, we might allow $\left\{\vec{x}\right\}$ to coordinatize a non-commutative embedding space, e.g., $R_{\theta }^{D}$, and let $M_{\theta }$ to be an *d*-dimensional submanifold by a induced set of relations(221)$${\left\{h_{\left(r\right)}\left(\vec{x}\right)=0\right\}}_{r=1,\ldots ,D-d}\;.$$
A good example of this is the fuzzy sphere induced from $R_{\theta }^{3}$ from the relation $h\left(\vec{x}\right)=\sum_{i}{\left(x^{i}\right)}^{2}-R^{2}$. We will assume that $M_{\theta }$ is compact so that its algebra of non-commutative functions can be represented finitely by N×N matrices. We will be interested particularly in the limits that N≫1 is large, as well as the non-commutativity $\theta^{ij}$ is small so that we can work perturbatively both in 1/N and θ.

The non-commutative coordinates of $M_{\theta }$ are then represented by N×N Hermitian matrices, $\left\{X^{i}\right\}$ subject to the non-commutation relations, (Equation 49) and $\left\{h_{\left(r\right)}\left(X\right)=0\right\}$. We will allow these matrices to be dynamical, and prescribe for them a Lagrangian that takes the following form:(222)$$L=\frac{1}{N}\mathrm{Tr}\left(\sum_{i}{\left({\dot{X}}^{i}\right)}^{2}+\sum_{i<j}{\left[X^{i},X^{j}\right]}^{2}-V\left(X\right)\right)\;,$$
which is the general structure of MQM’s describing D-brane worldvolumes and arising from the dimensional reduction N=4 SYM, as reviewed in Section 3. We will further assume that the potential is such that the ground state is sharply peaked around the classical configuration $\left\{X_{\mathrm{cl}}^{i}\right\}$ coordinatizing $M_{\theta }$. For example, the fuzzy sphere configuration described in Section 7 and Section 7.1 minimizes the potential $V\left(X\right)=\frac{\nu^{2}}{4}\sum_{i=1}^{3}{\left(X^{i}\right)}^{2}$. Fluctuations about the classical configuration are approximately described by a Gaussian wavefunction(223)$$\Psi \left[X\right]=N\;\exp\left(-\frac{1}{2N}\mathrm{Tr}\sum_{i}{\left(X^{i}-X_{\mathrm{cl}}^{i}\right)}^{2}\right)\;,$$
up to normalization. Note that (stable) states of this type may or may not be present in a given theory (and indeed in the bosonic sector of the BMN model described in Section 3, the sphere vacuua are only metastable [153]); here we take (Equation 223) as part of what we mean by an ‘emergent semi-classical geometry.’

In the commutative limit, we can consider a subregion Σ of a manifold through its characteristic function, $\theta_{\Sigma }\left(\vec{x}\right)$ taking values one or zero depending on if $\vec{x}\in \Sigma$ or not, respectively; see (Equation 117). See Figure 17 for a cartoon.

This naturally leads to a projector described in Section 6.3, $\Theta_{\Sigma }$, acting on the matrix representatives, $\left\{X^{i}\right\}$, of the coordinates the associated non-commutative manifold, $M_{\theta }$, by promoting the characteristic function to a function of $X^{i}$, as in (Equation 162). As we saw above, this decomposes the matrices into four blocks:(224)$$X_{AB}^{i}:=\Theta_{A}\;X^{i}\;\Theta_{B}\;,\;A,B\in \left\{\Sigma ,\bar{\Sigma }\right\}\;.$$
Note that the momentum conjugate to $X_{\Sigma \Sigma }^{i}$ is not necessary the same as $\Theta_{\Sigma }\Pi^{i}\Theta_{\Sigma }$ for the simple fact that $\Theta_{\Sigma }$ is a function of *X* and, therefore, could have possible time-dependence. Indeed, a careful analysis of the kinetic term of (Equation 222) reveals that the momentum conjugate to $X_{\Sigma \Sigma }^{i}$, which we will denote (We caution the reader to not confuse this conjugate momentum with the Hilbert space projectors, (Equation 110), introduced in Section 5.1, which have been notated with a bolded symbol, Π.) as $\Pi_{\Sigma \Sigma }^{i}$:(225)$$\Pi_{\Sigma \Sigma }^{i}={\dot{X}}_{\Sigma \Sigma }^{i}-{\dot{\Theta }}_{\Sigma }\;X^{i}\;\Theta_{\Sigma }-\Theta_{\Sigma }\;X^{i}\;{\dot{\Theta }}_{\Sigma }\;.$$
At this point, let us pause to mention a couple of important facts about matrix projectors, $\Theta_{\Sigma }$, and their interpretation as ‘fuzzy’ analogs of characteristic functions, $\theta_{\Sigma }$. Both $\Theta_{\Sigma }$ and $\theta_{\Sigma }\left(x\right)$ square to themselves (for $\theta_{\Sigma }\left(x\right)$ this is true outside of a measure zero subset comprising the boundary of Σ). Moreover, the relationship between commutators and derivatives (Equation 52) then allows us to identify the objects $\left[X^{i},\Theta_{\Sigma }\right]$ as derivatives of non-commutative step functions, i.e., fuzzy versions of Dirac delta functions. In an explicit N×N matrix representation, $X^{i}$, the basis where we diagonalize $\Theta_{\Sigma }$ the quantity $\left[X^{i},\Theta_{\Sigma }\right]$ picks out the off-diagonal blocks, $X_{\Sigma \bar{\Sigma }}^{i}$. We, therefore, see that these off-diagonal blocks somehow encode delta functions that localize to the boundary of the subregion Σ, and, therefore, encode the geometric data of the boundary (see Section 2 of [96] for a more detailed review).

As we described in Section 6.4, the matrix projector defines a choice of tensor factorization of the extended Hilbert space (spanned by all matrix elements of $X_{ab}^{i}$) as follows:(226)$$H_{\Sigma }⊗H_{\bar{\Sigma }}\;,\;H_{\Sigma }:=H_{\Sigma \Sigma }\;,\;H_{\bar{\Sigma }}:=H_{\Sigma \bar{\Sigma }}⊗H_{\bar{\Sigma }\bar{\Sigma }}\;.$$
It is with respect to this factorization that we will compute our entanglement entropy, $S_{\Sigma }$. We will focus in the section to the factorization map defined by partially gauge-fixing U(N)→U(M)×U(N−M), where, as before, *M* is the rank of $\Theta_{\Sigma }$. We discuss the discuss the more general partial gauge-fixing in Section 8. As we discussed in Section 6.4, U(M) invariance enforces that states reduced on $H_{\Sigma }$ are maximally entangled over representations μ of U(M),(227)$${\tilde{\rho }}_{\Sigma }=\underset{\mu }{⨁}\;p_{\mu }\;\frac{1_{\mu }}{d_{\mu }}⊗{\hat{\rho }}_{\mu }\;,$$
and the entanglement entropy is given by the following: (228)$$S_{\Sigma }=\langle \log d_{\mu }\rangle -S_{\mathrm{Shannon}}\left[\left\{p_{\mu }\right\}\right]-\sum_{\mu }p_{\mu }\;S_{\Sigma }\left[{\hat{\rho }}_{\mu }\right]\;.$$
In what follows we will focus on the first term, the expectation value of the representation dimension, in semiclassical states about a configuration describing a non-commutative geometry, Equation (Equation 223). Representations of U(M) are given by Young diagrams, such as the one in Figure 18, and it is our task to find the conditions that this the expectation value is dominated by a single diagram and to compute its dimension. We will do so by considering the higher Casimirs of U(M), $\mathrm{Tr}\left({\hat{G}}_{\Sigma }^{2p}\right)$ for arbitrary *p*, acting on the reduced state. We will do so first as a representation theoretic quantities in the large *M* and *N* limits, and then secondly directly as expectation values in the semiclassical state, relating these representation theoretic quantities to geometric features of the non-commutative geometry.

We begin by writing a generic matrix configuration as follows:(229)$$X^{i}=X_{\mathrm{cl}}^{i}+\sum_{a}\delta x_{a}^{i}\;Y_{a}^{i}\;,\;\Pi^{i}=\sum_{a}\delta \pi_{a}\;Y_{a}^{i}\;,$$
where $Y_{a}^{i}$ are fixed basis of matrix normal modes of the quadratic action of (Equation 222) expanded about $X_{\mathrm{cl}}^{i}$ and normalized to the following: (230)$$\frac{1}{N}\sum_{i=1}^{D}\mathrm{Tr}\left(Y_{a}^{i\;†}Y_{b}^{i}\right)=\delta_{ab}\;.$$
About the classical background the generator of U(M) acting on $H_{\Sigma }$ can be written, to leading order in fluctuations:(231)$${\hat{G}}_{\Sigma }\approx 2i\sum_{i=1}^{D}\sum_{a}\delta \pi_{a}\left(Y_{a\;\Sigma \bar{\Sigma }}^{i}\;X_{\mathrm{cl}\;\bar{\Sigma }\Sigma }^{i}-X_{\mathrm{cl}\;\Sigma \bar{\Sigma }}^{i}Y_{a\;\bar{\Sigma }\Sigma }^{i}\right)\;.$$
Note that as $X_{\mathrm{cl}}^{i}$ and $Y_{a}^{i}$ are fixed matrices, the only quantum operator in ${\hat{G}}_{\Sigma }$ is $\delta \pi_{a}$ and so particular matrix elements of ${\hat{G}}_{\Sigma }$ are mutually (quantum) commuting and there are no ordering ambiguities in higher Casimir operators $\mathrm{Tr}\left({\hat{G}}_{\Sigma }^{2p}\right)$. A given representation μ will possess a highest-weight state, |μ;hw〉 whose Young tableau has each box labeled by its row number. This state is annihilated by all ${\left({\hat{G}}_{\Sigma }\right)}_{a,b}$ with a<b: (232)$$\langle \mu ;\mathrm{hw}|\mathrm{Tr}\left({\hat{G}}_{\Sigma }^{2p}\right)|\mu ;\mathrm{hw}\rangle =\sum_{r}\ell_{r}^{2p}\;,$$
where the sum is over rows in the Young diagram and $\ell_{r}$ is number of boxes contained in the row *r*. Going into calculation is the assumption that the off-diagonal blocks of the classical matrix background, $X_{\Sigma \bar{\Sigma }}^{i}$ are low-rank, i.e., there are not many independent off-diagonal matrix elements (see [93,98] for further explanation). Physically this is the statement that in our state of interest that strings stretching across ∂Σ remain close to Σ. This is an assumption both about the classical background—there exists a basis where all $X^{i}$ are “close to diagonal” and the effect non-commutativity parameter, θ, is small—and the choice of entangling cut—this is approximately the same basis diagonalizing $\Theta_{\Sigma }$. In these scenarios (Equation 232) should be regarded as the leading order result in a perturbation theory in both θ and 1/N.

What (Equation 232) tells us is that to leading order the spectrum ${\hat{G}}_{\Sigma }$ in a given representation block are the row lengths of it Young diagram:(233)$${\mathrm{spec}}_{\mu }\left({\hat{G}}_{\Sigma }\right)=\left\{\pm \ell_{1},\pm \ell_{2},\ldots ,\pm \ell_{M/2}\right\}\;.$$
These row lengths are in rough correspondence to the representation dimension, $d_{\mu }$, through the ‘hook formula’(234)$$\log_{\mu }=\sum_{r,c}\log\frac{M+c-r}{h_{r,c}}=2\sum_{r}\left[\log\left(\begin{array}{c} M+\ell_{r}-r\\ \ell_{r} \end{array}\right)-\sum_{c}\log\left(1+\frac{d_{r,c}}{1+\ell_{r}-c}\right)\right]\;,$$
where the ‘hook length,’ $h_{r,c}$, is defined as the number of boxes including, to the right of, and below the box at (r,c). In the second equality we have written this in terms of the depth, $d_{r,c}$ (the number of boxes below (r,c)) and massaged. There are two interesting limits of (Equation 234), depicted in Figure 19, that will be important in what follows. The first are ‘flat’ Young diagrams having a single row with $M\gg \ell_{1}\gg 1$; in this case we can approximate (Equation 234) as follows: (235)$$\log_{\mu_{\mathrm{flat}}}\approx 2\ell_{1}\log\frac{eM}{\ell_{1}}\;,$$
The second limiting case are ‘tall’ Young diagrams with an O(M/2) number of rows each with $M/2\gg \ell_{r}\gg 1$. We can approximately bound their dimension as follows: (236)$$M\ell_{★}≲\log_{\mu_{\mathrm{tall}}}\leq 2\sum_{r}\ell_{r}\log\frac{eM}{\ell_{r}}∼M\ell_{★}\log\frac{eM}{\ell_{★}}\;,$$
where $\ell_{★}$ is a typical row length of $\mu_{\mathrm{tall}}$.

Taking stock, we have shown that the spectrum of the Casimirs are Young diagram row lengths, and subsequently shown that these row lengths determine the representation dimension. The expectation value of these dimensions determines the entropy. What remains is to evaluate the expectation value of ${\hat{G}}_{\Sigma }^{2p}$ in the semi-classical state explicitly to find which representation(s) dominate the expectation value. We again make the assumption that the background can be treated perturbatively in both the non-commutativity parameter and in 1/N.

When the non-commutativity length scale is much smaller than the curvature length scale, $X_{\mathrm{cl}}^{i}$ is a low-rank matrix in the sense that only order 1 of min(M,N−M) singular values are nonzero. This allows us to express the second Casimir as follows: (237)$$\mathrm{Tr}\left({\hat{G}}_{\Sigma }^{2}\right)=8\sum_{i,j=1}^{D}\sum_{a}\delta \pi_{a}^{2}\mathrm{Tr}\left[\left(X_{\mathrm{cl},\Sigma \bar{\Sigma }}^{i}X_{\mathrm{cl},\bar{\Sigma }\Sigma }^{j}\right)\left(Y_{a,\Sigma \bar{\Sigma }}^{i}Y_{a,\bar{\Sigma }\Sigma }^{j}\right)\right]+O\left(1/N\right),$$
where we have only kept all of the orderings where the $X_{\mathrm{cl}}^{i}$ are grouped together after expanding out the square. The same can be done for $\mathrm{Tr}\left({\hat{G}}_{\Sigma }^{2p}\right)$ up to corrections of order O(p/N). The low-rank nature of $X_{\mathrm{cl},\Sigma \bar{\Sigma }}^{i}$ further allows us to remove the $\Sigma \bar{\Sigma }$ indices from *Y*, resulting in the following: (238)$$\mathrm{Tr}\left({\hat{G}}_{\Sigma }^{2}\right)=8\sum_{i,j=1}^{D}\sum_{a}\delta \pi_{a}^{2}\mathrm{Tr}\left[\left(X_{\mathrm{cl},\Sigma \bar{\Sigma }}^{i}X_{\mathrm{cl},\bar{\Sigma }\Sigma }^{j}\right)\left(Y_{a}^{i}Y_{a}^{j}\right)\right]+O\left(1/N\right)\;,$$
However, now the index a is restricted to modes $Y_{a}^{j}$ whose $\Sigma \bar{\Sigma }$ components are nonzero. Because the state (Equation 223) is Gaussian and the generator ${\hat{G}}_{\Sigma }$ is linear in the momentum $\Pi^{i}=\sum_{a}\delta \pi_{a}^{i}Y_{a}^{i}$, we can evaluate $\langle {\hat{G}}_{\Sigma }^{2p}\rangle$ through Wick contracting down to the momentum two point function:(239)$$K^{ij}:=\langle \Pi^{i}\Pi^{j}\rangle =\sum_{a}\omega_{a}Y_{a}^{i}\;Y_{a}^{j}+O\left(\theta ,N^{-1}\right)\;.$$
For the choice of sub-algebra implicit in the U(M) generator (Equation 231), if we let a run over all $3N^{2}$ modes, the resulting second Casimir will be a non-geometric quantity. This is a consquence of the UV/IR mixing that we have discussed above. Analogous to what we saw in the simple case of the non-commutative scalar in Section 7.1, geometric behavior in both the Casimirs and entanglement is restored if we introduce a UV cutoff (it is natural to take this cutoff to be a bound on the Laplacian of the normal mode $Y_{a}^{i}$, such as $\sum_{i,j}\mathrm{Tr}[Y_{a}^{i}\left[X^{j},\left[X^{j},Y_{a}^{i}\right]\right]]\leq \Lambda$) to cap off the fluctuations [81].

Often, for a given fuzzy geometry, the sum in (Equation 239) can be taken explicitly with the help of completeness relations of normal modes of the Laplacian. This is done explicitly for the fuzzy sphere, e.g., in Section 7.3 of [81]. For a cutoff Λ that is much smaller than the scale of non-commutativity, this results in (Equation 239) being dominated by the structure of $X_{\Sigma \bar{\Sigma }}^{i}$, which as discussed above are the non-commutative analogs of delta functions localized to the boundary of the region Σ. It follows that(240)$$\langle \mathrm{Tr}\left({\hat{G}}_{\Sigma }^{2p}\right)\rangle =\alpha N^{\frac{2p}{d}}\Lambda^{\frac{2p(d+1)}{d}}\sum_{k}{\left|\partial \Sigma_{\left(k\right)}\right|}^{2p},$$
where *d* is the dimension of the fuzzy geometry and the $\partial \Sigma_{\left(k\right)}$ are the topologically disconnected components of the boundary of Σ.

The implication is that *${\hat{G}}_{\Sigma }$ has an eigenvalue for each disconnected component of the entangling cut.* Given our representation theoretic discussion on the spectrum of ${\hat{G}}_{\Sigma }$ above, we can immediately identify row lengths of the dominant Young diagram with the area laws of individual components of ∂Σ.

For regions consisting of a single connected component and with sufficiently smooth entangling cuts (i.e., |∂Σ|≪M/N) then ${\hat{G}}_{\Sigma }$ has one singular value which is the row length of a flat Young diagram. This is enough to determine the dimension of the dominant representation via (Equation 235). This is almost enough to establish an area law for the entanglement entropy, however we also must show that $S_{\Sigma }$ in (Equation 228) is well approximated by restricting to a single dominant representation, which requires computing the variance in the Casimir in the semiclassical state. There is no first principles reason that this variance is low and the variance is specific to the model and state. However, when $S_{\Sigma }$ is well approximated by the dominant representation (and the cutoff Λ is in the regime where (Equation 235) applies) we then have the following:(241)$$S_{\Sigma }=\alpha N^{\frac{1}{d}}\Lambda^{\frac{d+1}{d}}\left|\partial \Sigma \right|\log\left(\frac{\left|\Sigma \right|}{\left|\partial \Sigma \right|}\right)+\ldots \;,$$
where we recall that the volume of the subregion goes like |Σ|∼M/N. This is an area law for the entanglement entropy with a logarithmic coefficient. The origin of this logarithm is the counting problem of large representations of U(M), e.g., (Equation 235). In the commutative limit we can see view this entanglement as a signal that the edge modes are charged under volume preserving diffeomorphisms. Given the discussion of volume preserving diffeomorphisms in Section 4 we thus expect a logarithm of this form to be generic in non-commutative geometries.

### 7.3. Example: The Non-Commutative Sphere, Part 2

We illustrate the above general story by revisiting the fuzzy sphere, now computing the Gauss law entropy of all the fluctuations about a fixed entangling subregion, Σ. We will take this subregion to be a cap with polar angle, $\theta_{\mathrm{cap}}$, as depicted on the left of Figure 14.

The classical background is that described in Section 4(242)$$\left[X_{\mathrm{cl}}^{i},X_{\mathrm{cl}}^{j}\right]=i\;\nu \epsilon^{ijk}\;X_{\mathrm{cl}}^{k}\;,$$
and is solved by taking $X_{\mathrm{cl}}^{i}=\nu J^{i}$ where $J^{i}$ is the *N*-dimensional irreducible representation of su(2). The semiclassical state expanded around this background as in (Equation 229) is given by(243)$$\psi \left(\delta x\right)=N\;\prod_{a}\exp\left(-\frac{\nu }{2}\left|\omega_{a}\right|\delta x_{a}\right)\;.$$
The cap region can be represented in this background in the basis where $X^{3}$ is diagonal and ordered: in this basis the eigenvalues are the (discrete) latitudes of the sphere and the projector $\Theta_{\Sigma }$ can be chosen to keep the top *M* of those latitudes:(244)$$\Theta_{\Sigma }=\sum_{a=0}^{M-1}|a)(a|\;.$$
The polar angle of the cap is then given as follows:(245)$$\cos\theta_{\mathrm{cap}}=1-\frac{2M}{N}\;,$$
and, therefore, the area (perimeter) and volume (area) of Σ are given as follows:(246)$$\left|\partial \Sigma \right|=2\pi \sin\theta_{\mathrm{cap}}=4\pi \frac{\sqrt{M(N-M)}}{N}\;,\;\left|\Sigma \right|=4\pi \frac{M}{N}\;,$$
respectively. These are expressed in units of the fuzzy sphere radius, $R=\frac{\nu N}{2}$.

The classical background background enjoys an SO(3) symmetry which organizes the matrix normal modes into matrix spherical harmonics [154,155], $Y_{jm}^{i}=\sum_{jm}y_{jm}^{i}\;{\hat{Y}}_{jm}$, satisfying the following:(247)$$\left[J^{3},{\hat{Y}}_{jm}\right]=m\;{\hat{Y}}_{jm}\;,\;\sum_{i=1}^{3}\left[J^{i},\left[J^{i},{\hat{Y}}_{jm}\right]\right]=j\left(j+1\right){\hat{Y}}_{jm}\;.$$
The expansion coeffecients, $y_{jm}^{i}$ and normal mode frequencies, $\omega_{jm}$, have been constructed explicitly in [80]. The matrix spherical harmonics have a large *N* form [81](248)$${\hat{Y}}_{jm}=2\sqrt{\pi }c_{jm}\sum_{k=1}^{N-m}Y_{jm}\left(\theta_{k},0\right)|k-1)(k+m-1|\;,\;N\gg |m|\;,$$
where $Y_{jm}$ (without the hat) are the standard spherical harmonics and(249)$$\cos\theta_{k}:=1-\frac{2k+m}{N}\;.$$
This makes the computation of the higher Casimirs, $\langle \mathrm{Tr}(G_{\Sigma }^{2p})\rangle$, particularly tractable in the large *N* limit. Given the discussion of the previous section, since ∂Σ contains only connected component, we expect ${\hat{G}}_{\Sigma }$ has only one singular value (which can also be established explicitly from the rank of the off-diagonal components of ${\left(X_{\mathrm{cl}}^{i}\right)}_{\Sigma \bar{\Sigma }}$; see [81]) and so it is sufficient to consider the quadratic Casimir: (250)$$\langle \mathrm{Tr}\left({\hat{G}}_{\Sigma }^{2}\right)\rangle \approx 4\pi \nu^{3}\frac{M(N-M)}{N}\sum_{j=1}^{N}2j\sum_{m=1}^{\min(M,j)}c_{jm}^{2}\;{\left|Y_{jm}\left(\theta_{M-m}\right)\right|}^{2}\;.$$
We pause at this point to highlight the peculiar cutoff on the m mode sum appearing in (Equation 250). For large j>M, these modes have a *volume dependent* cutoff which is a clear signal of a mixing of the UV spectrum to IR features of the subregion. We have already seen such a phenomenon occur for the scalar field on the fuzzy sphere in Section 7.1 which we traced back to a geometric uncertainty relation stemming from the non-commutative background. The UV/IR mixing here has a similar origin. Namely the fuzzy sphere possesses an uncertainty relation in the polar and azimuthal angles of the following form:(251)$$∆\left(\cos\theta \right)∆\phi \geq \frac{1}{N}\;.$$
This implies that a mode localized to $∆\phi ∼m^{-1}$ has a polar uncertainty of ∆(cosθ)∼∆M/N≳m/N. The modes are projected onto a matrix block of rank *M* and so ∆M≤M which establishes an upper bound on m and how tightly we can localize modes on the entangling cut. As alluded to in Section 7.1, we can recover geometric features in the entanglement entropy by imposing a lower UV cutoff to smooth the non-commutative UV/IR mixing. In particular, we can impose a cutoff of the following form:(252)j≤Λ≪M,N,
in the ground state wavefunction which leads to a geometric expression for the quadratic Casimir(253)$$\begin{array}{cc} {\langle \mathrm{Tr}\left({\hat{G}}_{\Sigma }^{2}\right)\rangle }_{\Lambda } & =\pi \nu^{2}N\;\sin^{2}\theta_{\mathrm{cap}}\sum_{j=1}^{\Lambda }\sum_{m=1}^{j}2j{\left|Y_{jm}\left(\theta_{\mathrm{cap}}\right)\right|}^{2}\\ \; & \approx \frac{\nu^{3}\Lambda^{3}\;N}{6}\sin^{2}\theta_{\mathrm{cap}}\\ \; & \approx \frac{\nu^{3}\Lambda^{3}\;N}{24\pi^{2}}{\left|\partial \Sigma \right|}^{2}\;. \end{array}$$
This determines the row length of the dominant representation. We can additionally compute the variance of the Casimir to find it scales as follows: (254)$$\begin{array}{cc} {\left({∆}_{c_{2}}^{\left(\Lambda \right)}\right)}^{2} & :={\langle \mathrm{Tr}{\left({\hat{G}}_{\Sigma }^{2}\right)}^{2}\rangle }_{\Lambda }-{\langle \mathrm{Tr}\left({\hat{G}}_{\Sigma }^{2}\right)\rangle }_{\Lambda }^{2}\\ \; & \approx 2\pi^{2}\nu^{6}N^{2}\sin^{4}\theta_{\mathrm{cap}}\sum_{jj^{\prime }}^{\Lambda }\left(4jj^{\prime }\right)\sum_{m=1}^{j}|Y_{jm}\left(\theta_{\mathrm{cap}}\right){|}^{2}{\left|Y_{j^{\prime }m}\left(\theta_{\mathrm{cap}}\right)\right|}^{2}\\ \; & ∼\nu^{6}\Lambda^{4}N^{2}\;. \end{array}$$
This is to be compared, via (Equation 253), to the square of the Casimir itself with scales as $\nu^{6}\Lambda^{6}N^{2}$. Thus, the variance of the quadratic Casimir is suppressed by the cutoff (the cutoff is actually not necessary to suppress the variation in the dominant representation. When the cutoff is removed the variance in the quadratic Casimir is(255)$$\frac{{∆}_{c_{2}}}{\langle \mathrm{Tr}\left({\hat{G}}_{\Sigma }^{2}\right)\rangle }∼N^{-1/2}\;.$$

)(256)$$\frac{{∆}_{c_{2}}^{\left(\Lambda \right)}}{{\langle \mathrm{Tr}\left({\hat{G}}_{\Sigma }^{2}\right)\rangle }_{\Lambda }}∼\Lambda^{-1}\;.$$
Thus, at large cutoff the entanglement entropy is dominated by a single representation with a flat Young diagram with row length given by the square root of (Equation 253). Putting this all together, we find an entanglement entropy of the following:(257)$$S_{\Sigma }=\sqrt{\frac{\nu^{3}\Lambda^{3}\;N}{12\pi^{2}}}\left|\partial \Sigma \right|\log\left(\sqrt{\frac{N}{\nu^{3}\Lambda^{3}}}\frac{\left|\Sigma \right|}{\left|\partial \Sigma \right|}\right)\;.$$
which is precisely of the form we described before, (Equation 241). We can put this in a more illuminating form by noting that the effective non-commutative Maxwell theory describing fluctuations about fuzzy sphere has a coupling constant:(258)$$g_{M}^{2}=\frac{4\pi }{N\nu^{3}}\;.$$
The entanglement entropy expressed in terms of this coupling(259)$$S_{\Sigma }=\sqrt{\frac{\Lambda^{3}}{3\pi }}\frac{\left|\partial \Sigma \right|}{g_{M}}\log\left(\frac{N\;g_{M}}{\Lambda^{3/2}}\frac{\left|\Sigma \right|}{\left|\partial \Sigma \right|}\right)\;,$$
has it appear inversely to the area law, which is highly suggestive of the appearance of the gravitational coupling in the Bekenstein-Hawking entropy or the Ryu–Takayanagi formula.

## 8. Minimal Areas from MQM Entanglement

Up to this point, we have considered entanglement in MQMs and non-commutative spaces with factorizations that ‘fix’ an entangling region in target space. In terms of the U(N) gauge redundancy of MQM, the factorizations we have dealt with have gauge-fixed the subset of gauge transformations that move the entangling surface. In the previous section we have argued, on general grounds, how the remaining gauge transformations preserving the entangling surface lead to non-commutative edge modes with have an area law entanglement in sufficiently nice semi-classical states.

In this section, we loosen the gauge fixing and look at the effect of invariance under maps that act on the location of the entangling surface. We will find that doing so will lead us to consider a gauge orbit’s worth of area laws. Under suitable conditions, that we will describe below, the integral over this orbit can admit a saddle-point approximation resulting in it being dominated by a subregion of minimal area. The result is the expression of the entanglement entropy as a minimal area, much akin to the Ryu–Takanagi formula for holographic entanglement entropy. However, in contrast to holographic entanglement entropy, the minimization is not over entangling regions anchored to any boundary system, but however over regions with fixed total volume:(260)$$S_{\Sigma }∼\min_{\begin{array}{c} \Sigma \;,\\ |\Sigma |\;\mathrm{fixed} \end{array}}\left|\partial \Sigma \right|\log\frac{\left|\Sigma \right|}{\left|\partial \Sigma \right|}\;.$$

Let us briefly describe how this result is reached. We begin by revisiting the factorization map described in Section 6.4. In that section we described how the physical Hilbert space of states could be embedded into an extended Hilbert space consisting of all Hermitian matrix elements an space invariant under some subgroup G⊆U(N). For the choice G=U(N), the subalgebra considered in this section is equivalent to [137]. A general choice of *G* may be viewed as an interpolation between [137] and the target space entanglement approach reviewed in Section 6. A physical state, |ψ〉, can be embedded as a G−invariant state $|\tilde{\psi }\rangle \in H_{\mathrm{ext}}$ by starting with a state with completely gauge-fixed matrix elements and averaging it over its *G*-orbit:(261)$$|\tilde{\psi }\rangle =\int_{G}G\;|G\rangle |{\tilde{\psi }}_{\mathrm{gf}}\rangle \;.$$
We wish to reduce this state based on the factorization of the extended Hilbert space determined by a fiducial M×M block of matrix degrees of freedom, (Equation 185). We saw that the U(M)×U(N−M) portion of the orbit integral leads a reduced density matrix that is maximally entangled over edge modes valued a representation μ of U(M) and living at a fixed entangling cut. The portion of this orbit integral over elements V∈G/(U(M)×U(N−M)) change the matrix block; they are so-called ‘wiggle modes,’ acting as a volume preserving diffeomorphisms that warp Σ to new a subregion with a distinct dominant representation, $\mu_{\left(V\right)}$ and corresponding area, $|\partial \Sigma_{\left(V\right)}|$. Overall, we will find that the density matrix reduced in the extended Hilbert space takes the following form: (262)$${\tilde{\rho }}_{\Sigma }∼⊕\;\int dV\left[\frac{1_{\mu_{V}}}{d_{\mu_{V}}}⊗{\hat{\rho }}_{\mu_{V}}\right].$$
The ‘⊕∫’ notation here indicates that reduced states at different V are (nearly) distinguishable. This assumption additionally leads to a Rényi entropy of the form(263)$$S_{\Sigma }^{\left(n\right)}=\frac{1}{1-n}\log\mathrm{tr}{\tilde{\rho }}_{\Sigma }^{n}∼\frac{1}{1-n}\log\int dV\;\exp\left(-\left(n-1\right)\log d_{\mu V}\right)\;,$$
where, as before, we have focussed on the contribution from the representation dimension. Holding, n>1 fixed, we can approximate this integral by saddle point to find(264)$$S_{\Sigma }^{\left(n\right)}∼\min_{V}\log d_{\mu V}∼\min_{V}\left|\partial \Sigma_{V}\right|\log\frac{|\Sigma_{V}|}{|\partial \Sigma_{V}|}\;,$$
where in the final step we have used the general connection between representation dimension and areas from Section 7.2. This is precisely the minimization formula for the entanglement entropy that we advertised, Equation (Equation 260).

There are several technial assumptions that have gone into this simple roadmap that we will make clear in this section. The first, and perhaps most important, is the existence of a saddle point in the integral (Equation 263). This assumption is, in fact, typically false. The reason is that, despite our abuse of terminology, most elements of U(N) are non-geometric when interpreted as a volume preserving ‘diffeomorphisms’: indeed a element exchanging a single matrix entry of $X^{i}$ exchanges a Planck-sized unit of area on the non-commutative background. These non-geometric elements of U(N) are so numerous that they wash out any saddle-point approximation to the integral.

In order to arrive at (Equation 260) it will be necessary to ‘coarse-grain’ over gauge transformations to eliminate such non-geometric maps while also preserving invariance under honest (volume preserving) diffeomorphisms in the continuum limit. Fortunately, this process only represents a mild extension of the technology we have established in Section 6.4 and Section 7.2. In particular, we will implement the coarse-graining by considering embeddings of physical states not as an U(N) invariant subspace of $H_{\mathrm{ext}}$, but instead a subspace invariant under an appropriate subgroup, G⊂U(N), with U(N)/G gauge-fixed. For reasons that will be clear below, we will sometimes refer to *G* as the ‘frame transformation group.’ We implement this embedding as before, by starting with a fully gauge-fixed state and averaging over its *G*-orbit, e.g., Equation (Equation 261). However, in the case that U(M)×U(N−M) is no longer a proper subgroup of *G*, we will want to implement U(M)×U(N−M) invariance separately and so our embedding into the extended Hilbert space is, in total,(265)$$\left|\psi \rangle \to \right|\tilde{\psi }\rangle =\int_{U(M)}dU\int_{U(N-M)}d\bar{U}\int_{G}dG\;|U\bar{U}G\rangle |{\tilde{\psi }}_{\mathrm{gf}}\rangle \;.$$
We can notate(266)$$H:=G\cap U\left(M\right)\;,\;\bar{H}:=G\cap U\left(N-M\right)\;,\;F:=\left(H\times \bar{H}\right)\setminus G$$
(here F is a left-quotient). We can subsume the $H\times \bar{H}$ portions of the *G* integral into the Haar integration of U(M)×U(N−M) to write(267)$$|\tilde{\psi }\rangle =\int_{F}dV|{\tilde{\psi }}_{V}\rangle \;,\;|{\tilde{\psi }}_{V}\rangle ={∆}_{V}\int dUd\bar{U}|U\bar{U}V\rangle |{\tilde{\psi }}_{\mathrm{gf}}\rangle \;,$$
and where ${∆}_{V}$ is a potential measure in splitting $\int_{G}G$ into integrals over $H\times \bar{H}$ and F. The state $|{\tilde{\psi }}_{V}\rangle$ is precisely the type of U(M)×U(N−M) invariant state that we considered in Section 7.2 that leads to edge modes with area law entanglement. Reducing (Equation 267) within $H_{\mathrm{ext}}$ with the factorization (Equation 185) leads to(268)$${\tilde{\rho }}_{\Sigma }=\int dVdV^{\prime }{\mathrm{tr}}_{H_{\bar{\Sigma }}}\left(|{\tilde{\psi }}_{V}\rangle \langle {\tilde{\psi }}_{V^{\prime }}|\right)\;.$$
In [98], it was argued that (Equation 268) is strongly supported at $V\approx V^{\prime }$, i.e., that it is roughly proportional to $\delta (V-V^{\prime })$, in semi-classical states of sufficiently tame MQMs. This leads to the block-integral decomposition of ${\tilde{\rho }}_{\Sigma }$ in (Equation 262) as well as replica symmetry and is responsible for the flat Rényi entropies (Equation 264). However, this is a priori an assumption whose breaking has interesting physical consequences; see [98] for elaboration.

Let us pause to discuss the physics of this replica symmetry assumption in the context of contrasting the U(M)×U(N−M) integral vs. the integral over coarse-grained frame transformations. It is useful to consider what physical information a low-energy observer with access to a subregion Σ can distinguish. The assumption of replica symmetry implies that the reduced state in the form (Equation 262) is nearly perfectly distinguishable between different V’s. This is because F acts non-unitarily on the reduced density matrix and changes its eigenvalues. Coarse-graining U(N)→G ensures that an observer with reduced state ${\tilde{\rho }}_{\Sigma }$ cannot distinguish between Planck-sized maps changing the subregion. Elements of U(M)×U(N−M) however have a different flavor: they act entirely within a subregion, preserving the information of the state (that is they act unitarily on reduced state at a give V). The integral over this subgroup ensures that the reduced state is invariant under such subregion preserving redundancies.

### 8.1. Relational Observables and Quantum Reference Frames

While the structure of the extended Hilbert space described in the previous section is sufficient for computing the entanglement entropy, it is physically illuminating to recast the previous discussion into the language of algebras of observables. We do this briefly in this section. Our primary perspective is to realize gauge-fixed data as the eigenvalues of a set of gauge-invariant observables relational to a *quantum reference frame* (QRF) [156,157,158,159,160,161]. Namely, the basis defining a matrix partition (say the basis diagonalizing $\Theta_{\Sigma }$) provides a reference frame to which we fix gauge-invariant data. The subgroup *G* rotates this QRF (thus we refer to it as the ‘frame transformation group’) and we allow ourselves to uncertain about the orientation of this QRF in integrating over *G* (which we refer to as the ‘frame average integral’).

To see how the dressing to a QRF works, consider a generic matrix configuration written of a gauge-fixed configuration, $X^{i}=UX_{\mathrm{gf}}^{i}\;U^{†}$, for some U∈U(N). The corresponding basis state, in the notation of (Equation 189) is(269)$$|X^{i}\rangle =|U\rangle |X_{\mathrm{gf}}^{i}\rangle \;.$$
We define the reference frame operator as(270)$$\hat{U}:=\int_{U(N)}U\;{\hat{P}}_{U}\;,\;\;{\hat{P}}_{U}:=|U\rangle \langle U|\;,$$
which acts on a typical basis state (Equation 269) returning its place on a gauge-orbit relative to the gauge-fixing, $X_{\mathrm{gf}}^{i}$:(271)$$\hat{U}|X^{i}\rangle =U\;|X^{i}\rangle \;.$$
We can instead pick out the gauge-fixed data by considering (in the terminology of the QRF literature, ${\hat{X}}^{i}$ is a kinematical operator its conjugation by $\hat{U}$ is a “*G*-twirl” operation establishing an incoherent group average [160])(272)$${\hat{X}}_{1}^{i}:=\left({\hat{U}}^{†}\;{\hat{X}}^{i}\;\hat{U}\right)\;,$$
which acts on (Equation 269) as(273)$${\hat{X}}_{1}^{i}|X^{i}\rangle =X_{\mathrm{gf}}^{i}|X^{i}\rangle \;.$$
It is easy to check that matrix elements of ${\hat{X}}_{1}^{i}$ commute with the generators, $\pi_{U}$, of U(N),(274)$${\hat{\pi }}_{U}{\hat{X}}_{1}^{i}=X_{1}^{i}\;{\hat{\pi }}_{U}\;,$$
and so are a gauge-invariant operators. This emphasizes that gauge-fixed data is in fact physical: they are the eigenvalues of a gauge-invariant operator. (We are ignoring possible subtleties involving coincident eigenvalues of a configuration which can lead to residual gauge redundancies. We will assume that the classical backgrounds do not have support on such configurations. See [98] for details and [162] for further discussion.) They are dressed to the QRF defined by $X_{\mathrm{gf}}^{i}$.

Similar we can define relation momentum operators(275)$${\hat{\Pi }}_{1}^{i}:=\int_{U(N)}dU\left(U^{†}\;{\hat{P}}_{U}{\hat{\Pi }}^{i}{\hat{P}}_{U}\;U\right)\;.$$
The projection operators enforce that ${\hat{\Pi }}_{1}^{i}$ is a vector field projected to lie along the gauge slice as opposed to along the gauge orbit.

#### 8.1.1. Incomplete Reference Frames

The relative operators (Equation 272) and (Equation 275) are defined in relation to a QRF whose orientation is fully specified. However, the physical situation we are interested doesn’t require specifying a reference frame for the interior or exterior of Σ, but only the location of its boundary, ∂Σ. i.e., we are interested in an ‘incomplete QRF,’ ref. [160]. We can define incomplete projectors, labelled by $V\in U\left(N\right)/\left(U(M)\times U(N-M)\right)$ as(276)$${\hat{P}}_{V}^{\left(M\right)}:=\int_{U(M)}dU\int_{U(N-M)}d\bar{U}\;{\hat{P}}_{VU\bar{U}}\;,$$
and an incomplete reference frame operator as(277)$$\hat{V}:=\int dV\;V{\hat{P}}_{V}^{\left(M\right)}=\int_{U(N)}dU\;V\;{\hat{P}}_{U}\;.$$
This allows us to construct incomplete relative operators in a wholly analogous fashion to before: (278)$${\hat{X}}^{i,(M)}:=\left({\hat{V}}^{†}\;{\hat{X}}^{i}\;\hat{V}\right)\;,\;{\hat{\Pi }}^{i,(M)}:=\int dV\left(V^{†}\;{\hat{P}}_{V}^{\left(M\right)}\;{\hat{\Pi }}^{i}\;{\hat{P}}_{V}^{\left(M\right)}\;V\right)\;,$$
which specify gauge-fixed data only up to an action of U(M)×U(N−M), i.e., on the state (Equation 269),(279)$${\hat{X}}^{i,(M)}|X^{i}\rangle =\left(U\bar{U}\;X_{\mathrm{gf}}^{i}\;U^{†}{\bar{U}}^{†}\right)|X^{i}\rangle \;.$$
These incomplete relative operators retain the U(M)×U(N−M) portion of the gauge orbit. Using them, we construct a gauge subregion algebra, $A_{\Sigma ,1}$ as(280)$$A_{\Sigma ,1}:=U\left\{\mathrm{Tr}F\left(\Theta_{\Sigma }{\hat{X}}^{i,(M)}\Theta_{\Sigma },\Theta_{\Sigma }{\hat{\Pi }}^{i,(M)}\Theta_{\Sigma }\right)\right\}\;,$$
where *F* is any polynomial. The ‘1’ subscript above indicates that these operators are defined relative to a gauge-fixing of U(N)/(U(M)×U(N−M)). This subregion algebra is equivalent to that constructed in [127]. Operators in $A_{\Sigma ,1}$ naturally commute with U(M)×U(N−M) and can be represented on an invariant subspace of $H_{\mathrm{ext}}$ annihilated by generators of U(M)×U(N−M). Thus, this subalgebra corresponds to the factorization map(281)$$\left|\psi \rangle \to \right|\tilde{\psi }\rangle =\int_{U(M)}dU\int_{U(N-M)}d\bar{U}\;|U\bar{U}\rangle |{\tilde{\psi }}_{\mathrm{gf}}\rangle \;,$$
leading to U(M) edge modes in the reduced density matrix as discussed in Section 6.4.1.

#### 8.1.2. Frame Averaging and Coarse-Grained QRFs

The algebra we described above is defined in relation to a U(M)×U(N−M) orbit of a gauge-fixed configuration, $X_{\mathrm{gf}}^{i}$, which we think of as an (incomplete) QRF. More generally we can construct an analogous subregion algebra by gauge-fixing to the QRF in a different orientation, $U^{†}\;X_{\mathrm{gf}}^{i}U$. Repeating the construction before leads to a distinct, yet isomorphic algebra of gauge-invariant M×M-subblock operators, $A_{\Sigma ,U}$, defined in relation to the rotated QRF $U^{†}X_{\mathrm{gf}}^{i}U$. The eigenvalues of the operators in $A_{\Sigma ,U}$ are just as physical as before, i.e., there is no canonical preference to which orientation of QRF we dress this physical data.

It is natural in this context to consider physical quantities that are averaged over the QRF orientation. This is fitting with notions of ‘bulk subsystems’ in holography and quantum gravity. In AdS/CFT diffeomorphism invariant subregions are defined by a boundary subregion, *A*, while its bulk ‘dual,’ (in the subregion-subregion duality sense that all operators in *a* can be reconstructed in *A* [163]) *a*, is determined dynamically. In the MQMs of question here leading to compact non-commutative manifolds, there are no boundaries to dress gauge-invariant notions of subregion. Instead the natural gauge-invariant specification is the total volume of Σ which is determined by the size of a matrix subbock. By averaging over frames we essentially are considering the entanglement of all subregions of fixed volume. The dominant subregion is determined dynamically as we described above and which we discuss in more detail below.

An additional subtlety that we have mentioned above is many rotated QRF orientations, $U^{†}X_{\mathrm{gf}}^{i}\;U$ do not have a clear geometric intepretation for the reason that many elements of U(N) do not act continuously or differentiably when interpreted as volume preserving maps. As a technical hurdle, the proliferation of these elements overwhelm the saddle-point leading to a minimal area formula. On an interpretational level, we might expect that coarse-graining over QRFs is needed for arriving at semi-classical quantities at low energies: the effective state of a low-energy observer should not be sensitive to Planck-sized uncertainties of their QRF. Averaging over a subgroup, G⊂U(N), is simply one method for implementing such a coarse-graining. We note that the fully ‘U(N) invariant’ projector Θ is a very similar partition of the degrees of freedom as considered in [137].

As of now, we do not have a satisfactory method for implementing a coarse-grained frame average directly at the level of the subregion algebra. (Although the formalism of non-ideal QRFs [156,160,164] may prove helpful in this direction.) Instead we must operationally proceed from the extended Hilbert space with the factorization map (Equation 265).

### 8.2. Structure of the Frame Average Integral

We have so far outlined that the entanglement question of “What is the entanglement of any subregion of fixed volume?” in non-commutative geometries arising from MQMs leads to a factorization map involving a (coarse-grained) frame average, Equation (Equation 265). The further assumption of replica symmetry then leads to *n*-purities of the form (we have swept a factor of δ(0) under the rug in this formula which adds an addtive logδ(0) to the entropy. Such factors arise generically in continuum limits of probability distributions. It can be regulated as $\delta \left(0\right)∼∆V^{-1}$ where ∆V is the uncertainty of V in the state. We are assuming this contribution is subleading at large *N* [98])(282)$$\mathrm{tr}{\tilde{\rho }}_{\Sigma }^{n}\approx N^{n}\int_{F}dVd_{\mu_{V}}^{1-n}\mathrm{tr}{\hat{\rho }}_{\mu_{V}}\approx N^{n}\;\int_{F}dV\;e^{-I_{n}\left(V\right)}\;,\;I_{n}\left(V\right)=\left(n-1\right)\log d_{\mu_{V}}\;,$$
where $\mu_{V}$ is the dominant U(M) representation at a given V and we have assumed that its dimension gives the leading contribution at large *N*. When this frame average integral admits a saddle-point approximation, the saddle-point configuration leads to a minimal area formula for the entropy, Equation (Equation 260). Below we highlight the structure of this integral, as well as the conditions for its saddle-point approximation to be valid. This could fail because the width of saddle-point can be very small and the number of fluctuations in F can be very many. It is useful to model and approximate these fluctuations as Haar random matrices and express (this is working with a measure V such that volF=1)(283)$$Z_{n}:=\mathrm{tr}{\tilde{\rho }}_{\Sigma }^{n}\approx Z_{1\text{-}\mathrm{loop}}\;e^{-I_{n,\mathrm{saddle}}}+e^{-{\bar{I}}_{n}}\;.$$
See Figure 20 for a cartoon of this structure. The overline indicates the generic value of the integrand obtained averaging $I_{n}$ over F. The ability to move this average into the exponent requires that the variance of $I_{n}$ is small compared to ${\bar{I}}_{n}$.

Trusting the saddle-point approximation then requires(284)$$I_{n,\mathrm{saddle}}-\log Z_{1\text{-}\mathrm{loop}}\ll {\bar{I}}_{n}\;,$$
i.e., that, despite the small width, $I_{n,\mathrm{saddle}}$ remains truly a saddle, as well as(285)$$I_{n,\mathrm{saddle}}\gg -\log Z_{1\text{-}\mathrm{loop}}$$
such that integral is well approximated by the saddle-point value (as opposed to the one-loop term).

#### 8.2.1. The Saddle-Point

The saddle-point of the integral comes from the frame orientation where ∂Σ has minimal area. Without loss of generality, we can choose this frame to define the basis diagonalizing $\Theta_{\Sigma }$ and so occurs at V=1. That this is a saddle-point follows from our discussion in Section 4 that the integral over F is equivalent to an integral over volume preserving diffeomorphisms and the observation in Section 7.2 that the entropy of a region is proportional to its area.

A more rigorous treatment of this intuitive fact is given in [98] by studying the perturbations to the singular values of ${\hat{G}}_{\Sigma }$ about flat Young diagrams (i.e., those corresponding to simply connected Σ) under two types of perturbations (i) ones perturbing the singular value, i.e., perturbing the region while keeping it connected, and (ii) ones introducing new singular values, i.e., fragmenting Σ into multiple regions. A cartoon of these perturbations is depicted in Figure 21.

There it was argued that perturbations introducing new singular values ($H_{2}$ in the figure) always increase $\log_{\mu }$. Perturbations to the sole singular values ($H_{1}$ in the figure) that maintain weakly curved ∂Σ, i.e., their normal mode expansion remains much less than the cutoff, can be dealt with analytically and lead to a saddle-point equation(286)$$[\Theta_{\Sigma },K_{ij}\left[X_{\mathrm{cl}}^{i},\left[X_{\mathrm{cl}}^{j},\Theta_{\Sigma }\right]\right]]=0\;,$$
(where we recall $K^{ij}=\langle \Pi^{i}\Pi^{j}\rangle$) so minimizes the generalized area functional. Establishing that normal modes of $H_{1}$ that are on the order of, or above, the wavefunction cutoff increase $\log d_{\mu }$ falls outside the regime of tractable analytic calculation; instead this must be corroborated numerically. We will see an example shortly below.

#### 8.2.2. The One-Loop Term

At a heuristic level, $Z_{1\text{-}\mathrm{loop}}$ is roughly the inverse square root of volume under the peak at the saddle-point (normalized such that volF=1), which follows from the behavior of Gaussian integrals. In truth the integral about the saddle-point is not quite Gaussian due the non-analytic nature of perturbations that create new connected components of Σ, however at the level of estimation this statement is roughly correct. This is estimated then as to the number of independent directions that move one away from the saddle-point, up to a logarithmic factor:(287)$$\log Z_{1\text{-}\mathrm{loop}}∼-\frac{1}{2}\left(\#\;\mathrm{of}\;\mathrm{perturbations}\right)\times \log\left(\mathrm{width}\;\mathrm{of}\;\mathrm{typical}\;\mathrm{perturbation}\right)\;.$$
The precise value of $\log Z_{1\text{-}\mathrm{loop}}$ depends sensitively on the classical state and the nature of coarse-graining defining F, Equation (Equation 266). However, we can get a crude idea of the order of magnitude of $\log Z_{1\text{-}\mathrm{loop}}$ by first estimating its contribution when there is no coarse graining, i.e., F=(U(M)×U(N−M))∖U(N). Writing $V=e^{iH}$, the Gaussian integral over *H* is(288)$$\mathrm{no}\;\mathrm{coarse}\text{-}\mathrm{graining}:\;\log\int_{F}dH\;e^{-\alpha H^{2}}∼-\frac{M(N-M)}{2}\log N\;,$$
which is the large-*M*, large *N* scaling of $\frac{1}{2}\log{\mathrm{vol}}_{H}\;F$: the volume in the measure inherited from Hermitian matrices, normalized by the Haar measure on F [165,166].

In implementing the coarse-graining U(N)→G, it is useful to viewing this as eliminating some portion of generators of F (e.g., those generating Planck-scale diffeomorphisms). This has the effect of modifying the number of perturbations in (Equation 287) from M(N−M) to γM(N−M) for some γ<1. In principle, it is also possible to modify the width of a typical perturbation although it is difficult to argue that it deviates from an O(N) quantity. (For instance, although the semi-classical wavefunction might involve a UV/IR mixing cutoff, Λ, there is no a priori reason this cutoff regulates the frame average integral.) Thus, our expectation is that(289)$$\log Z_{1\text{-}\mathrm{loop}}≲-\gamma \frac{M(N-M)}{2}\log N\;,\;\;\gamma \leq 1\;,$$
in the large-M,N limit. We will see this scaling explicitly in an example below.

#### 8.2.3. The Generic Term

To estimate the generic term, we calculate the average(290)$$\bar{I_{n}}:=-\bar{\log\mathrm{tr}\rho_{\Sigma ,V}}\approx \left(n-1\right)\bar{\log d_{\mu V}}\;,$$
where the overline indicates the Haar average integral over F. In practice, due to the invariance of the reduced density matrix under $H\times \bar{H}$, we can extend this to a Haar average over *G* with the normalized Haar measure. This is particularly useful when *G* is itself a unitary group due to the well developed technology of unitary integrals. Much in the same vein as Section 7.2, we evaluate the representation dimension by considering the average of higher Casimir operators: (291)$$C_{2s}:=\mathrm{Tr}{\hat{G}}_{\Sigma }^{2s}\;.$$
Under suitable conditions one finds that ${\bar{I}}_{n}$ is given by the dimension of a ‘typical’ representation of *H*:(292)$$\bar{I_{n}}\approx \left(n-1\right)\log d_{\mu_{★}}\;,$$
where $d_{\mu_{★}}$ can be calculated from $\bar{\langle C_{2s}\rangle }$. In order to establish this, it is necessary to first establish the following:The Casimir is self-averaging:(293)$$\frac{\bar{{\langle C_{2s}\rangle }^{2}}-{\bar{\langle C_{2s}\rangle }}^{2}}{{\bar{\langle C_{2s}\rangle }}^{2}}\ll 1\;.$$This establishes that quantum distribution of irreps have the same average over F.The variance of Casimir ${∆}_{C_{2s}}^{2}$ is self-averaging:(294)$$\frac{\bar{{\left({∆}_{C_{2s}}^{2}\right)}^{2}}-{\left(\bar{{∆}_{C_{2s}}^{2}}\right)}^{2}}{{\left(\bar{{∆}_{C_{2s}}^{2}}\right)}^{2}}\ll 1\;.$$This establishes that the quantum distribution of irreps have the same width. This and the above bullet also justify treating $\bar{\exp(-I_{n})}$ as $\exp(-{\bar{I}}_{n})$.The variance is small:(295)$$\frac{\bar{{∆}_{C_{2s}}^{2}}}{{\bar{\langle C_{2s}\rangle }}^{2}}\ll 1\;.$$This establishes that the generic quantum state is dominated by a single irrep, $\mu_{★}$.

As mentioned before, when *G* is itself a unitary group, one can use the structure of unitary integrals to calculate the Haar averages, at least at large *N*. While the details of such calculations can be lengthy and sensitive to the state and the coarse-graining, *the* value of ${\bar{I}}_{n}$*is actually insensitive to the coarse-graining* (it is still sensitive to the details of the state, e.g., the cutoff on UV/IR mixing, Λ, or any couplings), at least at large *N*. This is because the conjugation of an N×N Hermitian matrix by a typical element of G⊂U(N), even when *G* is ‘small,’ will generate O(N) eigenvalues of magnitude O(N).

Thus, of the three terms, $I_{n,\mathrm{saddle}}$, $\log Z_{1-\mathrm{loop}}$, and ${\bar{I}}_{n}$, the details of the coarse-graining are contained in the width of the saddle-point, Equation (Equation 289). The conditions for saddle-point dominance, Equations (Equation 284) and (Equation 285), then provide criteria for when a coarse-grained frame average is sufficient to ensure $S_{\Sigma }$ is determined by a minimal area.

### 8.3. Example: The Non-Commutative Sphere, Part 3

To illustrate the above principles in a tractable model we will return to the ‘fuzzy sphere’ of Section 7.1 and Section 7.3. In particular, in Section 7.3 we considered the Gauss law entanglement for a fixed partition on corresponding to a ‘cap’ on the Northern hemisphere of the sphere with surface ‘area’ (perimeter) and ‘volume’ (area)(296)$$|\partial \Sigma_{\mathrm{cap}}|=4\pi \frac{\sqrt{M(N-M)}}{N}\;,\;\left|\Sigma_{\mathrm{cap}}\right|=4\pi \frac{M}{N}\;.$$
We will now introduce a (coarse-grained) frame average into this story and show how this subregion, and its area-law, can appear as a saddle-point. In order to do so, we need to say how we plan to coarse-grain the frame transformation group, *G*. We will follow the low-tech yet illustrative example in [98] with(297)$$G=U\left(N^{\prime }\right)⊗1_{p}\subset U\left(N\right)\;,\;N^{\prime }:=N/p\;,\;N\gg p\;,$$
with *G* embedded into U(N) as p×p blocks in the basis diagonalizing $\Theta_{\Sigma_{\mathrm{cap}}}$ such that(298)$$H=U\left(M^{\prime }\right)⊗1_{p}\;,\;\bar{H}=U\left(N^{\prime }-M^{\prime }\right)⊗1_{p}\;,\;M^{\prime }:=M/p\;.$$
We choose an integer *p* such that both $N^{\prime }$ and $M^{\prime }$ are integers. See [167,168] for further interpretations of this as a coarse-graining. This choice of coarse-graining will allow us to utilize the technology of unitary Haar integrals advertised above. Moreover we will find a parametric range of *p* such that the saddle-point dominance conditions, Equations (Equation 284) and (Equation 285), are satisfied and $S_{\Sigma }$ is dominated by the cap configuration.

The cap subregion, $\Sigma_{\mathrm{cap}}$, with a circular boundary is the minimal area configuration with |Σ| fixed, and provides a candidate saddle-point to the frame average integral. To establish this more rigorously, we can consider perturbations to the saddle-point value under $V=e^{i\epsilon H}$ for small ϵ. As discussed above, these perturbations come in two forms, those that perturb the sole singular value (i.e., preserve the connectivity of Σ) and those that introduce new singular values (i.e., fragment Σ), as depicted in Figure 21. Given the spherical background of $X_{\mathrm{cl}}^{i}$, we can also segregate perturbations bases upon their mode expansion in matrix spherical harmonics. Weakly curved perturbations, i.e., those with modes m≪Λ, of the former type, $H_{1}$, can be modelled as continuum volume preserving diffeomorphisms under which a circular ∂Σ is clearly a stable minimum (see, e.g., the analysis in [98]). For large perturbations (of either type) with m≳Λ one can numerically corroborate the stability of the saddle, as depicted in Figure 22. Thus, the cap entropy, Equation (Equation 257) provides the value of the saddle-point to frame average integral:(299)$$I_{n,\mathrm{saddle}}=\left(n-1\right)S_{\Sigma_{\mathrm{cap}}}=\frac{n-1}{\sqrt{3}}{\left(\nu \Lambda \right)}^{3/2}\sqrt{\frac{M(N-M)}{N}}\log\left(\frac{4\pi MN}{{\left(\nu \Lambda \right)}^{3}\left(N-M\right)}\right)\;.$$

To estimate the one-loop contribution to (Equation 283) we consider the fluctuations to quadaratic Casimir,(300)$$\langle \mathrm{Tr}{\hat{G}}_{\Sigma }^{2}\rangle ∼X_{\mathrm{cl},\bar{\Sigma }\Sigma }^{i}X_{\mathrm{cl},\Sigma \bar{\Sigma }}^{j}\langle \Pi_{\bar{\Sigma }\Sigma }^{j}\Pi_{\Sigma \bar{\Sigma }}^{i}\rangle \;,$$
under(301)$$\begin{array}{cc} X_{\mathrm{cl}}^{i} & \to X_{\mathrm{cl}}^{i}+i\epsilon \left[H,X_{\mathrm{cl}}^{i}\right]-i\epsilon^{2}\left[H,\left[H,X_{\mathrm{cl}}^{i}\right]\right]+\ldots \\ \Pi^{i} & \to \Pi^{i}+i\epsilon \left[H,\Pi^{i}\right]-i\epsilon^{2}\left[H,\left[H,\Pi^{i}\right]\right]+\ldots \end{array}$$
estimating orders of magnitude at large *N*. Using the fact that $X_{\mathrm{cl},\Sigma \bar{\Sigma }}^{i}$ is a rank one matrix [81] one finds that the perturbation of the initial singular value is of the order(302)$$\delta_{H}\ell_{1}∼{\left(\nu \Lambda \right)}^{5/2}\;N^{3/2}\epsilon^{2}\;H^{2}\;,$$
while perturbations creating new singluar values are of the order(303)$$\delta_{H}\ell_{r\neq 1}∼\left\{\begin{array}{cc} {\left(\nu \Lambda \right)}^{3/2}N^{1/2}\left|\epsilon \right|\sqrt{H^{2}} & 1<r\leq \Lambda \\ \nu^{3/2}\Lambda^{5/2}N^{1/2}\epsilon^{2}\sqrt{H^{4}} & \Lambda <r\leq \frac{M}{2} \end{array}\right.\;.$$

Note the non-analyticity in *H* stemming from the creation new singluar values in *H*; this is a mirror of their discontinuous nature as volume preserving diffeomorphisms creating new connected components. These estimates upper bound the one-loop integral as(304)$$Z_{1\text{-}\mathrm{loop}}≲\frac{1}{{\mathrm{vol}}_{H}F}\int He^{-N^{1/2}\left(M/2-\Lambda \right)\sqrt{H^{4}}}∼e^{-\frac{M(N-M)}{2p^{2}}\log N}\;,$$
where we have dropped possible factors of ν and Λ contributing to the logarithm, keeping only the large *N* scaling. This is precisely the scaling we argued for in (Equation 289) with $\gamma =p^{-2}$.

Lastly we can estimate the generic term in the fuzzy sphere configuration. As advertised above, due to the relatively simple nature of the coarse graining, $G=U(N^{\prime })$, we can utilize the technology of unitary Haar integrals which simplify in the large-$N^{\prime }$ limit. More specifically we evaluate quantities of the form(305)$$\bar{F}=\int_{G}G\;F\left(G\right)\;,$$
through Wick contraction [169], contracting all pairs of $G^{†}$ and G, with a power of $1/N^{\prime }$ for each contraction, e.g., the following: (306)$$\begin{array}{cc} \bar{G_{ab}^{†}G_{cd}G_{ef}^{†}G_{gh}} & =\overset{⎴}{G_{ab}^{†}G_{cd}}\;\overset{⎴}{G_{ef}^{†}G_{gh}}+\overset{⎴}{G_{ab}^{†}\overset{⎴}{G_{cd}G_{ef}^{†}}G_{gh}}+\ldots \\ \; & =\frac{\delta_{ad}\delta_{cb}\delta_{eh}\delta_{fg}+\delta_{ah}\delta_{bg}\delta_{cf}\delta_{ed}}{N^{\prime }d^{2}}+O\left(N^{\prime }d^{-3}\right)\;. \end{array}$$
This simplifies the calculations of the higher Casimirs considerably. Moreover, using the low-rank-ness of ${\left(X_{\mathrm{cl}}^{i}\right)}_{\Sigma \bar{\Sigma }}$ as well as the scaling of a typical matrix element of ${\left(X_{\mathrm{cl}}^{i}\right)}_{ab}∼N$, one can estimate the orders of magnitude in $N^{\prime }$ scaling of the various quantities appearing in Section 8.2. While systematic, these calculations can still be lengthy and encourage the reader to see [98] for details, only reporting the results here. In particular, one finds that at large $N^{\prime }$ and large cutoff, Λ, the conditions for the generic term to be dominated by a single, typical, representation, $\mu_{★}$ are satisfied, i.e., the distribution of irreps is uniform across F and this distribution is sharply peaked at a single irrep:(307)$$\frac{\bar{{\langle C_{2}\rangle }^{2}}-{\bar{\langle C_{2}\rangle }}^{2}}{{\bar{\langle C_{2}\rangle }}^{2}}∼\frac{\bar{{\left({∆}_{C_{2}}^{2}\right)}^{2}}-{\left(\bar{{∆}_{C_{2}}^{2}}\right)}^{2}}{{\left(\bar{{∆}_{C_{2}}^{2}}\right)}^{2}}∼\frac{1}{N^{\prime }d^{2}}\ll 1\;,\;\frac{\bar{{∆}_{C_{2}}^{2}}}{{\bar{\langle C_{2}\rangle }}^{2}}∼\frac{1}{\Lambda^{2}}\ll 1\;.$$
A calculation of the higher Casimirs reveals(308)$$\bar{\langle C_{2s}\rangle }∼{\left(\nu \Lambda \right)}^{3s}M{\left(\frac{M(N-M)}{N}\right)}^{s}\;,$$
and so ${\hat{G}}_{\Sigma }$ has O(M) singular values with a typical magnitude of $\ell_{★}∼{\left(\nu \Lambda \right)}^{3/2}\sqrt{\frac{M(N-M)}{2}}$. In our classification from Section 7.2, this representation has a ‘tall’ Young diagram depicted on the right in Figure 19 and with dimension given by (Equation 236). Thus, the generic term is estimated as(309)$${\bar{I}}_{n}\approx \left(n-1\right)\log d_{\mu_{★}}≳\left(n-1\right){\left(\nu \Lambda \right)}^{3/2}M\sqrt{\frac{M(N-M)}{N}}\;.$$
As we alluded to earlier, despite being averaged over $U(N^{\prime })$, this term is independent of the coarse-graining parameter, *p*. We have now assembled all the components necessary to describe the conditions on when the coarse-grained frame average admits a saddle-point. Indeed, a quick comparison of (Equation 299), (Equation 304), and (Equation 309) reveals that (Equation 284) and (Equation 285) are satisfied if(310)$$N^{3/4}\ll p\ll N\;,$$
(with the upper limit such that $N^{\prime }\to \infty$ in the large *N* limit). Thus, we find a parametric family of coarse-grainings such that the *n*-purities are well approximated by saddle-point(311)$$Z_{n}\approx e^{-I_{n,\mathrm{saddle}}}\;,$$
and the entanglement entropy in the n→1 limit is given by the region with minimal area:(312)$$S_{\Sigma }=\frac{1}{\sqrt{3\pi }}\frac{\Lambda^{3/2}}{g_{M}}\min_{\begin{array}{c} \Sigma \;,\\ |\Sigma |\;\mathrm{fixed} \end{array}}\left[|\partial \Sigma |\log\left(\frac{g_{M}N}{\Lambda^{3/2}}\frac{\left|\Sigma \right|}{\left|\partial \Sigma \right|}\right)\right]\;.$$

## 9. Conclusions

In this article, we reviewed a growing body of research on how geometric features arise from the entanglement of quantum mechanical matrices at large *N*. We motivated this study in Section 2 through a brief history of the connections between the ’t Hooft limit of MQMs and string theory. In Section 3 we summarized several standard MQM models that play a key role in string theory and M-theory. The relevance of MQMs to D-brane physics then motivated a discussion in Section 4 on non-commutative geometries and their matrix representations. Central to this discussion was the realization of the U(N) symmetry of MQMs as volume preserving diffeomorphisms (and symplectomorphisms, more generally) acting on non-commutative spaces in the large *N* limit. In Section 5, we built the necessary scaffolding for investigating entanglement in MQM by overviewing how subsystems and entanglement entropies are defined in the presence of gauge redundancy. In Section 6, we discussed the target space interpretation of this entanglement, starting with a review of the target space entanglement of identical particles and building up to a notion of entanglement in MQM based upon a gauge-fixed subsets of matrix degrees of freedom. We then illustrated, in Section 7, how target space entanglement can be realized geometrically on a non-commutative space, starting from a simple non-commutative scalar field theory and then presenting general criteria for when the entanglement entropy can take on an ‘area law’ form in an emergent geometry. Lastly, in Section 8, we drew connections between this area law and entanglement entropy in quantum gravity, and in particular the celebrated Ryu–Takayanagi formula for holographic entanglement entropy, by describing the conditions in which U(N) invariance leads to the expression of the entanglement entropy as a minimal area formula.

The history of MQM in string theory and holography is rich, and the study of entanglement in these areas has proven to be extremely fruitful. As such there are many topics that we unfortunately unable to cover in this review. While technically not a model of quantum mechanics, notably absent from our review in Section 3 is a discussion of the matrix integral of Ishibashi, Kawai, Kitazawa, and Tsuchiya [16] which has a strong relevance in string theory and D-brane physics and has recently seen a small renaissance in its holographic understanding [170,171,172,173]. Similarly, as we noted in beginning of Section 6 on target space entanglement, we have chosen to omit a discussion on target space entanglement in string theory (e.g., [33,117,118,119,120,121,122,123,124]) although the topic is clearly relevant to various aspects of this review and important to understanding entropy in quantum gravity more broadly.

In writing this review, it was very clear to us that the topic of entanglement entropy in MQM is far from a closed and fully understood subject. Despite the progress highlighted in this review, deeper connections between matrix and target space entanglement to AdS/CFT and D-brane holography more broadly, how a genuine Ryu–Takayanagi formula emerges, and the use of entanglement in probing the internal spaces of AdS/CFT are still being explored [84,137,174,175,176,177]. More mysterious is how target space entanglement is expressed in the flat limits of various matrix models. This reflects both the technical difficulties of working with the nine matrices of the BFSS model as well as the broader mystery of how flat space physics and Lorentz invariance manifests itself in the BFSS model beyond scattering amplitudes [68]. Further, it would be interesting to understand how the target space entanglement discussed in this review might apply to black hole states in MQM, which are conjectured to be thermal excitations of sub-blocks [42,72,73,74], and perhaps recover an entanglement of $\frac{A}{4G_{N}}$. We hope that many of these questions will be better understood in the near future.

## Figures and Tables

**Figure 1 entropy-28-00058-f001:**
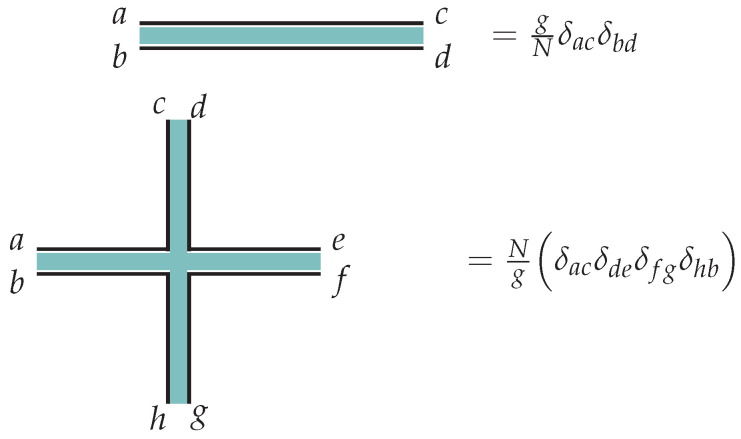
The Feynman rules for the matrix integral (Equation 7) in the ’t Hooft double line notation.

**Figure 2 entropy-28-00058-f002:**
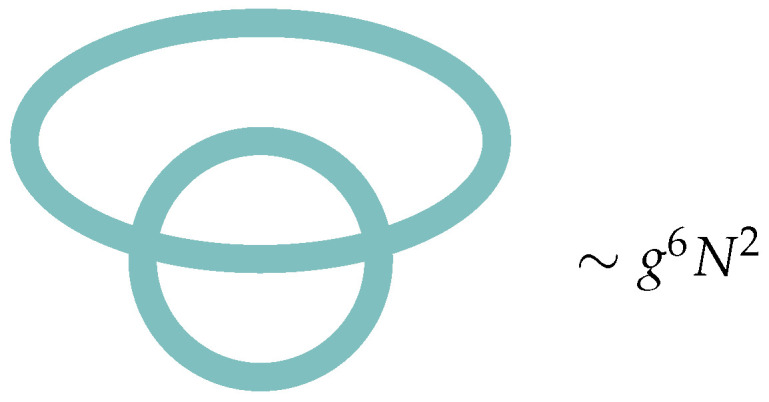
A planar Feynman diagram of the matrix integral (Equation 7) with $N_{E}=8$, $N_{V}=2$ and $N_{F}=4$. We have suppressed the bold lines carrying the matrix indices at the edges of the teal ribbon.

**Figure 3 entropy-28-00058-f003:**
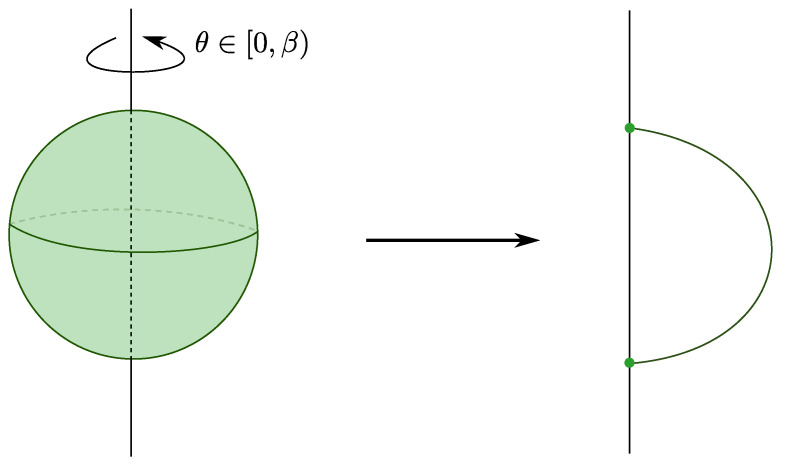
(**Left**): A string worldsheet is punctured by a conical deficit in spacetime. These configurations are those that dominate the contribution to the black hole entropy of $\frac{A}{4G_{N}}$, as found in [25]. (**Right**): When sliced along θ, Euclidean time, these diagrams become open strings anchored to the black hole event horizon.

**Figure 4 entropy-28-00058-f004:**
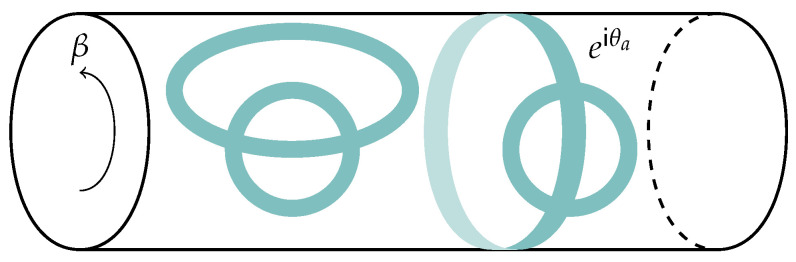
A cartoon of ‘t Hooft diagrams that can occur on a thermal circle. The compact direction is Euclidean time, the non-compact direction represents the target space of the matrices. The diagram on the left can occur with a non-compact base space direction as well. The diagram on the right includes propagators that wrap around the thermal circle. In a gauged model, where the periodic boundary conditions are $X\left(\beta \right)=UX\left(0\right)U^{†}$ for a unitary U that must be integrated over, propagators that wrap the circle are dressed by the phase $e^{i\theta_{a}}$, the eigenvalue of U. In an ungauged model, no such phases appear.

**Figure 5 entropy-28-00058-f005:**
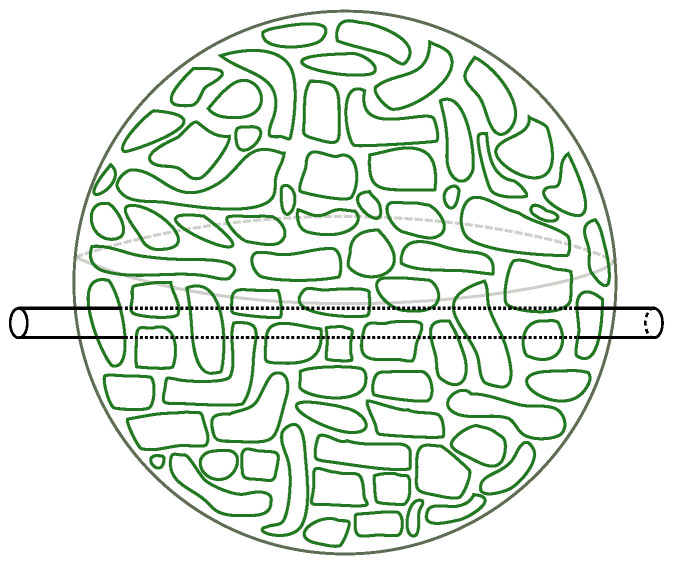
A diagram of the same type as on the right hand side of Figure 4, but in the regime of large temperature (small radius of thermal circle) and the double scaling limit. In this regime, there is little cost in free energy for the thermal circle to ‘puncture’ the sphere diagram, as depicted in the figure. As shown in [40], and further in [42], it is precisely these diagrams that are responsible for computing the black hole entropy in the c=1 matrix model. We, therefore, find that this figure bears qualitative and quantitative similarity to the left hand side of Figure 3.

**Figure 6 entropy-28-00058-f006:**
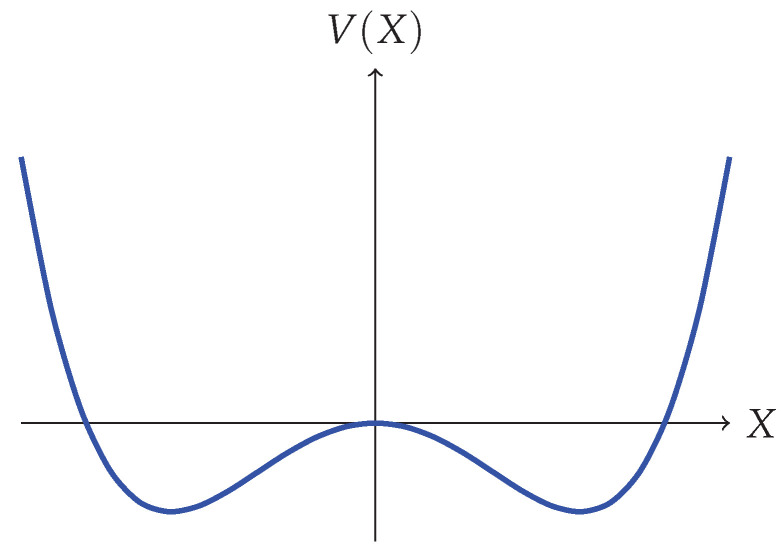
The c=1 matrix model potential. Near X∼0 it resembles an inverted harmonic oscillator, yet it is stablised by the quartic term at large *X*.

**Figure 7 entropy-28-00058-f007:**
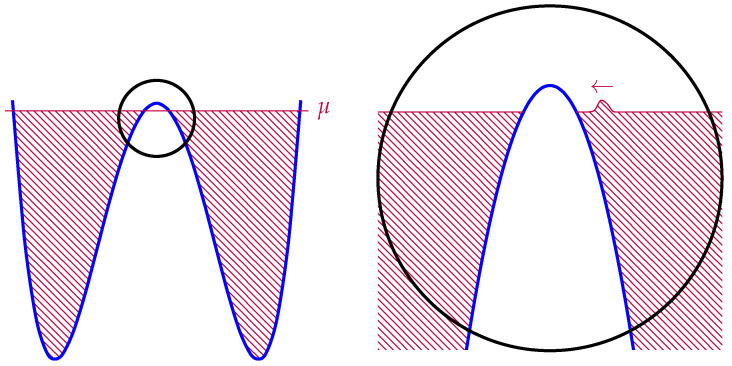
In the double scaling limit we ‘zoom in’ close to the crest of the inverted harmonic potential. The scattering of perturbations of the Fermi sea is dual to closed strings scattering off the Liouville wall.

**Figure 8 entropy-28-00058-f008:**
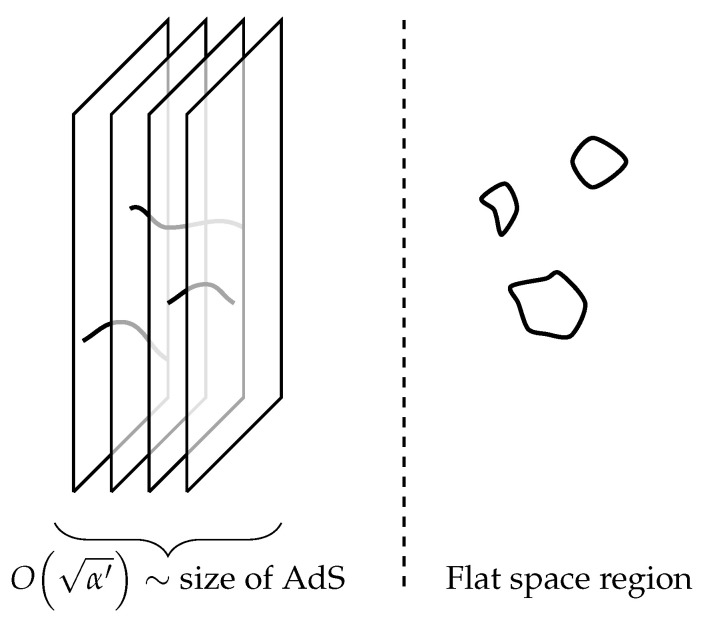
A cartoon of the decoupling limit. The branes, interacting via open strings, spread out over a region the size of the string length scale $\sqrt{\alpha^{\prime }}$. AdS spacetime emerges due to the backreaction of the branes on the ambient space and is found by zooming in to this $O(\sqrt{\alpha^{\prime }})$-sized region. Closed string modes propagating in the flat space region (which we have separated off with a dashed line) decouple from the SYM worldvolume theory as we take N→∞ while keeping $g_{\mathrm{YM}}^{2}N$ fixed.

**Figure 9 entropy-28-00058-f009:**
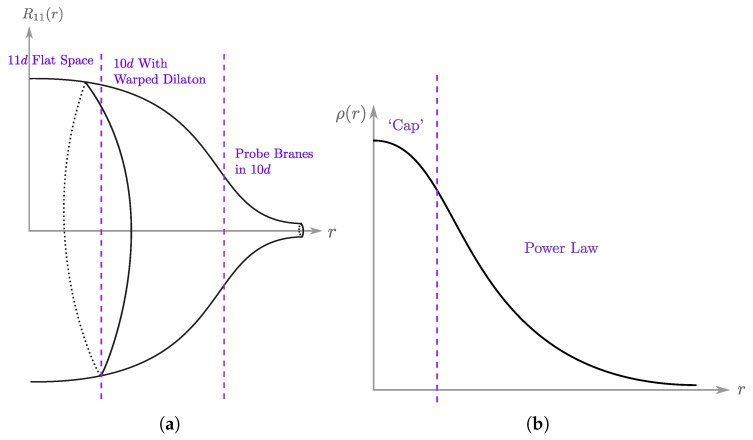
(Left): In (**a**), we draw a schematic sketch of the BFSS geometry—specifically, the radius $R_{11}\left(r\right)$ of the compact eleventh dimension as a function of the 10d radial coordinate *r*. At small radius there are a lot of eigenvlaues which strongly backreact, causing the 11th dimension to become large as in [57]. At intermediate radius the geometry becomes effectively 10 dimensional and we enter the decoupling regime of [13]. At very large radius there are almost no branes, and those that fluctuate out are effectively probe branes in flat 10d space. (Right): In (**b**), we draw a schematic sketch of the BFSS eigenvalue distribution. The large-*r* region follows an approximate power law, $\rho \left(r\right)∼r^{-17}$ [57]. The small-*r* region is expected to cap off this power law, with the density of eigenvalues approaching a constant at small *r*. It is in this constant-density regime that the 11d flat space physics may arise.

**Figure 10 entropy-28-00058-f010:**
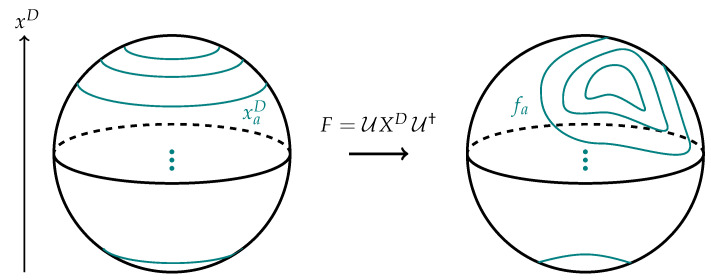
(**Left**): The ordered eigenvalues, $\left\{x_{a}^{D}\right\}$, of a given matrix $X^{D}$ provide a set of definite coordinates on the non-commutative geometry. (**Right**): We can instead use the eigenvalues, $\left\{f_{a}\right\}$, of a polynomial $F=f(X^{i})$, which are the level sets of a function $f(x^{i})$ as a set of curvilinear coordinates. These two are related by a U(N) change of basis, $F=UX^{D}U^{†}$. Figure adapted from [96].

**Figure 11 entropy-28-00058-f011:**
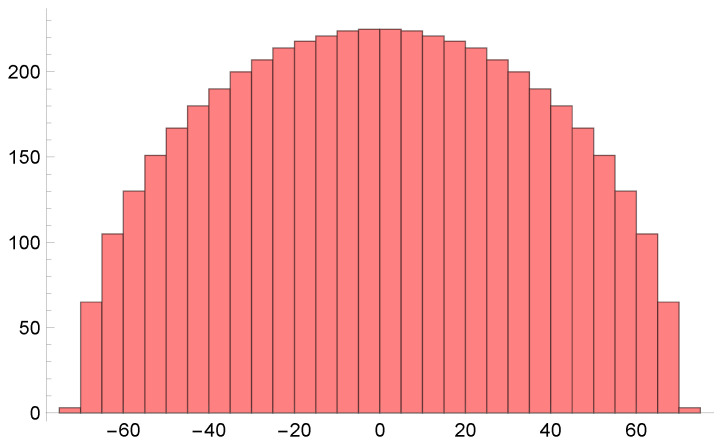
A histogram of the eigenvalue distribution of $X_{\mathrm{cl}}$ for N=5000 and k=1.

**Figure 12 entropy-28-00058-f012:**
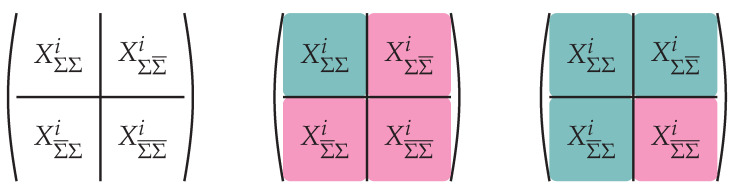
(**Left**): The decomposition of a matrix $X^{i}$ under the projectors (Equation 179) in a basis diagonalizing $f(X^{i})$. (**Middle** and **right**): Two choices of subalgebra associated to Σ as generated by matrix elements in the teal blocks.

**Figure 13 entropy-28-00058-f013:**
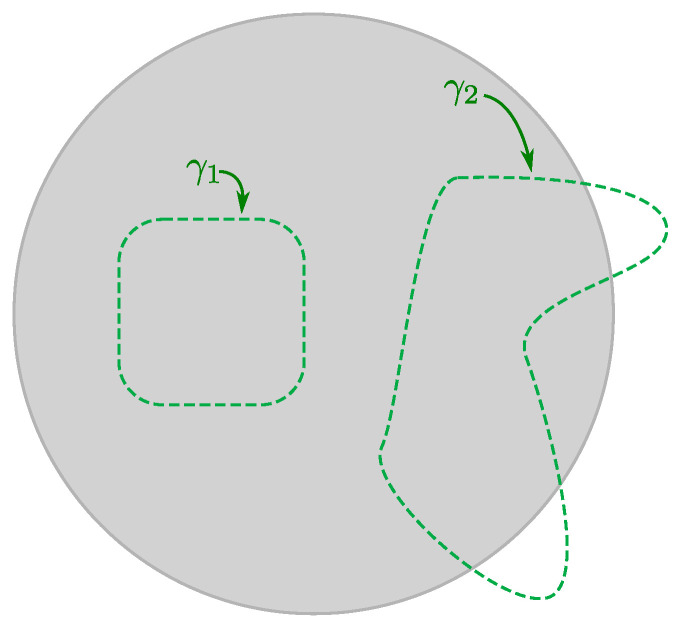
Two distinct choices of entanglement cut for the matrix quantum hall droplet. $\gamma_{1}$ is entirely contained within the droplet, whereas $\gamma_{2}$ crosses the boundary of the droplet. As we deform $\gamma_{1}$ into $\gamma_{2}$, a phase transition in the singular value spectrum of ${\left(Z_{\mathrm{cl}}\right)}_{\Sigma \bar{\Sigma }}$ occurs. For a cut of type $\gamma_{1}$, ${\left(Z_{\mathrm{cl}}\right)}_{\Sigma \bar{\Sigma }}$ will have one singular value of magnitude corresponding to the length $|\gamma_{1}|$. For a cut of type $\gamma_{2}$, ${\left(Z_{\mathrm{cl}}\right)}_{\Sigma \bar{\Sigma }}$ will have O(logN) singular values and pick up additional contributions from collective field fluctuations as in Section 6.3.

**Figure 14 entropy-28-00058-f014:**
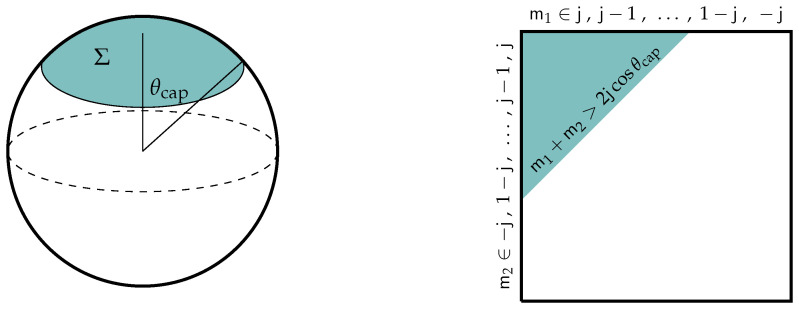
(**Left**): The subregion Σ, depicted in teal, as a cap on the fuzzy sphere, centered at the North pole and determined by an angle $\theta_{\mathrm{cap}}$. (**Right**): The association of matrix elements, shaded in teal, of π and Φ to Σ according to the prescription of [148]. Figure adapted from [148].

**Figure 15 entropy-28-00058-f015:**
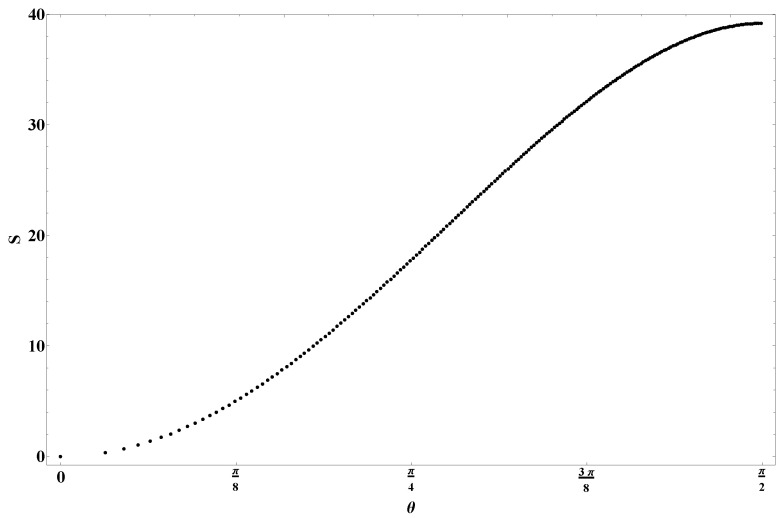
The entanglement entropy, $S_{\Sigma }$, of the ground state (S in the above plot), (Equation 218), plotted against $\theta_{\mathrm{cap}}$ (θ in the above plot) for N=200 and $m^{2}R^{2}=1$. Figure taken from [148].

**Figure 16 entropy-28-00058-f016:**
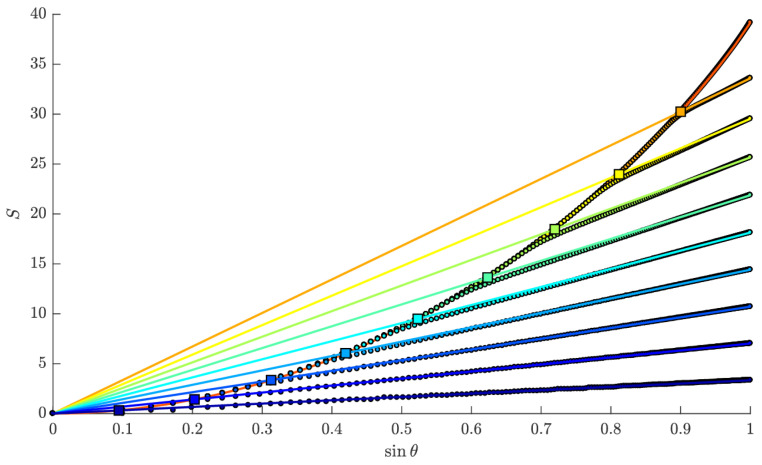
Ground state entanglement entropy, $S_{\Sigma }$ (denoted *S* in the plot) plotted against $\sin\theta_{\mathrm{cap}}$ (denoted sinθ in the plot) for N=200, $m^{2}R^{2}=1$, and *n* ranging from 20 (dark blue) to *N* (red) in increments of twenty. For n<N, linear plots indicating the area law, $S_{\Sigma }=a\sin\theta_{\mathrm{cap}}$ (with $a=S_{\Sigma }{|}_{\theta_{\mathrm{cap}}=\frac{\pi }{2}}$), are present to guide the eye. The intersection of $S_{\Sigma }$ with these linear plots, depicted as square points, defines the transition angle, $\theta_{\mathrm{crit}}$. Figure taken from [150].

**Figure 17 entropy-28-00058-f017:**
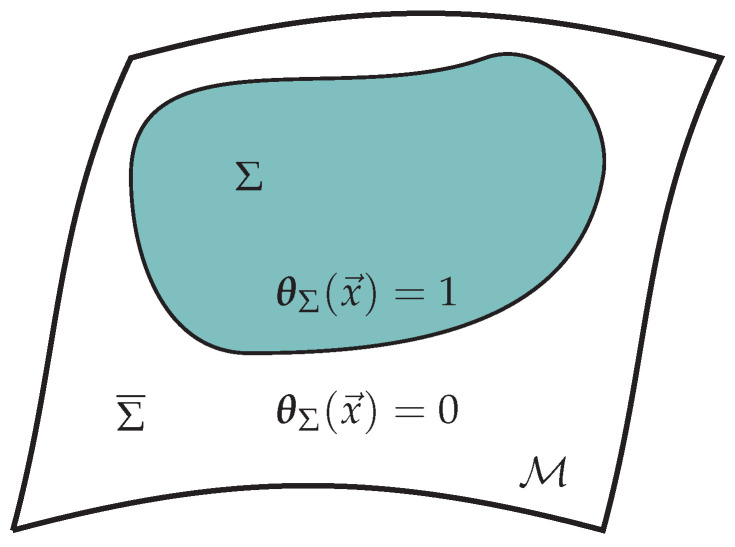
A cartoon of a compact manifold with a subregion Σ depicted in teal. This subregion is defined by a characteristic function $\theta_{\Sigma }$ taking values 1 and 0 inside and outside Σ, respectively.

**Figure 18 entropy-28-00058-f018:**
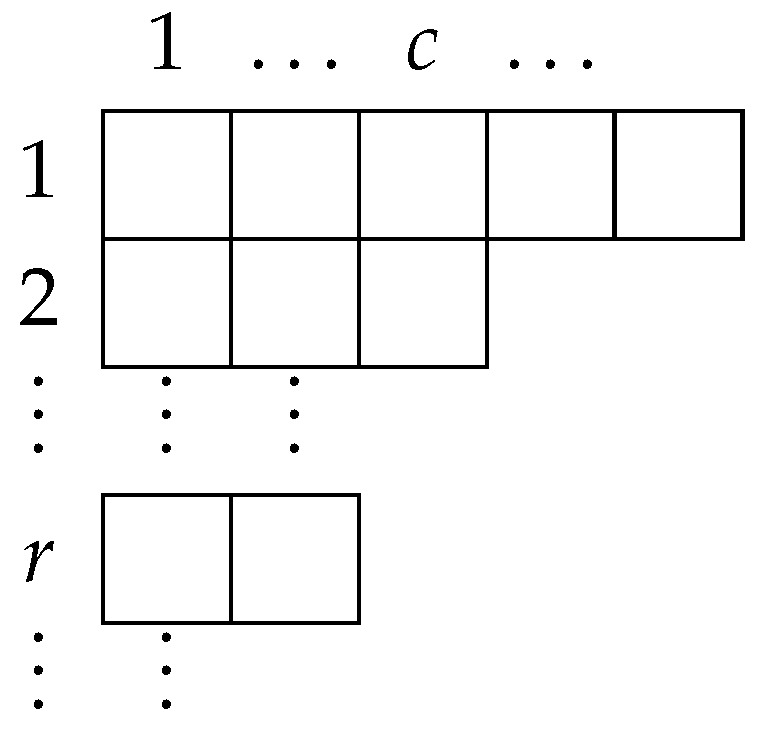
A representative Young diagram corresponding to an irreducible representation of U(M) with rows indexed by *r* and columns indexed by *c*. The row length, $\ell_{r}$ is the number of boxes in a row, the depth, $d_{r,c}$ is the number of boxes underneath the box at (r,c) and the hook length, $h_{r,c}=1+\ell_{r}-c+d_{r,c}$ the depth plus the number of boxes inluding and to the right of (r,c).

**Figure 19 entropy-28-00058-f019:**
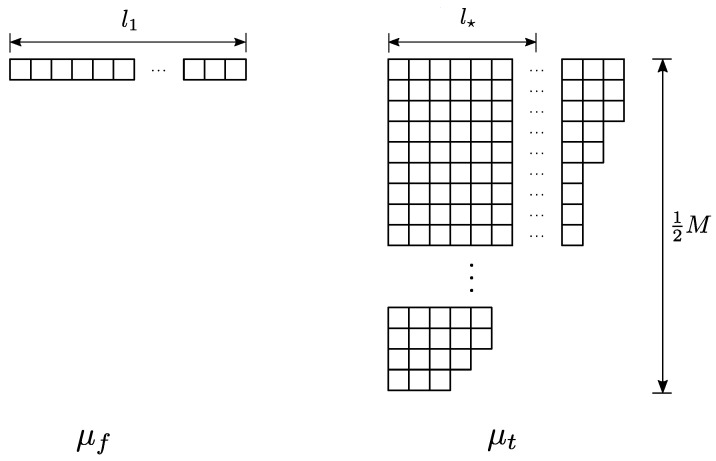
On the left a ‘flat’ Young diagram with a single long row. On the right a ‘tall’ Young diagram with an order M/2 number of rows and typical row length $1\ll l_{★}\ll M/2$. Figure taken from [98].

**Figure 20 entropy-28-00058-f020:**
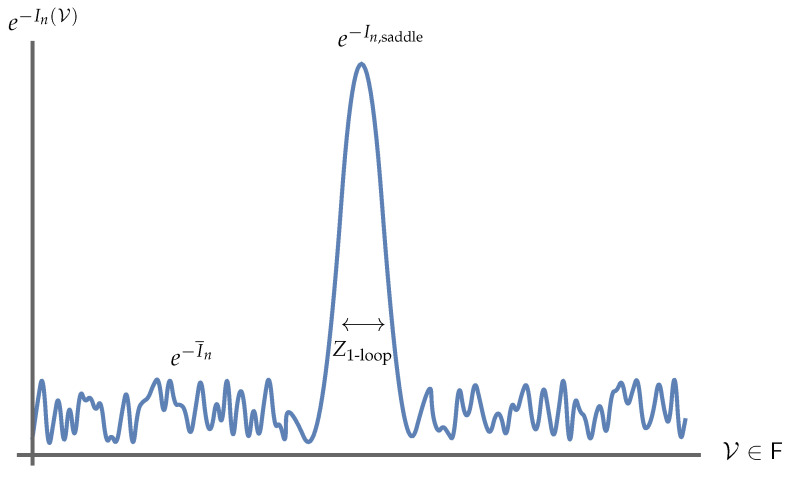
The structure of the frame average integral, $Z_{n}\approx N^{n}\;\int_{F}dV\;e^{-I_{n}\left(V\right)}\;\approx Z_{1\text{-}\mathrm{loop}}\;e^{-I_{n,\mathrm{saddle}}}+e^{-{\bar{I}}_{n}}\;.$ Figure adapted from [98].

**Figure 21 entropy-28-00058-f021:**
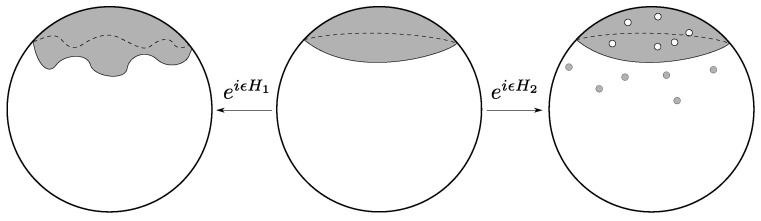
Types of perturbations to Σ. $H_{1}$ perturbs the sole singular value of ${\hat{G}}_{\Sigma }$, corresponding to perturbing Σ by a continuous volume preserving diffeomorphism. $H_{2}$ introduces new singular values corresponding to the production of new connected components of Σ. Figure taken from [98].

**Figure 22 entropy-28-00058-f022:**
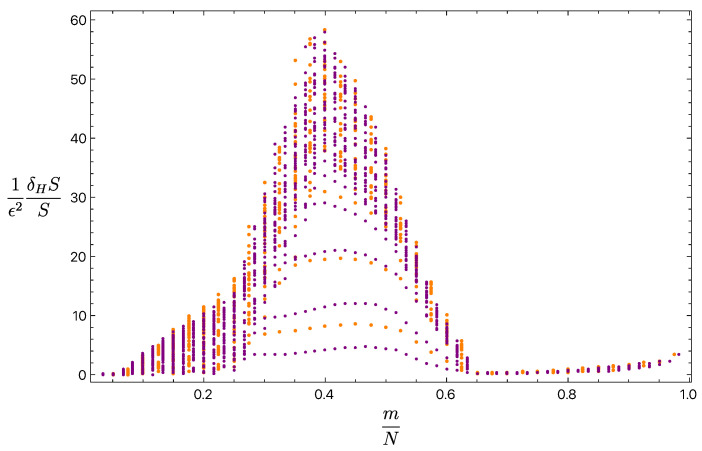
The relative change of the entropy ($\frac{1}{\epsilon^{2}}\frac{\delta_{H}S}{S}=\frac{\sum_{r}\delta_{H}\ell_{r}}{\epsilon^{2}\sum_{r}\ell_{r}}$ in the above figure) under perturbations of various matrix spherical harmonic modes (*m* in the above figure labels the mode, $H={\hat{Y}}_{lm}+{\hat{Y}}_{lm}^{†}$). Modes are color-coded as orange for (N=40, M=16, Λ=10) and purple for (N=60, M=24, Λ=15). The entropy is only increased by these perturbations. Figure taken from [98].

## Data Availability

No new data were created or analyzed in this study. Data sharing is not applicable to this article.

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
