# Peer review of "Matrix Quantum Mechanics and Entanglement Entropy: A Review"

_entropy, 2025, doi:10.3390/e28010058_

Round 1
Reviewer 1 Report
Comments and Suggestions for Authors
The paper is a review of the entanglement entropy, one of most widely used numerical characteristics of quantum entanglement, a basic form of quantum correlations between the parts (subsystems) of a quantum system. Quantum entanglement was discovered by Einstein et al in 1935 and since then proved to be of great importance in a wide variety of branches of modern physics and adjacent sciences, including quantum gravity, string theory, quantum statistical mechanics condensed matter, quantum information science, etc.
The authors confine themselves to the systems whose theoretical description includes large matrices as a basic ingredient, including black holes, branes and quantum metals, to name a few. They focus on the asymptotic behaviour of the entanglement entropy as a function of the size of subsystem, specifically on the validity and interpretation of the so-called area law (asymptotic growth of the entanglement entropy with the size of the boundary, not the volume). This is quite broad and active area of research dealing with numerous problems and approaches, but the authors managed to organize the material in a sufficieently comprehensive way.
I believe that this review paper will of interest and use for researchers of various fields and recomend it to be published.
Author Response
We thank the referee for their review of our manuscript for their favorable determination of its suitability for publishing.
Reviewer 2 Report
Comments and Suggestions for Authors
Please see the attachment.

Author Response
We thank the referee for their very careful reading of our manuscript. We have found many of the changes suggested by the referee helpful for improving the technical and grammatical clarity of our work and we additionally thank them for their care in spotting typos. We have implemented almost all of the suggestions of referee 2, however let us highlight specifically the more substantial changes:
"Eq. 7: Change the order of appearance of the two integrals. Conceptually define g involved with the double scaling limit."
-We added sentence below equation (7) stating that g is a coupling and stating the strong coupling limit.
"Line 197: The concept 'Stringy' was not mentioned in subsection 1.2"
- We changed 'Stringy Edge Modes' to ‘Open String Edge Modes’ in Subsection title
"Eq. 36: I suggest including at least one reference where this supersymmetric transformation is explained."
-We added a reference.
"Eq. 74: What is the acronym NCCS for?"
- This is now defined in the sentence preceding equation (74).
"Figures. 20 and 22: Include in the figure caption the equation from the text linked to the graph."
- Figure captions now have explanatory equations.
"I believe that the most appropriate place for the lines 1954 - 1973 should be in the introduction of the paper.
With the above in mind, I suggest using this space (lines 1954-1973) to briefly describe how the seven parts of the article are interconnected."
-We disagree with the reviewer on this stylistic point. A summary of the content of the article within the conclusion section is standard practice and we maintain that this paragraph makes sufficiently clear how the sections are interconnected.
"Unfortunately, the review does not include reference books, only [116]. For this reason, I suggest including a larger number of bibliographic references."
-Unfortunately, there exist few standard textbooks on matrix quantum mechanics. We believe the standard references of Ginsparg and Moore’s review on 2d gravity, Klebanov’s review on 2d gravity, and Washington Taylor’s review on matrix theory in the context of quantum gravity (all of which are referenced in our review article) function as standard textbooks for this particular topic. Additionally we have included more recent (non-standard) reviews of matrix quantum mechanics, such as Henry Lin's review.
We hope that the referee will find the above changes and replies satisfactory to judge this work suitable for publishing.